# Generation of a $\mu$-1,2-hydroperoxo Fe$^{III}$Fe$^{III}$ and a $\mu$-1,2-peroxo Fe$^{IV}$Fe$^{III}$ Complex

Stephan Walleck[1], Thomas Philipp Zimmermann[1], Henning Hachmeister[2], Christian Pilger [2], Thomas Huser [2], Sagie Katz[3], Peter Hildebrandt [3], Anja Stammler[1], Hartmut Bögge[1], Eckhard Bill [4] & Thorsten Glaser [1✉]

$\mu$-1,2-Peroxo-diferric intermediates (**P**) of non-heme diiron enzymes are proposed to convert upon protonation either to high-valent active species or to activated **P'** intermediates via hydroperoxo-diferric intermediates. Protonation of synthetic $\mu$-1,2-peroxo model complexes occurred at the $\mu$-oxo and not at the $\mu$-1,2-peroxo bridge. Here we report a stable $\mu$-1,2-peroxo complex {Fe$^{III}$($\mu$-O)($\mu$-1,2-O$_2$)Fe$^{III}$} using a dinucleating ligand and study its reactivity. The reversible oxidation and protonation of the $\mu$-1,2-peroxo-diferric complex provide $\mu$-1,2-peroxo Fe$^{IV}$Fe$^{III}$ and $\mu$-1,2-hydroperoxo-diferric species, respectively. Neither the oxidation nor the protonation induces a strong electrophilic reactivity. Hence, the observed intramolecular C-H hydroxylation of preorganized methyl groups of the parent $\mu$-1,2-peroxo-diferric complex should occur via conversion to a more electrophilic high-valent species. The thorough characterization of these species provides structure-spectroscopy correlations allowing insights into the formation and reactivities of hydroperoxo intermediates in diiron enzymes and their conversion to activated **P'** or high-valent intermediates.

[1] Lehrstuhl für Anorganische Chemie I, Fakultät für Chemie, Universität Bielefeld, Universitätsstr. 25, D-33615 Bielefeld, Germany. [2] Biomolekulare Photonik, Fakultät für Physik, Universität Bielefeld, Universitätsstr. 25, D-33615 Bielefeld, Germany. [3] Institut für Chemie, Technische Universität Berlin, Straße des 17. Juni 135, D-10623 Berlin, Germany. [4] Max-Planck-Institut für Chemische Energiekonversion, Stiftstr. 34-36, D-45470 Mülheim an der Ruhr, Germany. ✉email: thorsten.glaser@uni-bielefeld.de

Non-heme diiron enzymes are employed by nature to activate dioxygen for various catalytic oxidation and/or oxygenation reactions[1,2]. Their catalytic cycles generally employ a diferrous form that reacts with dioxygen to a peroxo-diferric intermediate (**P**, Fig. 1a). The active species is supposed to be either this peroxo-diferric species or a species derived from it. In soluble methane monooxygenase (sMMO)[3–5], the peroxo intermediate **P** converts to a high-valent $Fe^{IV}Fe^{IV}$ active species (**Q**, Fig. 1a)[1–8]. Kinetic studies revealed a pH-dependence indicating that this step is proton-promoted[1,3,9,10]. The site of protonation is still unknown. Proposals include protonation of the peroxo ligands resulting in bridging $\mu$-1,1- or $\mu$-1,2-hydroperoxo ligands[2,3,9–11]. In other non-heme diiron enzymes[12–20], a peroxo activation step has been proposed by the conversion of **P**-type to **P′**-type intermediates that lack the peroxo → $Fe^{III}$ LMCT around 14,000–15,000 cm$^{-1}$ and the higher Mössbauer isomer shift characteristic for **P**-type intermediates. For this peroxo activation step, also a protonation has been suggested[15,16,20]. For the two diiron arylamine oxygenases AurF and CmlI, $\mu$-1,2-hydroperoxo[15] and $\mu$-1,1-peroxo intermediates[21] have been proposed, respectively, or a $\mu$-1,1-hydroperoxo intermediate for both (Fig. 1a)[14].

Syntheses and detailed spectroscopy and reactivity studies of $\mu$-1,2-peroxo-diferric model complexes[22–35] provided not only important structure-spectroscopy correlations to establish peroxo intermediates in the enzymes but also variations in their stabilities and reactivities by slight variations of the ligands. In most cases, the $\mu$-1,2-peroxo-diferric species could only be identified spectroscopically as transient intermediates. Interestingly, protonation of different complexes with a $\{Fe^{III}(\mu\text{-}O)(\mu\text{-}1,2\text{-}O_2)Fe^{III}\}$ core afforded $\mu$-hydroxo-bridged $\{Fe^{III}(\mu\text{-}OH)(\mu\text{-}1,2\text{-}O_2)Fe^{III}\}$ species[22,27,36] questioning the principle accessibility of hydroperoxo-diferric species.

Here, we present the synthesis, characterization, and reactivity of the rationally stabilized $\mu$-1,2-peroxo complex $[(\text{susan}^{6\text{-Me}})\{Fe^{III}(\mu\text{-}O)(\mu\text{-}1,2\text{-}O_2)Fe^{III}\}](ClO_4)_2$ using the dinucleating ligand susan$^{6\text{-Me}}$ (Fig. 1b)[37–39]. This $\mu$-1,2-peroxo complex is stable even in solution at $-40\,°C$ and shows nucleophilic character of the $\mu$-1,2-peroxo ligand attenuated for exogenous organic substrates by encapsulation of the ligand scaffold. $[(\text{susan}^{6\text{-Me}})\{Fe^{III}(\mu\text{-}O)(\mu\text{-}O_2)Fe^{III}\}]^{2+}$ is reversibly oxidized to the high-valent $\mu$-1,2-peroxo complex $[(\text{susan}^{6\text{-Me}})\{Fe^{IV}(\mu\text{-}O)(\mu\text{-}1,2\text{-}O_2)$ $Fe^{III}\}]^{3+}$ and reversibly protonated to the $\mu$-1,2-hydroperoxo complex $[(\text{susan}^{6\text{-Me}})\{Fe^{III}(\mu\text{-}O)(\mu\text{-}1,2\text{-}OOH)Fe^{III}\}]^{3+}$. The study of the electrophilic reactivity for oxygen-atom transfer (OAT) using PPh$_3$ and hydrogen-atom transfer (HAT) using DHA and TEMPOH provides not only a low electrophilic character of the parent $\mu$-1,2-peroxo-$Fe^{III}Fe^{III}$ complex but also for the oxidized $\mu$-1,2-peroxo-$Fe^{IV}Fe^{III}$ and protonated $\mu$-1,2-hydroperoxo-$Fe^{III}Fe^{III}$ species. Only the oxidized $\mu$-1,2-peroxo-$Fe^{IV}Fe^{III}$ species reacts with the relatively weak substrate TEMPOH. The determination of the $pK_a = 9.5 \pm 0.1$ and the bond dissociation free energy BDFE(OH)$_{CH_3CN} = 78 \pm 2$ kcal mol$^{-1}$ of the protonated $\mu$-1,2-hydroperoxo-$Fe^{III}Fe^{III}$ species quantifies the low electrophilic character even of the oxidized $\mu$-1,2-peroxo-$Fe^{IV}Fe^{III}$ species. Therefore, the intramolecular C–H activation of preorganized 6-methyl pyridine groups to benzylalcoholato and carboxylato donors in the parent 1,2-peroxo-$Fe^{III}Fe^{III}$ complex should not occur via the 1,2-peroxo-ligand but via conversion to a more reactive but fluent high-valent species. The low electrophilic character and the spectroscopic signatures of this $\mu$-1,2-hydroperoxo-diferric model are discussed in relation to assignments of reactive intermediates postulated for diiron enzymes.

## Results

**The complex $[(\text{susan}^{6\text{-Me}})\{Fe^{III}(\mu\text{-}O)(\mu\text{-}1,2\text{-}O_2)Fe^{III}\}]^{2+}$.** The reaction of susan$^{6\text{-Me}}$ and Fe(ClO$_4$)$_2$·6H$_2$O provided $[(\text{susan}^{6\text{-Me}})$ $\{Fe^{II}(\mu\text{-}OH)_2Fe^{II}\}](ClO_4)_2$ (Fig. S1a) and subsequent reaction with O$_2$ at $-15\,°C$ the $\mu$-1,2-peroxo complex $[(\text{susan}^{6\text{-Me}})$ $\{Fe^{III}(\mu\text{-}O)(\mu\text{-}1,2\text{-}O_2)Fe^{III}\}](ClO_4)_2$ (Fig. 2a). Single-crystal X-ray diffraction provides an asymmetric core structure: the $\mu$-1,2-peroxo ligand is coordinated with O1 trans to a tert-amine (N2) and with O2 trans to a pyridine (N44). This results in a shorter Fe1–O1$^{peroxo}$ and a longer Fe2–O2$^{peroxo}$ bond. The resulting different charge donation is compensated by a longer Fe1–O3$^{oxo}$ and a shorter Fe2–O3$^{oxo}$ bond. The O1–O2 bond length of 1.432(2) Å is the longest yet established for a peroxo-diferric complex (1.396–1.426 Å)[22–26].

Despite this structural asymmetry, the Mössbauer spectrum exhibits one quadrupole doublet with isomer shift $\delta = 0.53$ mm s$^{-1}$ and quadrupole splitting $|\Delta E_Q| = 1.69$ mm s$^{-1}$ (Fig. 2b and Fig. S7). This isomer shift is higher than 0.47 mm s$^{-1}$ of $[(\text{susan}^{6\text{-Me}})$ $\{Fe^{III}F(\mu\text{--}O)Fe^{III}F\}]^{2+}$ (Table 1)[40], which is in line with higher values frequently observed for peroxo-diferric complexes[1,41]. Magnetic measurements (Fig. 2c) revealed an exchange coupling of $J = -155$ cm$^{-1}$ (in the convention $H = -2 J \mathbf{S}_1\mathbf{S}_2$) providing an exact value for a $\mu$-1,2-peroxo,$\mu$-oxo-diferric complex. This antiferromagnetic coupling is significantly stronger than $-100 \pm 20$ cm$^{-1}$ usually found for $\mu$-oxo-diferric complexes[42,43] including those with our dinucleating ligands[38]. This applies also to $\mu$-oxo,$\mu$-carboxylato-diferric complexes which exhibit decreased $\sphericalangle(Fe^{III}\text{-}(\mu\text{-}O)\text{-}Fe^{III})$ angles as the $\mu$-1,2-peroxo,$\mu$-oxo-diferric complex. This comparison indicates a significant contribution of the $Fe^{III}$-$\mu$-1,2-peroxo-$Fe^{III}$ exchange pathway due to shorter (1.88 and 1.93 Å) and hence more covalent $Fe^{III}$-$\mu$-O$^{peroxo}$ bonds[44] compared to the longer $Fe^{III}$-$\mu$-carboxylato bonds (e.g., 1.97 and 2.07 in $[(\text{susan})\{Fe^{III}(\mu\text{-}O)(\mu\text{-}OAc)Fe^{III}\}]^{3+}$)[45] that are considered not to contribute significantly to the exchange coupling.

The UV-Vis-NIR spectrum (Fig. 2d) exhibits three prominent bands at 19,300 cm$^{-1}$ ($\varepsilon = 1180$ M$^{-1}$ cm$^{-1}$), 15,400 cm$^{-1}$ ($\varepsilon = 1000$ M$^{-1}$ cm$^{-1}$), and 11,800 cm$^{-1}$ ($\varepsilon = 190$ M$^{-1}$ cm$^{-1}$) that are characteristic for $\{Fe^{III}(\mu\text{-}O)(\mu\text{-}1,2\text{-}O_2)Fe^{III}\}$ complexes[29] with the 19,300 cm$^{-1}$ band assigned to the $\mu$-oxo→$Fe^{III}$ LMCT and the 15,400 cm$^{-1}$ band to the $\mu$-1,2-peroxo→$Fe^{III}$ LMCT involving $\pi^*_\pi \to t_{2g}$ transitions[11]. This low LMCT energy demonstrates strong covalency of the $Fe^{III}$–O$^{peroxo}$ bond representing the origin of the strong antiferromagnetic coupling.

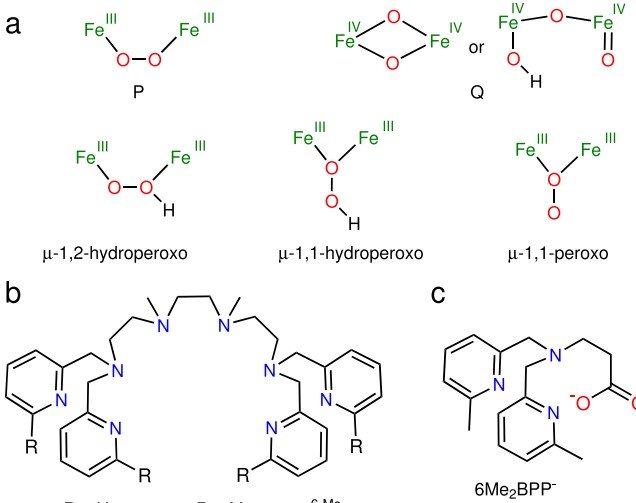

**Fig. 1 Structural formula. a** Supposed intermediates in non-heme diiron enzymes. **b** The dinucleating ligands susan and susan$^{6\text{-Me}}$. **c** The mononucleating ligand 6Me$_2$BPP$^-$.

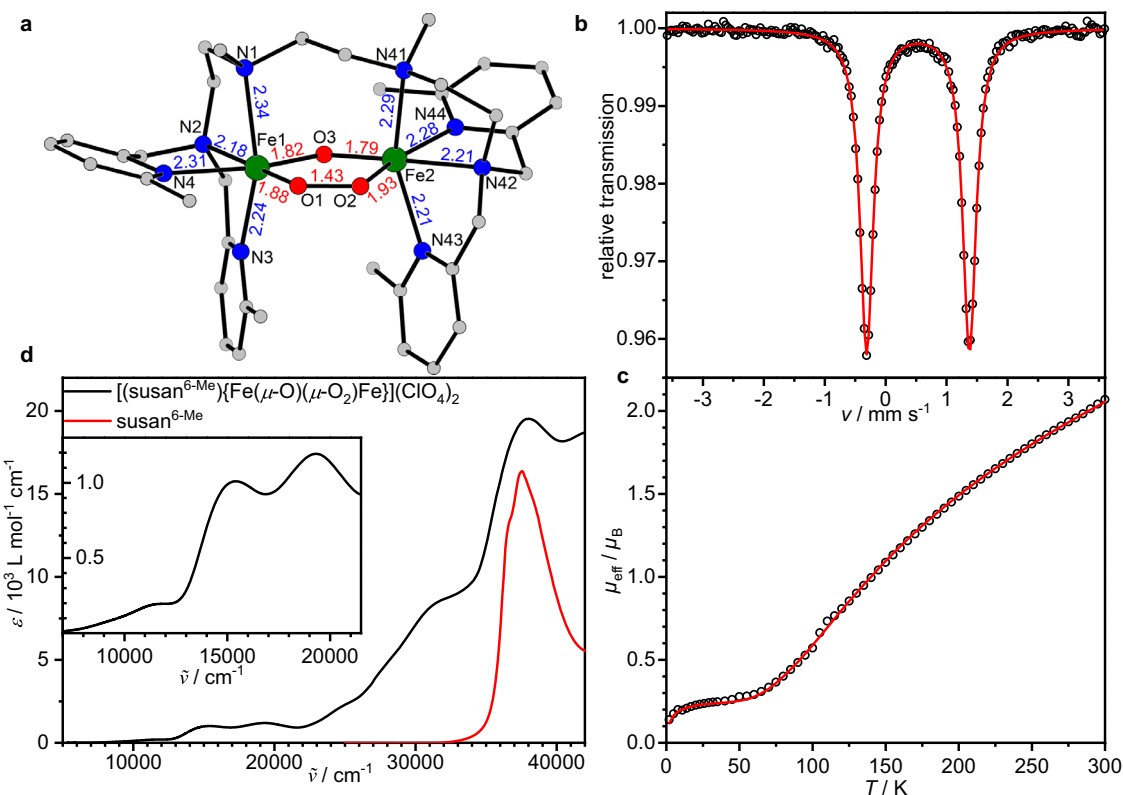

**Fig. 2 Characterization of [(susan$^{6-Me}$){Fe$^{III}$($\mu$-O)($\mu$-O$_2$)Fe$^{III}$}](ClO$_4$)$_2$. a** Molecular structure of [(susan$^{6-Me}$){Fe$^{III}$($\mu$-O)($\mu$-O$_2$)Fe$^{III}$}]$^{2+}$ in single-crystals of [(susan$^{6-Me}$){Fe$^{III}$($\mu$-O)($\mu$-O$_2$)Fe$^{III}$}](ClO$_4$)$_2$•0.85CH$_3$CN•0.7H$_2$O. Hydrogen atoms have been omitted for clarity. **b** $^{57}$Fe Mössbauer spectrum of [(susan$^{6-Me}$){Fe$^{III}$($\mu$-O)($\mu$-O$_2$)Fe$^{III}$}](ClO$_4$)$_2$ at 80 K. The solid line is a simulation with $\delta = 0.53$ mm s$^{-1}$, $|\Delta E_Q| = 1.69$ mm s$^{-1}$, and $\Gamma = 0.26$ mm s$^{-1}$. **c** Temperature-dependence of the effective magnetic moment, $\mu_{eff}$, of [(susan$^{6-Me}$){Fe$^{III}$($\mu$-O)($\mu$-O$_2$)Fe$^{III}$}](ClO$_4$)$_2$. The solid line is a simulation to the spin-Hamiltonian (1) in the Supplementary Information with $J = -155$ cm$^{-1}$, $g_i = 2.05$, 0.2% p.i. ($S = 5/2$) of same molecular mass and $\Theta_{w,pi} = -8$ K. **d** UV-Vis-NIR spectrum of [(susan$^{6-Me}$){Fe$^{III}$($\mu$-O)($\mu$-O$_2$)Fe$^{III}$}](ClO$_4$)$_2$ dissolved in CH$_3$CN at −10 °C and the ligand susan$^{6-Me}$ for comparison.

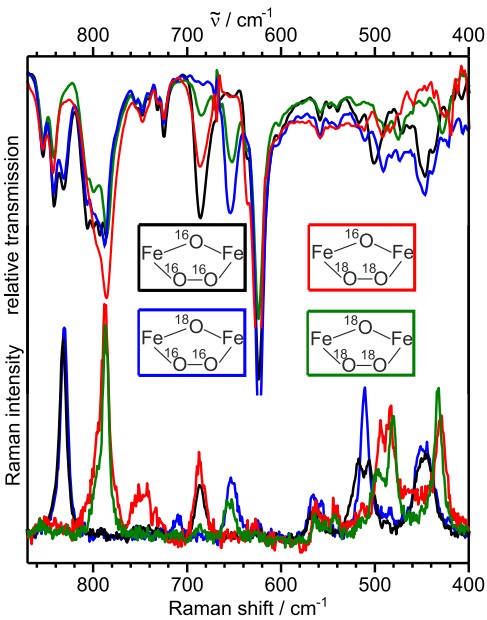

**Fig. 3 Vibrational characterization of [(susan$^{6-Me}$){Fe$^{III}$($\mu$-O)($\mu$-O$_2$) Fe$^{III}$}](ClO$_4$)$_2$.** Resonance Raman (bottom, 633 nm (15,800 cm$^{-1}$) excitation) and FTIR spectra (top) measured on solids at room temperature for the four different isotopomers of [(susan$^{6-Me}$){Fe$^{III}$($\mu$-O)($\mu$-O$_2$)Fe$^{III}$}] (ClO$_4$)$_2$ indicated by the representation of their central cores and the color scheme used.

We synthesized all four possible $^{18}$O/$^{18}$O$_2$-isotopomers as microcrystalline solids. The resonance Raman (rR) and FTIR spectra (Fig. 3) both reveal several $^{18}$O-sensitive vibrations, which allow their assignments to the {Fe$^{III}$($\mu$-O)($\mu$-1,2-O$_2$)Fe$^{III}$} core (Table 2)[22,29]. Only slight changes are observed upon dissolution in CH$_3$CN (Fig. S9). The 831 cm$^{-1}$ band for the two $^{16}$O$_2$-isotopomers is assigned to the $\nu$(O–O) stretch by isotopic labeling ($\Delta$($^{16}$O$_2$–$^{18}$O$_2$) = 46 cm$^{-1}$ consistent with a Hooke's law calculation for a harmonic O–O vibration of 48 cm$^{-1}$).

The $\nu$(O–O) stretch of the crystallographically characterized [(6Me$_2$BPP){Fe$^{III}$($\mu$-O)($\mu$-1,2-O$_2$)Fe$^{III}$}(6Me$_2$BPP)] (**A**) appears at higher energy at 847 cm$^{-1}$ (Table 2)[22]. Higher $\nu$(O–O) stretches were attributed[22] to increasing $\sphericalangle$(Fe–O–O) angles[11]. However, $\sphericalangle$(Fe–O–O) is slightly smaller in **A** than in our complex (115° vs 117°). Interestingly, the lower $\nu$(O–O) stretch correlates with a longer O–O distance (1.432(2) Å vs 1.411 Å) indicating a stronger dependence of $\nu$(O–O) on $d$(O–O) than on $\sphericalangle$(Fe–O–O).

[(susan$^{6-Me}$){Fe$^{III}$($\mu$-O)($\mu$-1,2-O$_2$)Fe$^{III}$}]$^{2+}$ shows no indication of decay for hours in CH$_3$CN at −40 °C (Fig. S10). This stability provides the opportunity for the electro- and spectroelectrochemical investigation of a peroxo-diferric complex. [(susan$^{6-Me}$){Fe$^{III}$($\mu$-O) ($\mu$-O$_2$)Fe$^{III}$}]$^{2+}$ can be reversibly oxidized at $E_{1/2}^{ox} = 0.55$ V and irreversibly reduced at $E_p^{red} = -1.28$ V vs Fc$^+$/Fc (Fig. 4a).

**Oxidation to [(susan$^{6-Me}$){Fe($\mu$-O)($\mu$-1,2-O$_2$)Fe}]$^{3+}$.** Coulometric oxidation of [(susan$^{6-Me}$){Fe$^{III}$($\mu$-O)($\mu$-1,2-O$_2$)Fe$^{III}$}]$^{2+}$ at 0.68 V vs Fc$^+$/Fc (1.18 C, 98% of one-electron) in CH$_3$CN at −40 °C resulted in slight changes of the absorption features

**Table 1 Compilation of experimental (on solids or on frozen $CH_3CN$ solutions) and DFT-calculated (shown in italics) $^{57}Fe$ Mössbauer parameters of the complexes reported here and some complexes with the ligands susan and susan$^{6-Me}$ for comparison.**

|  |  | $\delta$/mm s$^{-1}$ | $|\Delta E_Q|$/mm s$^{-1}$ | T/K | ref. |
|---|---|---|---|---|---|
| [(susan$^{6-Me}$){Fe$^{II}$($\mu$-OH)$_2$Fe$^{II}$}]$^{2+}$ | Solid | 1.13 | 2.23 | 80 | a |
| [(susan$^{6-Me}$){Fe$^{III}$($\mu$-O)($\mu$-1,2-O$_2$)Fe$^{III}$}]$^{2+}$ | Solid | 0.53 | 1.69 | 80 | a |
|  | CH$_3$CN | 0.53 | 1.68 | 80 | a |
|  | Solid | 0.49 | 1.68 | 200 | a |
|  | *DFT: Fe1* | *0.58* | *−1.28* |  | a |
|  | *Fe2* | *0.53* | *−1.45* |  |  |
| [(susan$^{6-Me}$){Fe$^{III}$($\mu$-O)($\mu$-1,2-O$_2$)Fe$^{IV}$}]$^{3+}$ | CH$_3$CN | 0.27 | +0.57 | 180 | a |
|  |  | 0.39 | −1.29 |  |  |
| *DFT: Fe$^{IV}$1 Fe$^{III}$2 configuration* | *Fe$^{IV}$1* | *0.17* | *+1.00* |  | a |
|  | *Fe$^{III}$2* | *0.43* | *−1.21* |  |  |
| *DFT: Fe$^{III}$1 Fe$^{IV}$2 configuration* | *Fe$^{IV}$2* | *0.14* | *+0.67* |  | a |
|  | *Fe$^{III}$1* | *0.43* | *−1.00* |  |  |
| [(susan$^{6-Me}$){Fe$^{III}$($\mu$-O)($\mu$-1,2-OOH)Fe$^{III}$}]$^{3+}$ | CH$_3$CN | 0.49 | 2.48 | 80 | a |
|  |  | 0.45 | 1.37 |  |  |
| *DFT: $\mu$-peroxo-O1 protonated* | *Fe1* | *0.54* | *−1.69* |  | a |
|  | *Fe2* | *0.47* | *−1.38* |  |  |
| *DFT: $\mu$-peroxo-O2 protonated* | *Fe1* | *0.49* | *+1.05* |  | a |
|  | *Fe2* | *0.48* | *−2.51* |  |  |
| *DFT: $\mu$-oxo-O3 protonated* | *Fe1* | *0.61* | *−1.74* |  | a |
|  | *Fe2* | *0.56* | *−0.97* |  |  |
| [(susan$^{6-Me}$){Fe$^{III}$F($\mu$-O)Fe$^{III}$F}]$^{2+}$ | Solid | 0.47 | 1.38 | 80 | 40 |
| [(susan){Fe$^{III}$F($\mu$-O)Fe$^{III}$F}]$^{2+}$ | Solid | 0.45 | 1.68 | 80 | 61 |
| [(susan){Fe$^{III}$(OH)($\mu$-O)Fe$^{III}$(OH)}]$^{2+}$ | CH$_3$CN | 0.45 | 1.71 | 80 | 62 |

aThis work
Quadrupole splittings provided without a sign represent absolute values $|\Delta E_Q|$ obtained from zero-field Mössbauer spectra. The signs provided for [(susan$^{6-Me}$){Fe$^{III}$($\mu$-O)($\mu$-1,2-O$_2$)Fe$^{IV}$}]$^{3+}$ were extracted from fits to the magnetic Mössbauer spectra, while signs provided for DFT-calculated values arise from the DFT calculations. Atom labeling scheme related to Fig. 2a.

**Table 2 Vibrational (resonance Raman and FTIR) frequencies (cm$^{-1}$) and isotopic shifts together with some metrical parameters for [(susan$^{6-Me}$){Fe$^{III}$($\mu$-O)($\mu$-1,2-O$_2$)Fe$^{III}$}]$^{2+}$ and comparisons to those for A.**

|  | exp. rR | exp. FTIR | A$^{22}$ |
|---|---|---|---|
| $\nu$(O-O) [$\Delta^{18}$O, $\Delta^{18}$O$_2$, $\Delta$($^{18}$O,$^{18}$O$_2$)] | 831 [0, −46, −46] | 831 [0, −46, −46] | 847 [−, −33, −] |
| $\nu_{as}$(Fe-O-Fe) [$\Delta^{18}$O, $\Delta^{18}$O$_2$, $\Delta$($^{18}$O,$^{18}$O$_2$)] | 687 [−35, 0, −37] | 686 [−32, 0, −34] | 695 [−, −2, −] |
| $\nu_s$(Fe-O-Fe) [$\Delta^{18}$O, $\Delta^{18}$O$_2$, $\Delta$($^{18}$O,$^{18}$O$_2$)] | ≈ 510 [0, −25,−] | n.o. | n.o. |
| $\nu_{as}$(Fe-O$_2$-Fe) [$\Delta^{18}$O, $\Delta^{18}$O$_2$, $\Delta$($^{18}$O,$^{18}$O$_2$)] | 511 (517/506)$^a$ [−1, −23, −23] | n.o. | n.o. |
| $\nu_s$(Fe-O$_2$-Fe) [$\Delta^{18}$O, $\Delta^{18}$O$_2$, $\Delta$($^{18}$O,$^{18}$O$_2$)] | 448 [−1, −23$^a$, −23$^a$] | 447 [0, −19, −19] | 465 [−, −19, −] |
| d(O-O)/Å | 1.432(2) |  | 1.41$^b$ |
| d(Fe-($\mu$-O))/Å | 1.875(1)/1.928(1) |  | 1.72$^b$/1.74$^b$ |
| d(Fe-($\mu$-1,2-O$_2$))/Å | 1.824(1)/1.790(1) |  | 2.07$^b$/2.10$^b$ |
| ∠(Fe-O-O)/° | 116.4(1)/117.5(1) |  | 114.7$^b$/115.3$^b$ |

aSplit by Fermi-resonance.
bThe $\mu$-oxo and $\mu$-1,2-peroxo groups are disordered$^{22}$ prohibiting a rigid comparison.

(Fig. 4b). Re-reduction at 0.16 V vs Fc$^+$/Fc (0.92 C, 76% of one-electron) restored the initial UV-Vis spectrum to ~95% (Fig. 4c and Fig. S11). The lower charge necessary for re-reduction implies some chemical reduction during the coulometric experiments (1 h, Fig. S12).

Chemical oxidation with thianthrenium perchlorate ((thia)ClO$_4$) generated the UV-Vis spectrum of [(susan$^{6-Me}$){Fe($\mu$-O)$_2$($\mu$-O$_2$)Fe}]$^{3+}$ (Fig. 4d) within 10 s. Addition of excess NEt$_3$ as reductant regenerated the starting spectrum. This emphasizes again the chemical reversibility of the oxidation and indicates the conservation of the $\mu$-oxo,$\mu$-1,2-peroxo motive in the oxidized species.

**Characterization of [(susan$^{6-Me}$){Fe($\mu$-O)($\mu$-1,2-O$_2$)Fe}]$^{3+}$.** The EPR spectrum of [(susan$^{6-Me}$){Fe($\mu$-O)($\mu$-1,2-O$_2$)Fe}]$^{3+}$ (Fig. 5a, S13) shows a broad anisotropic $S_t = 1/2$ signal with **g** = (2.272,

2.152, 2.021). Mössbauer spectroscopy (vide infra) demonstrates that the iron ions remain high-spin ruling out an interpretation as Fe$^{III}$ l.s. species. The severe deviation of $g_{av}$ = 2.15 from 2.0023 and the large g-anisotropy demonstrate a strong contribution of orbital angular momentum and rule out a ligand-centered oxidation to a $\mu$-oxo,$\mu$-1,2-superoxo-Fe$^{III}$Fe$^{III}$ complex, disclosing a metal-centered oxidation to a $\mu$-oxo,$\mu$-1,2-peroxo-Fe$^{IV}$Fe$^{III}$ complex with antiferromagnetically coupled Fe$^{IV}$($S_1$ = 2) and Fe$^{III}$($S_2$ = 5/2) ions. The Fe$^{IV}$ h.s. $S_1$ = 2 configuration results in $g_1 < 2.0$ by spin-orbit coupling, while the Fe$^{III}$ h.s. is close to isotropic $g_2 \approx 2.00$[46]. Projection of the local spins onto the antiferromagnetic $S_t = 1/2$ ground state[47] results in $g_{av} > 2.00$ for the $S_t = 1/2$ as observed experimentally[39]. Despite significant efforts, we were not able to obtain resonance-enhanced Raman features of [(susan$^{6-Me}$){Fe($\mu$-O)($\mu$-1,2-O$_2$)Fe}]$^{3+}$ by excitation at 647 (15,500), 568 (17,600), and 514 nm (19,500 cm$^{-1}$) corresponding to the three absorption features of this oxidized complex.

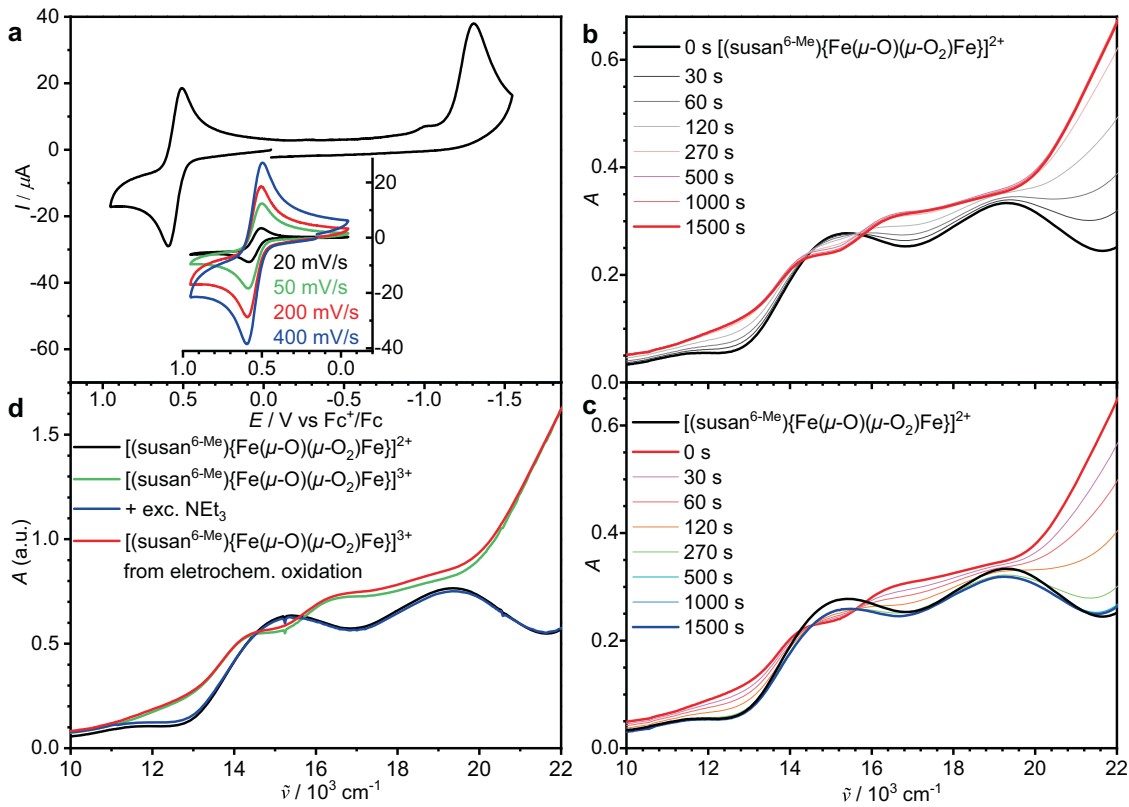

**Fig. 4 Electrochemical/spectroelectrochemical characterization of [(susan$^{6\text{-Me}}$){Fe$^{III}$($\mu$-O)($\mu$-O$_2$)Fe$^{III}$}](ClO$_4$)$_2$ and oxidation to [(susan$^{6\text{-Me}}$){Fe$^{IV}$($\mu$-O)($\mu$-1,2-O$_2$)Fe$^{III}$}]$^{3+}$. a** Cyclic voltammograms of [(susan$^{6\text{-Me}}$){Fe$^{III}$($\mu$-O)($\mu$-O$_2$)Fe$^{III}$}](ClO$_4$)$_2$ at −20 °C in CH$_3$CN solution (0.2 M (NBu$_4$)PF$_6$) recorded at a GC working electrode. Scan rate 200 mV s$^{-1}$ unless noted otherwise. Spectroelectrochemical measurements at −40 °C in CH$_3$CN (0.2 mM with 0.1 M (NBu$_4$)PF$_6$) during **b** oxidation at 0.68 V vs Fc$^+$/Fc and **c** re-reduction at 0.16 V vs Fc$^+$/Fc. **d** Chemical oxidation with one equivalent (thia)ClO$_4$ at −60 °C in CH$_3$CN/CH$_2$Cl$_2$ (1:1) and re-reduction with excess NEt$_3$. The chemically oxidized species (green) almost superimpose with the electrochemically oxidized species (red).

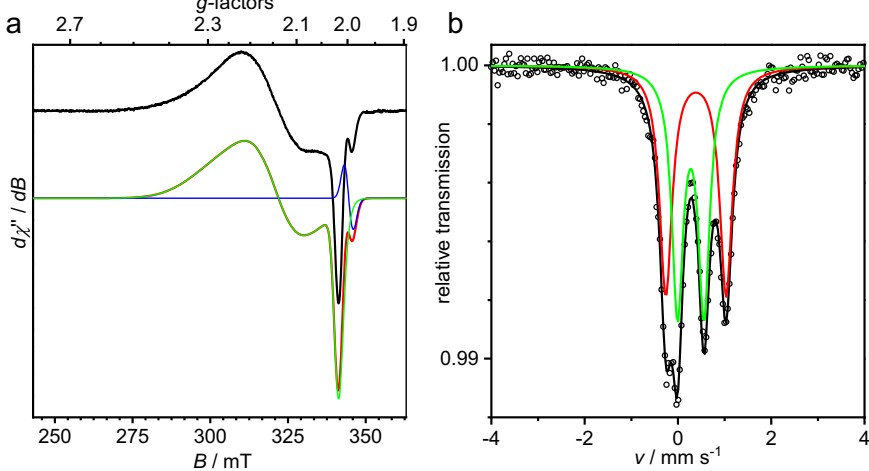

**Fig. 5 Spectroscopic characterization of the oxidized complex [(susan$^{6\text{-Me}}$){Fe$^{IV}$($\mu$-O)($\mu$-O$_2$)Fe$^{III}$}]$^{3+}$. a** X-band EPR spectrum of chemically oxidized [(susan$^{6\text{-Me}}$){Fe$^{IV}$($\mu$-O)($\mu$-O$_2$)Fe$^{III}$}]$^{3+}$ in ≈0.4 mM CH$_3$CN solution at 10 K (top, 9.63312 GHz, 80 $\mu$W power, 0.75 mT modulation) and its simulation (bottom; simulation red trace, green and blue traces subspectra) with the parameters provided in the text. **b** $^{57}$Fe Mössbauer spectra of $^{57}$Fe-enriched, chemically oxidized [(susan$^{6\text{-Me}}$){Fe$^{IV}$($\mu$-O)($\mu$-O$_2$)Fe$^{III}$}]$^{3+}$ in frozen CH$_3$CN solution at 180 K. The solid lines are simulations with parameters provided in Table 1.

The 180 K Mössbauer spectrum of $^{57}$Fe-labeled [(susan$^{6\text{-Me}}$){Fe($\mu$-O)($\mu$-1,2-O$_2$)Fe}]$^{3+}$ (Fig. 5b) exhibits a 4-line spectrum suggesting the presence of two quadrupole doublets. Two different fit models are possible, but considerations explained in the Supplementary

Information (Fig. S14 and S15) strongly favor the model with $\delta_1 = 0.39$ mm s$^{-1}$/$\Delta E_{Q1} = -1.29$ mm s$^{-1}$ and $\delta_2 = 0.27$ mm s$^{-1}$ / $\Delta E_{Q2} = +0.57$ mm s$^{-1}$ (Table 1). Interestingly, adding excess NEt$_3$ as reductant to the oxidized [(susan$^{6\text{-Me}}$){Fe($\mu$-O)($\mu$-1,2-O$_2$)Fe}]$^{3+}$

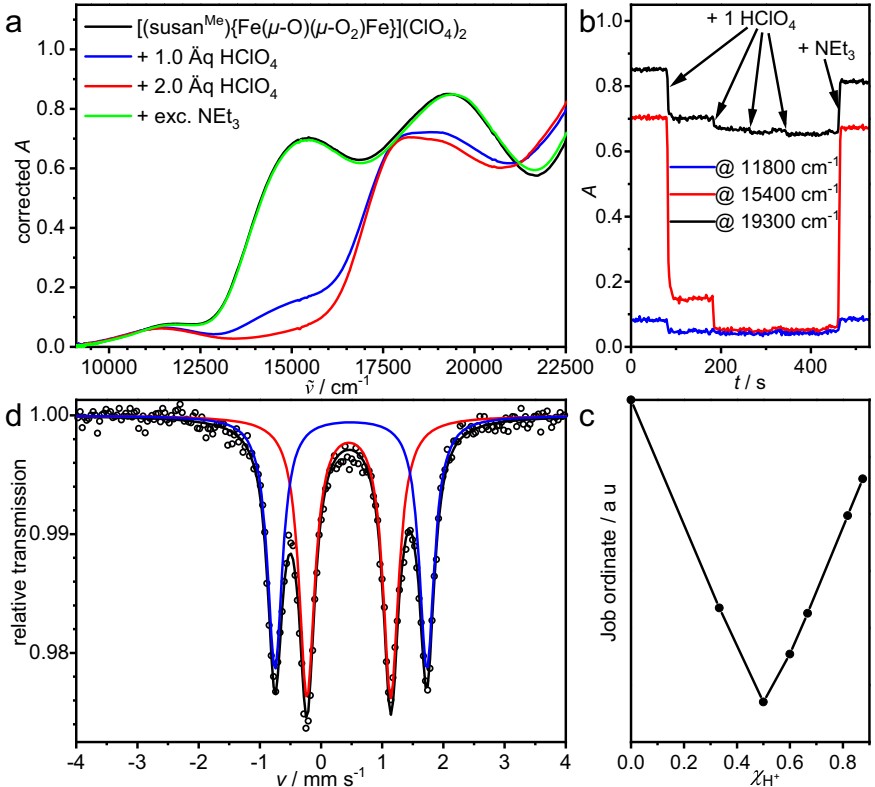

**Fig. 6 Protonation of [(susan$^{6\text{-Me}}$){Fe$^{III}$($\mu$-O)($\mu$-1,2-O$_2$)Fe$^{III}$}]$^{2+}$ and spectroscopic characterization of the protonated complex [(susan$^{6\text{-Me}}$){Fe$^{III}$($\mu$-O)($\mu$-1,2-OOH)Fe$^{III}$}]$^{3+}$. a** UV-Vis spectroscopic characterization of protonation and deprotonation of [(susan$^{6\text{-Me}}$){Fe$^{III}$($\mu$-O)($\mu$-1,2-O$_2$)Fe$^{III}$}]$^{2+}$ in CH$_3$CN/CH$_2$Cl$_2$ (1:2) at −60 °C ($c = 0.77$ mM). **b** Selected time traces for the protonation/deprotonation in **a**. **c** Job-plot analysis at 15,400 cm$^{-1}$ of the protonation in **a**. **d** $^{57}$Fe Mössbauer spectrum of a frozen solution of [(susan$^{6\text{-Me}}$){Fe$^{III}$($\mu$-O)($\mu$-1,2-OOH)Fe$^{III}$}]$^{3+}$ at 80 K generated by treating [(susan$^{6\text{-Me}}$){Fe$^{III}$($\mu$-O)($\mu$-1,2-O$_2$)Fe$^{III}$}]$^{2+}$ with 1.5 equivalents HClO$_4$ in CH$_3$CN at −40 °C. The solid lines are simulations with parameters provided in Table 1.

restores the Mössbauer spectrum of the starting complex [(susan$^{6\text{-Me}}$){Fe$^{III}$($\mu$-O)($\mu$-O$_2$)Fe$^{III}$}]$^{2+}$ (Fig. S16) confirming the chemical reversibility.

The decrease of the isomer shift from 0.53 mm s$^{-1}$ of the starting complex to 0.27 and 0.39 mm s$^{-1}$ confirms a mainly metal-centered oxidation to a high-valent $\mu$-1,2-peroxo complex with both Fe$^{III}$ ions involved resulting in a mixed-valence Fe$^{IV}$Fe$^{III}$ species. In the Robin-and-Day classification for mixed-valence systems[48], class I implies no interaction (ruled out by the coupled $S_t = 1/2$ spin ground state) while class III stands for quantum-mechanically delocalized states. In class II systems, two different states exist that correspond roughly to the excess electron localized on the one or the other metal ion. Between these two states is an energy barrier and there can be temperature-dependent and light-induced mechanisms to transfer the excess electron from the reduced to the oxidized metal ion. In symmetric cases, the two states are energetically degenerate while the asymmetry observed here results in an energy difference between these two states (Fig. S17). The two quadrupole doublets can arise from one of these states populated exclusively up to 180 K or from an electron hopping between these two states at a rate faster than the Mössbauer timescale (10$^{-7}$ s). To differentiate between these two possibilities, we recorded Mössbauer spectra at lower temperatures (Fig. S18). Unfortunately, the spectra broadened with decreasing temperature due to a relaxation process that is fast relative to the Mössbauer timescale at only 180 K. The origin cannot only arise from a decrease of the electron hopping rate but also from paramagnetic effects. DFT calculations (Supplementary Information) provided two localized configurations Fe$^{IV}$1Fe$^{III}$2 and Fe$^{III}$1Fe$^{IV}$2 that both reproduce

the isomer shift decrease of both iron ions (Table 1) and hence confirm the assignment to class II. However, although these DFT calculations provided Fe$^{IV}$1Fe$^{III}$2 being lower in energy by ≈820 cm$^{-1}$, more advanced MO calculations are required to obtain further insight that are beyond this study.

**Protonation to [(susan$^{6\text{-Me}}$){Fe$^{III}$($\mu$-O)($\mu$-1,2-OOH)Fe$^{III}$}]$^{3+}$.** Treatment of a solution of [(susan$^{6\text{-Me}}$){Fe$^{III}$($\mu$-O)($\mu$-1,2-O$_2$)Fe$^{III}$}]$^{2+}$ with HClO$_4$ at −60 °C resulted in the loss of the 15,400 cm$^{-1}$ band while the 11,800 and 19,300 cm$^{-1}$ bands are only slightly affected (Fig. 6a, b). A Job plot analysis[49,50] (Fig. 6c) provided a 1:1 stoichiometry for the reaction between the $\mu$-1,2-peroxo complex and H$^+$. Adding NEt$_3$ as a base restores the initial spectrum (Fig. 6a) showing the reversibility of this protonation.

**Characterization of [(susan$^{6\text{-Me}}$){Fe$^{III}$($\mu$-O)($\mu$-1,2-OOH)Fe$^{III}$}]$^{3+}$.** The disappearance of the $\mu$-1,2-peroxo→Fe$^{III}$ LMCT at 15400 cm$^{-1}$, while the $\mu$-oxo→Fe$^{III}$ LMCT around 19,000 cm$^{-1}$ persists, is consistent with protonation of the $\mu$-1,2-peroxo ligand. In contrast, protonation of complex **A** to the $\mu$-hydroxo,$\mu$-1,2-peroxo complex is accompanied by a shift of the $\mu$-1,2-peroxo→Fe$^{III}$ LMCT from 17300 cm$^{-1}$ to 15,500 cm$^{-1}$[22]. A similar shift was observed using a linear N$_4$ ligand[36]. Moreover, the persistence of the weaker feature at 11800 cm$^{-1}$ assigns this to a $\mu$-oxo→Fe$^{III}$ LMCT. However, the absence of a $\mu$-1,2-hydroperoxo→Fe$^{III}$ LMCT prohibits excitation for rR spectra at 647 nm (15,500 cm$^{-1}$). Neither were resonance-enhanced vibrations observed by excitation at 568

(17,600) and 514 nm (19,500 cm$^{-1}$) close to the $\mu$-oxo$\rightarrow$Fe$^{III}$ LMCT (Fig. 6a).

The Mössbauer spectrum provided two quadrupole doublets with $\delta_1 = 0.49 \, / \, |\Delta E_Q|_1 = 2.48$ mm s$^{-1}$ and $\delta_2 = 0.45 \, / \, |\Delta E_Q|_2 = 1.37$ mm s$^{-1}$ (Fig. 6d). Since the parent [(susan$^{6\text{-}Me}$){Fe$^{III}$($\mu$-O)($\mu$-1,2-O$_2$)Fe$^{III}$}]$^{2+}$ exhibits only one quadrupole doublet despite the significant differences in its two coordination sites (vide supra) requires a different source of asymmetry for the protonated species to explain its two strongly differing quadrupole doublets. This rules out a $\mu$-1,1-hydroperoxo-bridge, while a $\mu$-1,2-hydroperoxo-bridge provides the different source of asymmetry as the protonated site becomes much less charge-donating. Although a terminally bound hydroperoxo-ligand would be in-line with the absence of a $\mu$-1,2-peroxo$\rightarrow$Fe$^{III}$ LMCT and two quadrupole doublets, the persistence of the $\mu$-oxo $\rightarrow$ Fe$^{III}$ LMCTs around 19,000 and 11,800 cm$^{-1}$ strongly favors a doubly-bridged structure of almost the same $\sphericalangle$(Fe$^{III}$-($\mu$-O)-Fe$^{III}$) angle hence ruling out an almost linear {Fe$^{III}$(OOH)($\mu$-O)Fe$^{III}$X} core[38,40] that is also inaccessible with the ligand susan$^{6\text{-}Me}$ (vide infra).

The formation of a $\mu$-1,2-hydroperoxo-bridged complex is supported by DFT calculations (Supplementary Information). Geometry optimizations were achieved for three different tautomers with protonation of the $\mu$-peroxo-O1 being 1200 cm$^{-1}$ higher in energy. Although the $\mu$-hydroxo species (protonation of the $\mu$-oxo-O3) seems energetically feasible, its isomer shifts are even higher than for the starting peroxo complex (Table 1). In contrast, protonation at the $\mu$-peroxo-O2 decreases the isomer shifts as experimentally observed. Thus, the $\mu$-1,2-hydroperoxo ligand is most likely protonated at O2.

**Further reactivity studies.** The straightforward fast protonation of [(susan$^{6\text{-}Me}$){Fe$^{III}$($\mu$-O)($\mu$-O$_2$)Fe$^{III}$}]$^{2+}$ with HClO$_4$ demonstrates the nucleophilic character of the $\mu$-1,2-peroxo ligand. On the other hand, the reaction of [(susan$^{6\text{-}Me}$){Fe$^{III}$($\mu$-O)($\mu$-O$_2$)Fe$^{III}$}]$^{2+}$ with 2-phenylpropanal as a typical substrate to evaluate the nucleophilic character of peroxo ligands[51,52] is slow in CH$_3$CN at $-5\,^{\circ}$C (Fig. S19). However, the formation of roughly one equivalent of acetophenone by performing this reaction on a preparative scale for 5 days supports the slow nucleophilic reactivity of the $\mu$-1,2-peroxo ligand.

The electrophilic character was not only evaluated for [(susan$^{6\text{-}Me}$){Fe$^{III}$($\mu$-O)($\mu$-O$_2$)Fe$^{III}$}]$^{2+}$ but also for the oxidized $\mu$-1,2-peroxo-Fe$^{IV}$Fe$^{III}$ and the protonated $\mu$-1,2-hydroperoxo-diferric species as both oxidation and protonation should increase the electrophilic character. This is already reflected in the different stabilities of these species (Fig. S10). As [(susan$^{6\text{-}Me}$){Fe$^{III}$($\mu$-O)($\mu$-1,2-O$_2$)Fe$^{III}$}]$^{2+}$ shows no indication of decay for hours in CH$_3$CN at $-40\,^{\circ}$C, oxidized [(susan$^{6\text{-}Me}$){Fe$^{IV}$($\mu$-O)($\mu$-O$_2$)Fe$^{III}$}]$^{3+}$ and protonated [(susan$^{6\text{-}Me}$){Fe$^{III}$($\mu$-O)($\mu$-1,2-OOH)Fe$^{III}$}]$^{3+}$ decay with half-lives of $\tau_{1/2} \approx 90$ min $\tau_{1/2} \approx 11$ min, respectively. Therefore, the clean characterization of the latter two species required lower temperatures of $-60\,^{\circ}$C and hence addition of a certain amount of CH$_2$Cl$_2$.

The electrophilic character of the three complexes were initially investigated using 9,10-dihydroanthracene (DHA) and PPh$_3$ as typical substrates for HAT and OAT, respectively. The reactions with [(susan$^{6\text{-}Me}$){Fe$^{III}$($\mu$-O)($\mu$-O$_2$)Fe$^{III}$}]$^{2+}$ were performed in CH$_3$CN at $-40\,^{\circ}$C, while that with [(susan$^{6\text{-}Me}$){Fe$^{IV}$($\mu$-O)($\mu$-O$_2$)Fe$^{III}$}]$^{3+}$ and [(susan$^{6\text{-}Me}$){Fe$^{III}$($\mu$-O)($\mu$-1,2-OOH)Fe$^{III}$}]$^{3+}$at $-60\,^{\circ}$C in CH$_3$CN/CH$_2$Cl$_2$ mixtures (vide supra). The parent [(susan$^{6\text{-}Me}$){Fe$^{III}$($\mu$-O)($\mu$-O$_2$)Fe$^{III}$}]$^{2+}$ showed no reactivity towards DHA and PPh$_3$ (Figs. S20, S21). The oxidized [(susan$^{6\text{-}Me}$){Fe$^{IV}$($\mu$-O)($\mu$-O$_2$)Fe$^{III}$}]$^{3+}$ also showed no reactivity towards DHA (Fig. S22), while the reaction with PPh$_3$ resulted in the reoccurence of the UV-Vis signature of the parent [(susan$^{6\text{-}Me}$){Fe$^{III}$($\mu$-O)($\mu$-O$_2$)Fe$^{III}$}]$^{2+}$ (Fig. S23). The reformation of the $\mu$-1,2-peroxo$\rightarrow$Fe$^{III}$ LMCT excludes an OAT reactivity between [(susan$^{6\text{-}Me}$){Fe$^{IV}$($\mu$-O)($\mu$-O$_2$)Fe$^{III}$}]$^{3+}$ and PPh$_3$ but suggests an oxidation of PPh$_3$[53] by [(susan$^{6\text{-}Me}$){Fe$^{IV}$($\mu$-O)($\mu$-O$_2$)Fe$^{III}$}]$^{3+}$. Analogous observations were made with the protonated [(susan$^{6\text{-}Me}$){Fe$^{III}$($\mu$-O)($\mu$-1,2-OOH)Fe$^{III}$}]$^{3+}$ that showed no reactivity with DHA (Fig. S24) and with PPh$_3$ the partial recovery of the UV-Vis signature of the parent [(susan$^{6\text{-}Me}$){Fe$^{III}$($\mu$-O)($\mu$-O$_2$)Fe$^{III}$}]$^{2+}$ (Fig. S25). Again, the reformation of the $\mu$-1,2-peroxo$\rightarrow$Fe$^{III}$ LMCT excludes an OAT reactivity between [(susan$^{6\text{-}Me}$){Fe$^{III}$($\mu$-O)($\mu$-1,2-OOH)Fe$^{III}$}]$^{3+}$ and PPh$_3$. The partial recovery of [(susan$^{6\text{-}Me}$){Fe$^{III}$($\mu$-O)($\mu$-O$_2$)Fe$^{III}$}]$^{2+}$ indicates a protonation equilibrium between PPh$_3$[54] and [(susan$^{6\text{-}Me}$){Fe$^{III}$($\mu$-O)($\mu$-1,2-OOH)Fe$^{III}$}]$^{3+}$.

As it is not surprising[35,55] that [(susan$^{6\text{-}Me}$){Fe$^{III}$($\mu$-O)($\mu$-O$_2$)Fe$^{III}$}]$^{2+}$ exhibits no electrophilic reactivity against DHA, the non-reactivity of both oxidized [(susan$^{6\text{-}Me}$){Fe$^{IV}$($\mu$-O)$_2$($\mu$-O$_2$)Fe$^{III}$}]$^{3+}$ and protonated [(susan$^{6\text{-}Me}$){Fe$^{III}$($\mu$-O)($\mu$-1,2-OOH)Fe$^{III}$}]$^{3+}$ is quite surprising. To further understand this non-reactivity, we determined the BDFE(OH)$_{CH_3CN}$ of [(susan$^{6\text{-}Me}$){Fe$^{III}$($\mu$-O)($\mu$-1,2-OOH)Fe$^{III}$}]$^{3+}$. In this respect, we determined the $pK_a$ of [(susan$^{6\text{-}Me}$){Fe$^{III}$($\mu$-O)($\mu$-1,2-OOH)Fe$^{III}$}]$^{3+}$ in CH$_3$CN that provided $9.5 \pm 0.1$ (Fig. S26). Using the typical square scheme (Fig. 7a) and the Bordwell relation Eq. (1)[56–58]

$$\text{BDFE}(O-H)_{CH_3CN} = 1.37 \, pK_a + 23.06 \, E^{0\prime} + 52.6 \, \text{kcal mol}^{-1}$$

(1)

provided BDFE(O–H)$_{CH3CN}$ = 78 $\pm$ 2 kcal mol$^{-1}$. This means that [(susan$^{6\text{-}Me}$){Fe$^{IV}$($\mu$-O)($\mu$-O$_2$)Fe$^{III}$}]$^{3+}$ should be capable as oxidant for HAT for substrates with a lower BDFE(X-H)$_{CH_3CN}$. The BDE(C–H) of DHA is 76.3 kcal mol$^{-1}$[59]. However, the intrinsic difference between BDFE and BDE[57], the temperature-dependence of BDFE especially for transition metal complexes, and the experimental error explain the non-reactivity of [(susan$^{6\text{-}Me}$){Fe$^{IV}$($\mu$-O)($\mu$-O$_2$)Fe$^{III}$}]$^{3+}$ with DHA. In this respect, TEMPOH should be a suitable HAT substrate with BDFE(O–H)$_{CH_3CN}$ = 66.5 kcal mol$^{-1}$ and BDE(O–H) = 70.6 kcal mol$^{-1}$[57] especially for [(susan$^{6\text{-}Me}$){Fe$^{IV}$($\mu$-O)($\mu$-O$_2$)Fe$^{III}$}]$^{3+}$.

The parent [(susan$^{6\text{-}Me}$){Fe$^{III}$($\mu$-O)($\mu$-O$_2$)Fe$^{III}$}]$^{2+}$ showed no reactivity with TEMPOH (Fig. S27), which is in-line that this one-electron reduced species [(susan$^{6\text{-}Me}$){Fe$^{III}$($\mu$-O)($\mu$-O$_2$)Fe$^{III}$}]$^{2+}$ should have a driving force for HAT significantly lower than 78 kcal mol$^{-1}$ of [(susan$^{6\text{-}Me}$){Fe$^{IV}$($\mu$-O)$_2$($\mu$-O$_2$)Fe$^{III}$}]$^{3+}$. The reaction of protonated [(susan$^{6\text{-}Me}$){Fe$^{III}$($\mu$-O)($\mu$-1,2-OOH)Fe$^{III}$}]$^{3+}$ with TEMPOH resulted in the reoccurence of the UV-Vis signature of the parent [(susan$^{6\text{-}Me}$){Fe$^{III}$($\mu$-O)($\mu$-O$_2$)Fe$^{III}$}]$^{2+}$ (Fig. S28). The reformation of the parent $\mu$-1,2-peroxo-diferric complex excludes a HAT reactivity between [(susan$^{6\text{-}Me}$){Fe$^{III}$($\mu$-O)($\mu$-1,2-OOH)Fe$^{III}$}]$^{3+}$ and TEMPOH and suggests a protonation of TEMPOH[60]. The reaction of [(susan$^{6\text{-}Me}$){Fe$^{IV}$($\mu$-O)($\mu$-O$_2$)Fe$^{III}$}]$^{3+}$ with TEMPOH resulted in the reoccurence of the UV-Vis signature of the parent [(susan$^{6\text{-}Me}$){Fe$^{III}$($\mu$-O)($\mu$-O$_2$)Fe$^{III}$}]$^{2+}$ (Fig. S29). This is in-line with HAT from TEMPOH to [(susan$^{6\text{-}Me}$){Fe$^{IV}$($\mu$-O)($\mu$-O$_2$)Fe$^{III}$}]$^{3+}$ resulting in [(susan$^{6\text{-}Me}$){Fe$^{III}$($\mu$-O)($\mu$-1,2-OOH)Fe$^{III}$}]$^{3+}$ that reacts again by protonation of excess TEMPOH to [(susan$^{6\text{-}Me}$){Fe$^{III}$($\mu$-O)($\mu$-O$_2$)Fe$^{III}$}]$^{2+}$.

Thus, only the oxidized [(susan$^{6\text{-}Me}$){Fe$^{IV}$($\mu$-O)($\mu$-O$_2$)Fe$^{III}$}]$^{3+}$ is capable for HAT from TEMPOH corroborated by the BDFE while the parent [(susan$^{6\text{-}Me}$){Fe$^{III}$($\mu$-O)($\mu$-O$_2$)Fe$^{III}$}]$^{2+}$ and the protonated [(susan$^{6\text{-}Me}$){Fe$^{III}$($\mu$-O)($\mu$-1,2-OOH)Fe$^{III}$}]$^{3+}$ does not exhibit enough electrophilic character. However, [(susan$^{6\text{-}Me}$){Fe$^{III}$($\mu$-O)($\mu$-O$_2$)Fe$^{III}$}]$^{2+}$ that shows no decay at $-40\,^{\circ}$C exhibits a change in the UV-Vis spectra at room temperature with the formation of the typical signature of complexes with a {Fe$^{III}$X($\mu$-O)Fe$^{III}$X} core[38] without the observation of intermediates

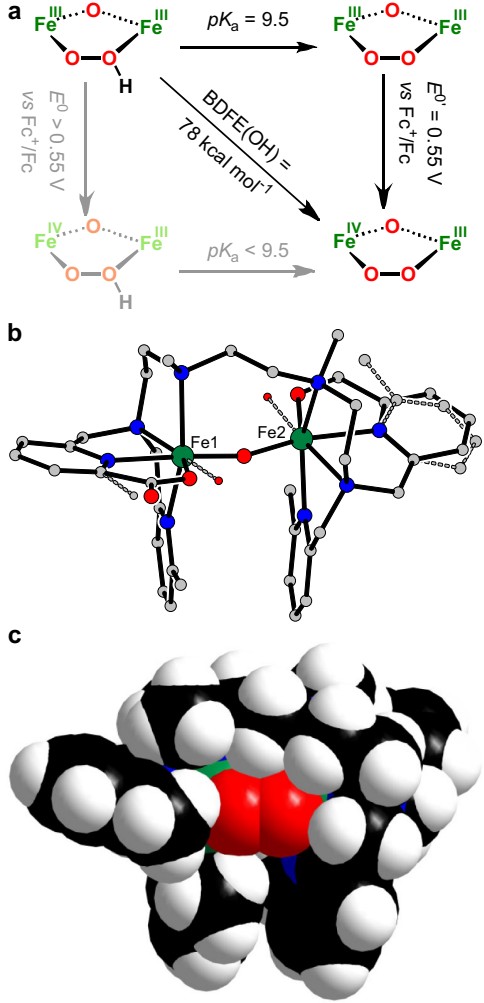

**Fig. 7 Reactivity studies. a** Square scheme showing the PCET thermochemistry of [(susan$^{6-Me}$){Fe$^{III}$($\mu$-O)($\mu$-1,2-OOH)Fe$^{III}$}]$^{3+}$ in CH$_3$CN. **b** Molecular structure of the decay product of [(susan$^{6-Me}$){Fe$^{III}$($\mu$-O)($\mu$-1,2-O$_2$)Fe$^{III}$}]$^{2+}$ including the ligand disorders (please see also Supplementary Fig. 1c): at Fe1 80% carboxylate and 20% hydroxide (shown with dotted lines), at Fe2 35% benzylalcoholato and 65% hydroxide (shown with dotted lines). **c** Space-filling model of [(susan$^{6-Me}$){Fe$^{III}$($\mu$-O)($\mu$-1,2-O$_2$)Fe$^{III}$}]$^{2+}$ to illustrate the encapsulation of the peroxo ligand by a CH$_2$ group (right Fe) and a 6-methyl group (left Fe).

accompanied by deposition of an inhomogenous solid and a few single-crystals. The crystallographic analysis provided the structure of the decay product (Fig. 7b) based on [(susan$^{6-Me}$){Fe$^{III}$(OH)($\mu$-O)Fe$^{III}$(OH)}]$^{2+}$[61] with a disorder of the coordinated OH$^-$ ligands, which could be resolved to coordination of oxidized 6-methyl groups. At Fe1, only 20% is OH$^-$ while 80% is a carboxylate while at Fe2, 65% is OH$^-$ and 35% consists of a benzylalcoholato donor. NMR spectroscopy of the demetalated bulk decay product shows the formation of more than one product but a significant signal at 173.4 ppm for a benzoic acid group in the $^{13}$C NMR spectrum indicates that the hydroxylation of the 6-methyl group is not only a minor reaction path.

## Discussion

To mimic dinuclear active sites of metalloenzymes, we have developed a dinucleating ligand system with varying terminal donors[38,39]. With the ligand susan, we obtained in straightforward reactions a series of $\mu$-oxo-diferric complexes {Fe$^{III}$X($\mu$-O) Fe$^{III}$X} bearing anionic exogenous ligands X$^-$[37,39,45,61]. The complex with hydroxides, [(susan){Fe$^{III}$(OH)($\mu$-O)Fe$^{III}$(OH)}]$^{2+}$, catalyzes the oxidation of CH$_3$OH with H$_2$O$_2$ to HCHO[62]. We could observe the $\mu$-1,2-peroxo intermediate [(susan){Fe$^{III}$($\mu$-O) ($\mu$-1,2-O$_2$)Fe$^{III}$}]$^{2+}$. However, the temperature-dependencies ruled out this $\mu$-1,2-peroxo intermediate to be the active species indicating the conversion to a high-valent active species. This conversion is faster in the presence of a proton suggesting a transient $\mu$-1,2-hydroperoxo species.

In contrast, formation of $\mu$-oxo-diferric complexes with susan$^{6-Me}$ was not possible under identical aerobic conditions. Only the small ligand F$^-$ with H$_2$O$_2$ as oxidant allowed the isolation of [(susan$^{6-Me}$){Fe$^{III}$F($\mu$-O)Fe$^{III}$F}]$^{2+}$[40]. Its comparison to [(susan){Fe$^{III}$F($\mu$-O)Fe$^{III}$F}]$^{2+}$ showed a steric repulsion between the 6-methyl group and the terminal F$^-$ ligand in cis-position, which explains the inaccessibility of susan$^{6-Me}$ complexes with larger terminal ligands as Cl$^-$ or OAc$^-$ that are easily accessible with susan. Thus, the $\mu$-oxo-brigded core {Fe$^{III}$X($\mu$-O)Fe$^{III}$X} is not the thermodynamic sink for susan$^{6-Me}$ as it is for susan. Moreover, the steric repulsion of the 6-methyl group enforces longer Fe–N$^{6-Me-py}$ than Fe–N$^{py}$ bonds[63–68], and hence a lower electron donation. This results in an anodic shift of +250 mV making Fe$^{IV}$ less accessible with susan$^{6-Me}$ than with susan.

In this respect, we thought that susan$^{6-Me}$ should be able to stabilize a $\mu$-1,2-peroxo complex {Fe$^{III}$($\mu$-O)($\mu$-1,2-O$_2$)Fe$^{III}$} that is with susan only a reactive intermediate decaying via a high-valent Fe$^{IV}$ species to its thermodynamic sink {Fe$^{III}$X($\mu$-O) Fe$^{III}$X}. In contrast, with susan$^{6-Me}$ not only this thermodynamic driving force is absent but Fe$^{IV}$ is also less accessible. Indeed, we could present here the synthesis and characterization of the stable $\mu$-1,2-peroxo complex [(susan$^{6-Me}$){Fe$^{III}$($\mu$-O)($\mu$-1,2-O$_2$)Fe$^{III}$}]$^{2+}$. Inspection of the space-filling model (Fig. 7c) shows that the $\mu$-1,2-peroxo ligand is even further stabilized by a better encapsulation with the 6-methyl group of susan$^{6-Me}$ (left peroxo-oxygen atom in Fig. 7c) that would be absent with susan. This ligand encapsulation also explains the slower nucleophilic reactivity of the $\mu$-1,2-peroxo ligand for the organic substrate 2-phenylpropanal than for the small H$^+$.

Although susan$^{6-Me}$ is less suited for stabilization of Fe$^{IV}$ than susan, the principal accessibility of Fe$^{IV}$ with susan$^{6-Me}$ is demonstrated by the reversible oxidation to [(susan$^{6-Me}$){Fe$^{IV}$($\mu$-O)($\mu$-1,2-O$_2$)Fe$^{III}$}]$^{3+}$, which is stabilized by the additional highly covalent $\mu$-1,2-peroxo ligand. It is interesting to note, that this high-valent $\mu$-1,2-peroxo species stores one oxidation-equivalent more than intermediate **Q** of sMMO. Comparing the Fe$^{IV}$Fe$^{III}$/Fe$^{III}$Fe$^{III}$ redox potential of $E_{1/2} = 0.55$ V to 0.41 V vs Fc$^+$/Fc for the analogous redox couple of [(tpa$^{6-Me}$){Fe$^{III}$($\mu$-O)$_2$Fe$^{III}$} (tpa$^{6-Me}$)]$^{2+}$[69] shows only a slightly lower electron-donating character of the $\mu$-1,2-peroxo-bridge than a $\mu$-oxo-bridge.

We could further demonstrate the reversible protonation to the $\mu$-1,2-hydroperoxo-diferric complex [(susan$^{6-Me}$){Fe$^{III}$($\mu$-O)($\mu$-1,2-OOH)Fe$^{III}$}]$^{3+}$. Generally, protonation of a Fe-coordinated peroxo ligand is regarded to enhance its reactivity, e. g. protonation of the cis-$\mu$-1,2-peroxo intermediate **P** of sMMO was proposed to promote the conversion to intermediate **Q**[11]. The relatively high stability of the $\mu$-1,2-hydroperoxo complex [(susan$^{6-Me}$){Fe$^{III}$($\mu$-O)($\mu$-1,2-OOH)Fe$^{III}$}]$^{3+}$ (no decay at −60 °C, $\tau_{1/2} \approx 11$ min at −40 °C) is thus remarkable and must owe its origin to a low stabilization of the Fe$^{IV}$ conversion product by susan$^{6-Me}$.

In contrast to [(susan$^{6-Me}$){Fe$^{III}$($\mu$-O)($\mu$-1,2-O$_2$)Fe$^{III}$}]$^{2+}$, the $\mu$-1,2-peroxo complex **A** is protonated at the $\mu$-oxo-bridge forming a {Fe$^{III}$($\mu$-1,2-O$_2$)($\mu$-OH)Fe$^{III}$} species[22] indicating different nucleophilicities. The nucleophilic character of a ligand should increase with less electron donation to the Fe$^{III}$ ions, i.e. less covalent, longer bonds. But for **A**, the Fe$^{III}$-$\mu$-O$^{oxo}$ bonds are shorter than for [(susan$^{6-Me}$){Fe$^{III}$($\mu$-O)($\mu$-1,2-O$_2$)Fe$^{III}$}]$^{2+}$

(1.72/1.74 Å vs 1.82/1.89 Å), whereas the situation is reversed for the $Fe^{III}$-$\mu$-O$^{peroxo}$ bonds (2.07/2.10 Å vs 1.88/1.93 Å). This structural argumentation is in contrast to the experimentally determined protonation sites. However, the disorder of the $\mu$-oxo/ $\mu$-1,2-peroxo ligands in **A** questions the significance of this comparison. Moreover, as protonation should occur at a p$^\pi$ orbital and a Fe–O bond consists of $\sigma$- and $\pi$-bonding, a pure structural analysis does not need to provide the answer for the reactivity.

Thus, spectroscopic markers might provide a better correlation to the nucleophilic character of the peroxo group than structural parameters. Solomon and coworkers proposed to extract the donor strength of a given ligand from the integrated absorption intensities of all CT transitions associated with this ligand[70]. Here, the $\pi$-charge donation from the peroxo $\pi^*_\pi$ donor orbital into the Fe 3d$_\pi$ acceptor orbitals should be extractable from the prominent $\mu$-1,2-peroxo → Fe LMCTs. Complex **A** exhibits this $\mu$-1,2-peroxo→$Fe^{III}$ LMCT with $\varepsilon = 1500$ M$^{-1}$ cm$^{-1}$ and a much less intense $\mu$-oxo→$Fe^{III}$ LMCT. In contrast, the $\mu$-oxo → $Fe^{III}$ LMCT in [(susan$^{6-Me}$){$Fe^{III}$($\mu$-O)($\mu$-1,2-O$_2$)$Fe^{III}$}]$^{2+}$ is more intense ($\varepsilon = 1180$ M$^{-1}$ cm$^{-1}$) than the $\mu$-peroxo → $Fe^{III}$ LMCT ($\varepsilon = 1000$ M$^{-1}$ cm$^{-1}$). Note that also the integrated absorption intensity is smaller in the susan$^{6-Me}$ complex than in **A** for the $\mu$-1,2-peroxo→$Fe^{III}$ LMCT indicating - without the intention of a quantitative analysis - less charge-donation and hence more nucleophilic character of the $\mu$-1,2-peroxo ligand. This UV-Vis spectroscopic argumentation is supported by a comparison of the vibrational signature in the rR spectra. Interestingly, the strongest difference is observed for the $\nu_s$(Fe–O$_2$–Fe), which are at 465 and 448 cm$^{-1}$ for **A** and [(susan$^{6-Me}$){$Fe^{III}$($\mu$-O)($\mu$-1,2-O$_2$)$Fe^{III}$}]$^{2+}$, respectively, indicative for less covalent $Fe^{III}$-$\mu$-O$^{peroxo}$ bonds and hence a higher nucleophilicity of the $\mu$-1,2-peroxo ligand in [(susan$^{6-Me}$){$Fe^{III}$($\mu$-O)($\mu$-1,2-O$_2$)$Fe^{III}$}]$^{2+}$.

The study of the electrophilic reactivity demonstrated only a low electrophilic character of the parent $\mu$-1,2-peroxo-$Fe^{III}Fe^{III}$ complex that is not unexpected for such complexes[35,55]. Interestingly, also protonation to the $\mu$-1,2-hydroperoxo-$Fe^{III}Fe^{III}$ species turned out to be not sufficient to increase the electrophilic character for HAT with substrates of weak to modest BDE (TEMPOH and DHA). Only the oxidized $\mu$-1,2-peroxo-$Fe^{IV}Fe^{III}$ species reacts with the relatively weak substrate TEMPOH. The determination of the $pK_a = 9.5 \pm 0.1$ and the bond dissociation free energy BDFE(O–H)$_{CH_3CN} = 78 \pm 2$ kcal mol$^{-1}$ of the protonated $\mu$-1,2-hydroperoxo-$Fe^{III}Fe^{III}$ species quantifies this low electrophilic character even of the oxidized $\mu$-1,2-peroxo-$Fe^{IV}$-$Fe^{III}$ species. In this respect, the intramolecular C–H activation of preorganized 6-methyl pyridine groups to benzylalcoholato and carboxylato donors by the parent $\mu$-1,2-peroxo-$Fe^{III}Fe^{III}$ complex is remarkable. Considering that this complex does not react with TEMPOH (BDE(O–H) = 70.6 kcal mol$^{-1}$) and using the BDE(C–H) = 90 kcal mol$^{-1}$[59] of the methyl group of toluene as an approximation for the BDE(C–H) of the 6-methyl groups of the coordinated pyridines, HAT should not occur via the bridging $\mu$-1,2-peroxo-ligand. This indicates that this intramolecular reaction requires the conversion of the $\mu$-1,2-peroxo-diferric core to a more reactive high-valent species as already postulated for the CH$_3$OH oxidation[62] of the analogous $\mu$-1,2-peroxo-diferric complex of susan (*vide supra*).

[(susan$^{6-Me}$){$Fe^{III}$($\mu$-O)($\mu$-1,2-OOH)$Fe^{III}$}]$^{3+}$ is an $\mu$-1,2-hydroperoxo model complex and provides spectroscopic signatures for the assignment of postulated hydroperoxo intermediates in diiron enzymes: Upon protonation, the prominent $\mu$-1,2-peroxo→$Fe^{III}$ LMCT around 14,000–16,000 cm$^{-1}$ disappears and the isomer shift decreases. The presence of an $\mu$-oxo-bridge is indicated by the typical $\mu$-oxo→$Fe^{III}$ LMCTs around 12,000 and 19,000 cm$^{-1}$ and large values of $|\Delta E_Q| \geq 1.3$ mm s$^{-1}$. The best signature to

differentiate between a $\mu$-1,1-hydroperoxo and a $\mu$-1,2-hydroperoxo is the appearance of two strongly differing quadrupole doublets for the latter due to the strongly differing donation abilities of the two $\mu$-1,2-hydroperoxo-oxygen atoms.

The protonation of [(susan$^{6-Me}$){$Fe^{III}$($\mu$-O)($\mu$-1,2-O$_2$)$Fe^{III}$}]$^{2+}$ to [(susan$^{6-Me}$){$Fe^{III}$($\mu$-O)($\mu$-1,2-OOH)$Fe^{III}$}]$^{3+}$ reflect the UV-Vis-NIR and Mössbauer spectroscopic differences between **P** and the more reactive **P′** intermediates in non-heme diiron enzymes[2,15,16,20]. For the latter, a $\mu$-1,2-hydroperoxo structure has been suggested, which is thus strongly supported by the results of this study.

The diferrous form of AurF exhibits one quadrupole doublet demonstrating structurally rather similar iron places[12]. Reaction with O$_2$ provides a diferric species that lacks the typical $\mu$-1,2-peroxo → $Fe^{III}$ LMCT around 14,000 cm$^{-1}$. The Mössbauer spectrum contains two quite different quadrupole doublets with $\delta_1 = 0.54$, $\Delta E_{Q1} = 0.66$ mm s$^{-1}$ and $\delta_2 = 0.61$, $\Delta E_{Q2} = 0.35$ mm s$^{-1}$[12,13]. This coupled with the low values of $|\Delta E_Q|$ and a lack of the $\mu$-1,2-peroxo → $Fe^{III}$ LMCT leads us to suggest the formulation of the **P′**-type intermediate in AurF as a {$Fe^{III}$($\mu$-1,2-hydroperoxo)$Fe^{III}$} core without a $\mu$-oxo-bridge, which also supports a recent proposal[15].

## Methods

**Synthesis of [(susan$^{6-Me}$){$Fe^{II}$($\mu$-OH)$_2Fe^{II}$}](ClO$_4$)$_2$·H$_2$O.** A solution of susan$^{6-Me}$ (595 mg, 1.00 mmol) in MeOH (25 mL) was added to a solution of Fe(ClO$_4$)$_2$·6H$_2$O (726 mg, 2.00 mmol, 2.0 equiv) in MeOH (20 mL). This yellow solution was stirred at room temperature for 10 min followed by an addition of a 25% aqueous solution of ammonia (0.16 mL, 2.1 mmol, 2.1 equiv) resulting in a slight intensity increase of the yellow color. Yellow crystals of [(susan$^{6-Me}$){$Fe^{II}$($\mu$-OH)$_2Fe^{II}$}](ClO$_4$)$_2$·MeOH precipitated at 0 °C, which were filtered off, washed three times with water, a small amount of cold MeOH, three times with Et$_2$O, and dried under reduced pressure. Yield: 704 mg (7.50 × 10$^{-4}$ mol, 75%). IR (KBr): $\tilde{\nu}$/cm$^{-1}$ = 3642 w, 3083 w, 2976 w, 2856 m, 1603 s, 1576 m, 1462 s, 1429 m, 1093 vs, 1009 m, 956 m, 847 m, 780 s, 623 s, 490 m. ESI-MS (+) (CH$_2$Cl$_2$): m/z = 361.2 [(susan$^{6-Me}$){Fe($\mu$-O)Fe}]$^{2+}$. Anal. Found: C 44.90, H 5.76, N 11.51. Calcd. for [(susan$^{6-Me}$){Fe($\mu$-OH)$_2$Fe}](ClO$_4$)$_2$·H$_2$O C$_{36}$H$_{54}$N$_8$Cl$_2$Fe$_2$O$_{11}$: C 45.16, H 5.68, N 11.70.

**Synthesis of [(susan$^{6-Me}$){$Fe^{III}$($\mu$-O)($\mu$-O$_2$)$Fe^{III}$}](ClO$_4$)$_2$·2H$_2$O.** A solution of [(susan$^{6-Me}$){$Fe^{II}$($\mu$-OH)$_2Fe^{II}$}](ClO$_4$)$_2$ (63 mg, 6.7 × 10$^{-5}$ mol) in CH$_2$Cl$_2$ (2 mL) was added to a solution of 25% aqueous ammonia (20 µL, 2.6 × 10$^{-4}$ mol, 3.9 equiv) in MeOH (6 mL) at −15 °C. Pouring an O$_2$ stream of approximate 0.1 L min$^{-1}$ through this yellow solution for 1.5 h at −15 °C results in immediately color change to dark green followed by precipitation of a black microcrystalline solid, which was filtered off, washed three times with cold water, a small amount of cold MeOH, three times with Et$_2$O, and dried under reduced pressure. Yield: 52 mg (5.3 × 10$^{-5}$ mol, 78%). Anal. Found: C 43.34, H 5.20, N 11.02. Calcd. for [(susan$^{6-Me}$){$Fe^{III}$($\mu$-O)($\mu$-O$_2$)$Fe^{III}$}](ClO$_4$)$_2$·2H$_2$O C$_{36}$H$_{54}$N$_8$Cl$_2$Fe$_2$O$_{13}$: C 43.30, H 5.55, N 11.22. For single-crystal X-ray diffraction and magnetic measurements, the sample was recrystallized by slow evaporation of a filtered (0.2 µm pores size PTFE filter) solution of [(susan$^{6-Me}$){$Fe^{III}$($\mu$-O)($\mu$-O$_2$)$Fe^{III}$}](ClO$_4$)$_2$ (200 mg, 2.02 × 10$^{-4}$ mol) in MeCN (20 mL) at −30 °C. The resulting black crystals of [(susan$^{6-Me}$){$Fe^{III}$($\mu$-O)($\mu$-O$_2$)$Fe^{III}$}](ClO$_4$)$_2$·0.85 MeCN·0.7 H$_2$O were filtered off, washed three times with cold water, three times with cold MeOH, three times with Et$_2$O, and dried under reduced pressure. Yield: 52 mg (5.2 × 10$^{-5}$ mol, 26%). IR (KBr): $\tilde{\nu}$/cm$^{-1}$ = 3079 w, 2999 w, 2918 w, 2881 w, 2811w, 1605s, 1573 w, 1466 s, 1454 s, 1090 vs, 1002 m, 955 m, 933 w, 842 m, 832 w, 788 s, 686 m, 624 s, 500 w, 447 w. ESI-MS (+) (CH$_2$Cl$_2$/MeCN): m/z = 377.1 [(susan$^{6-Me}$){Fe($\mu$-O)($\mu$-O$_2$)Fe}]$^{2+}$. Anal. Found: C 44.66, H 5.21, N 12.27. Calcd. for [(susan$^{6-Me}$){Fe($\mu$-O)($\mu$-O$_2$)Fe}](ClO$_4$)$_2$·0.8MeCN·H$_2$O C$_{37.6}$H$_{54.4}$N$_{8.8}$Cl$_2$Fe$_2$O$_{12}$: C 44.97, H 5.46, N 12.27.

## Data availability

The crystallographic data generated in this study have been deposited at the Cambridge Crystallographic Data Centre under accession codes 2072804–2072806 (www.ccdc.cam.ac.uk/data_request/cif). Experimental details on synthesis and crystal structure determination, details on DFT calculations, analysis of Mössbauer spectra, ESI-MS, thermal ellipsoid plots, Mössbauer spectra, UV-Vis spectra, rR spectra, X-ray crystallographic data (cif) generated in this study are provided in the Supplementary Information. Source data are available from the corresponding author upon request.

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

## Acknowledgements
T.G. acknowledges Bielefeld University of financial support.

## Author contributions
S.W. conducted all syntheses, oxidation experiments, protonation experiments, reactivity studies, characterizations not otherwise mentioned here, and the DFT calculations. T.Z. conducted the initial syntheses of the peroxo-diferric complex. S.W., H.H., C.P. and S.K. recorded and analyzed the resonance Raman data. T.H. and P.H. assisted with the analysis of the resonance Raman data. A.S. and H.B. collected, solved, and refined all the crystallographic data. E.B. recorded and interpreted the EPR and magnetic Mössbauer data. T.G. designed experiments, assisted with data analysis, and wrote the manuscript with input from all the authors.

## Funding

## Competing interests
The authors declare no competing interests.
