## [Peer Review File · Nature Communications]

Generation of a μ -1,2-Hydroperoxo $\text{Fe}^{\text{III}}\text{Fe}^{\text{III}}$ and a μ -1,2-Peroxo $\text{Fe}^{\text{IV}}\text{Fe}^{\text{III}}$ ComplexREVIEWER COMMENTS

Reviewer #1 (Remarks to the Author):

This manuscript describes a diferric peroxide complex which stability permits to study reversible protonation and single electron oxidation. Reactions provide unprecedented examples of diferric hydroperoxide and Fe^{III}Fe^{IV}-peroxide compounds. These molecules are of relevance to numerous diiron dependent enzymes and may be valuable tools to understand their chemistry and spectroscopic properties. Overall, this is an interesting and relevant work to the bioinorganic and inorganic chemistry communities. On the other hand, while the spectroscopy of these novel and interesting molecules is analysed in some detail, the reactivity falls behind and I think this should be addressed before publication. An obvious point of interest raised by the current work is the oxidation ability of the species resulting from protonation and also from single electron oxidation. From simple charge considerations one can infer that these may be more reactive (oxidizing) against external substrates. It is surprising that the authors do not dedicate some attention to this specific question, which seem straightforward to address given their ability to generate the hydroperoxide and high valent species in clean spectroscopic form.

The reversible protonation is studied with a strong acid and NEt₃. It will be quite interesting to titrate the complex with a weaker acid, in order to obtain an equilibrium constant that could be translated into a pK_a type of value. Combination of the pK_a value and the red-ox potential for the single e⁻ oxidation can provide a value of the BDE of the O-H bond in the Fe(OOH)Fe species. I believe this will be a valuable information for the field, since it can help place the oxidation ability of these species in the context of other metal-oxo/peroxo species of relevance to oxygenases.

Related to the previous question, the ability to protonate and oxidize the diferric peroxide strongly suggests that the Fe^{IV}Fe^{III} species should be a competent HAT agent, leading to the ferric hydroperoxide complex. This can be tested by reaction against some substrates with known X-H bond BDE.

On the other hand, it will be quite interesting to know the oxygen atom transfer ability of these novel molecules is investigated.

The authors describe the formation of small amounts of crystals where the ligand is hydroxylated.

This is quite an interesting reaction but it is not possible to deduce from this analysis if this is a minor path or if it is a major reaction path. To address this question authors can demetallate the final compound and analyse the organics to know the percentage of ligand oxidized.

The reaction with propanal requires authors to identify the final product. The decay observed upon addition of the aldehyde does not necessarily imply the suggested nucleophilic reaction.

The MB analysis convincingly shows that the iron centers remain high spin in the peroxide, hydroperoxide and high valent species. I recommend authors to make an explicit mention to this aspect. Notice that the EPR of the mixed valence compound shows signals corresponding to a S = 1/2 system that could be also interpreted as a low spin ferric center.

I wonder if the spectroscopic data excludes the possibility that protonation leads to a terminally bound hydroperoxide ligand ligated to only one of the two centers. Some discussion will be appreciated.

Minor questions;

Page two, authors refer to "higher isomer shift". For chemists not familiar with iron chemistry it may not be obvious that the authors refer to Mossbauer spectroscopy. I recommend explicit mention.

Page 7. Authors argue on a change from 98 to 95% implying some chemical reduction during the coulometric experiments. This seems a value within experimental error.

Page 10. I don't understand how the authors assign the weak 11800 cm⁻¹ feature to a CT band. Please explain.

Page 10. It is a bit confusing that the UV-vis spectra use cm⁻¹ as units and then the discussion in the text uses nm to describe the wavelength of Raman excitations. I recommend to plot the UV-vis data in nm.

Page 14. I am not sure the intensity extracted from extinction coefficients can be directly used to compare the donor strength, but instead the area of the bands should be used. Please reconsider.

Page 14. I can not understand how the authors translate the Fe-O₂-Fe vibration energy into nucleophilicity. Please explain.

Reviewer #2 (Remarks to the Author):

The authors report the reactivity of a dinuclear Fe(II) complex using a ligand similar to that described over 30 years ago by Armstrong and Toftlund (might be an idea to cite these studies!), which have been studied for catalytic properties in oxidation catalysis over the years. The presence of a methyl group at the 6 position of the pyridyl has been explored before mainly by the Que group and the oxidative degradation of similar ligands by several groups.

Typically these complexes are isolated already in the Fe(III) oxidation state with oxido/carboxylato bridges and so their study in the Fe(II) state is novel and especially their reactivity with molecular oxygen is of interest for bioinorganic chemistry. Hence the work is novel and relevant.

The characterization, in particular by vibrational spectroscopy with labelling is indeed of a good standard but nowadays routine. Hence the fact that the manuscript is filled with superlatives (thoroughly, unprecedented...) which make the reading of it somewhat irksome. There is no need to oversell relatively routine/expected characterization as if it were well beyond the norm.

The discussion of resonance enhancement is somewhat presumptive. The concentrations used (20 mM) should be on the edge of allowing for Raman spectra to be obtained without enhancement and hence the enhancement is probably only a factor of 10 or so at most. The absence of enhancement for the other complexes is expected since the transitions are metal centered. Excitation in the UV may be more enlightening, but that of course brings the possibility of photochemistry.

all in all a well worked study well described.

minor comments

in figure 5 'eq' is written 'Aq'

trans-stilben should be trans-stilbene

Reviewer #3 (Remarks to the Author):

Review of the manuscript referenced NCOMMS-21-15904-T

The manuscript referenced NCOMMS-21-15904-T by Glaser and coworkers describes the preparation of a series of diiron (μ -oxo)(μ -peroxo/hydroperoxo) complexes of interest as potential models for diiron oxygenases and oxidation catalysts.

The characterization of the compounds is realized by combined spectroscopic analyses and is established beyond doubt. Overall the experimental work is very competently done and the conclusions appear sound. I find the DFT support confusing and unconvincing.

Hereafter are a few comments and questions on the experimental work.

1 - page 3: the second to last sentence concerning bond distances is not clear at all and should be rewritten.

2 - pages 4 and 5: a direct link between covalency and antiferromagnetic coupling is stressed twice which is not fully obvious to me. This deserves to be explained.

3 - page 7, last sentence: an explanation (or at least an hypothesis) why no rR spectrum can be recorded for the oxidized compound should be provided. Is it due to photoreduction or lack of enhancement or anything else?

4 - page 8, last paragraph on Mössbauer: this presentation is unclear if not confusing, whereas the

explanation in the SI is quite instructive. The expression "not-localized" is ambiguous: it suggests "delocalized" which is not exact in this case. I suggest that the authors rewrite this paragraph with the help of the SI to clarify the situation.

5 - page 9: in the expression "a job plot analysis" job must be written with a capital J. As a matter of fact, Paul Job was an analytical chemist of the XXth century who introduced this plot, and there is no relationship with the labor market.

6 - page 10, Figure 5: it is stated that the peroxo \rightarrow Fe LMCT "disappears". Figure 5 shows that the oxo \rightarrow Fe LMCT @ 19000 cm^{-1} appears "doubled". Accordingly, is it possible that the peroxo \rightarrow Fe LMCT does not disappear, but moves to 18000 cm^{-1} ? The authors should discuss and possibly eliminate this possibility before concluding.

Comments on the DFT calculations

7 - page 8: For the oxidized complex, two configurations are envisaged and one is preferred on energetic grounds, the difference being 820 cm^{-1} . The two configurations cannot be distinguished from the calculated Mössbauer parameters. It is stated that the calculated Mössbauer parameters of both configurations match the experimental ones.

page 11: For the protonated complex, in spite of the fact that protonation at O1 is favored by DFT calculations by 1200 cm^{-1} , the authors prefer protonation at O2 because the calculated Mössbauer parameters match better the experimental ones.

8 - Concerning the oxidized compound, it is probably a class II mixed-valent species (stated in SI) or in other words it exhibits temperature-dependent partial residency of the extra electron on the two sites. How was this feature taken into account in the calculations? Could it be taken into account in the framework of DFT calculations or did it require CASSCF calculations? The ultimate question may be: is the computational approach adapted to the problem?

9 - General comment on the DFT calculations. Their presentation is not sufficiently detailed to warrant the conclusions. These open-shell systems are intrinsically tough to calculate and one must be very cautious and provide valuable information on the way the calculations were conducted and eventually controlled to convince the reader. In particular, the authors must indicate how they proceeded to carry out broken-symmetry calculations. In addition, for the geometry optimizations they must indicate how they imposed the localization of the electron hole and provide the spin density distribution of the optimized structure. Same analysis for the protonation site. Finally, it is usual practice to communicate in the SI the cartesian coordinates of the optimized geometries.

To summarize, the results presented are new and significant and deserve publication. However, their presentation must be improved, in particular by moving key arguments from the SI to the text. In addition, the computational procedures must be detailed to prove the validity of the calculations. I recommend publication once these points have been implemented.

Reviewer #4 (Remarks to the Author):

The paper deals with a highly relevant topic, namely the enzymatic activation of dioxygen, and identifies experimentally various possible intermediates along the catalytic process. The main results are also backed up or further studied with theoretical calculations. Overall, the paper is written in very precise language, which is, on the one hand, a positive aspect, but also leads to the paper being very technical. It is extremely hard, if not impossible, to really "read" the paper. For the results part, this might be justified, but for the discussion part, the language should probably be less technical and more suited for a general readership.

I have only a few more detailed comments and suggestions:

1) Introduction: There are a few intermediate structures highlighted explicitly in the introduction with labels P, P' and Q. It would be very helpful to visualize them so that at the outset of the paper it is clear on the first glance what the overall fundament of the paper is. A sentence such as "the peroxo intermediate P converts to a high-valent active species Q" is not very helpful as the general reader has no idea what is hidden behind the labels P and Q. [As discussed above, the technical focus of the paper makes it very difficult to digest for a general reader.]

2) On page 3, in the introduction, there are statements referring to past work without citations. "allowed an unprecedented ... reactivity study" and "was observed by X-ray crystallography" need a citation each to these individual studies.

3) Page 3, last few sentences. The structural parameters discussed have been obtained from single-crystal X-ray diffraction? This should be clarified.

4) Page 5: "We synthesized all four possible isotopomers as solids." Crystalline powder? Amorphous?

5) According to the introduction and the title, one of the most exciting findings of this study is the protonated peroxo form, discussed on pages 9-11. But very little is said about the chemistry. How was it synthesized? "Treatment of xx with yy" is not very helpful. Was it in solution? At room temperature? Open to air? Is the final compound a solid, or does it exist only in solution? Or only observed in situ? How was it isolated and purified? Only in the discussion part, the authors mention "the high stability" of the compound. What does this mean, and relative to what?

6) The crystal structure determination and refinement have been done accurately and expertly. However, I think it is very important to show all three structures with atomic displacement ellipsoids in Figure S4 in the SI. Especially the most complicated one is left out. If the model with all disorder parts is too difficult to visualize clearly, maybe only the major disorder components could be shown. But the sketch in Figure 6 as a ball-and-stick model cannot replace a representation of a model in the SI that includes the (possible problems with the) ADPs of the atoms.

Response to Reviewer #1

This manuscript describes a diferric peroxide complex which stability permits to study reversible protonation and single electron oxidation. Reactions provide unprecedented examples of diferric hydroperoxide and Fe^{III}Fe^{IV}-peroxide compounds. These molecules are of relevance to numerous diiron dependent enzymes and may be valuable tools to understand their chemistry and spectroscopic properties. Overall, this is an interesting and relevant work to the bioinorganic and inorganic chemistry communities.

We are thankful to Reviewer #1 for this favorable overall comment on our study.

On the other hand, while the spectroscopy of these novel and interesting molecules is analysed in some detail, the reactivity falls behind and I think this should be addressed before publication.

Reviewer #1 raised several questions on reactivities. Therefore, we have performed many new experiments, which required a relatively long period of time not only due to the number of different experiments and delays caused by the Corona-pandemic but also due to the parental leave of Dr. Stephan Walleck - the coworker performing these experiments - after receiving the Decision email from *Nature Communications*.

An obvious point of interest raised by the current work is the oxidation ability of the species resulting from protonation and also from single electron oxidation. From simple charge considerations one can infer that these may be more reactive (oxidizing) against external substrates. It is surprising that the authors do not dedicate some attention to this specific question, which seem straightforward to address given their ability to generate the hydroperoxide and high valent species in clean spectroscopic form.

The reversible protonation is studied with a strong acid and NEt₃. It will be quite interesting to titrate the complex with a weaker acid, in order to obtain an equilibrium constant that could be translated into a pK_a type of value. Combination of the pK_a value and the red-ox potential for the single e⁻ oxidation can provide a value of the BDE of the O-H bond in the Fe(OOH)Fe species. I believe this will be a valuable information for the field, since it can help place the oxidation ability of these species in the context of other metal-oxo/peroxo species of relevance to oxygenases.

Related to the previous question, the ability to protonate and oxidize the diferric peroxide strongly suggests that the Fe^{IV}Fe^{III} species should be a competent HAT agent, leading to the ferric hydroperoxide complex. This can be tested by reaction against some substrates with

known X-H bond BDE.

On the other hand, it will be quite interesting to know the oxygen atom transfer ability of these novel molecules is investigated.

We have performed all these reactivity studies encouraged by Reviewer #1 and revised the manuscript accordingly. In this respect, we are very thankful to Reviewer #1 as this really improved the manuscript.

We started by reacting the parent $[(\text{susan}^{6-\text{Me}})\{\text{Fe}^{\text{III}}(\mu\text{-O})(\mu\text{-O}_2)\text{Fe}^{\text{III}}\}]^{2+}$ complex, the oxidized $[(\text{susan}^{6-\text{Me}})\{\text{Fe}^{\text{IV}}(\mu\text{-O})(\mu\text{-O}_2)\text{Fe}^{\text{III}}\}]^{3+}$ complex, and the protonated $[(\text{susan}^{6-\text{Me}})\{\text{Fe}^{\text{III}}(\mu\text{-O})(\mu\text{-OOH})\text{Fe}^{\text{III}}\}]^{3+}$ complex with DHA as standard substrate to test for HAT and PPh_3 as standard substrate to test for OAT. Quite surprisingly, no complex showed HAT or OAT reactivity. Especially the non-reactivity of the oxidized and protonated complexes is not expected.

Therefore, we followed the advice of Reviewer #1 and determined the pK_a value of the $\mu\text{-1,2}$ -peroxo complex $[(\text{susan}^{6-\text{Me}})\{\text{Fe}^{\text{III}}(\mu\text{-O})(\mu\text{-1,2-O}_2)\text{Fe}^{\text{III}}\}]^{2+}$ providing $pK_a = 9.50 \pm 0.04$ in CH_3CN of the corresponding acid, the $\mu\text{-1,2}$ -hydroperoxo complex $[(\text{susan}^{6-\text{Me}})\{\text{Fe}^{\text{III}}(\mu\text{-O})(\mu\text{-1,2-OOH})\text{Fe}^{\text{III}}\}]^{3+}$. We have used this pK_a value together with the reduction potential for the redox couple $[(\text{susan}^{6-\text{Me}})\{\text{Fe}^{\text{IV}}(\mu\text{-O})(\mu\text{-1,2-O}_2)\text{Fe}^{\text{III}}\}]^{3+} / [(\text{susan}^{6-\text{Me}})\{\text{Fe}^{\text{III}}(\mu\text{-O})(\mu\text{-1,2-O}_2)\text{Fe}^{\text{III}}\}]^{2+}$ in CH_3CN of 0.55 V vs Fc^+/Fc to obtain the $\text{BDFE}_{\text{ACN}}(\text{O-H}) = 78 \pm 2 \text{ kcal mol}^{-1}$ of the $\mu\text{-1,2}$ -hydroperoxo complex $[(\text{susan}^{6-\text{Me}})\{\text{Fe}^{\text{III}}(\mu\text{-O})(\mu\text{-1,2-OOH})\text{Fe}^{\text{III}}\}]^{3+}$ by applying the Bordwell equation.

This quantification of the HAT reactivity of the oxidized $[(\text{susan}^{6-\text{Me}})\{\text{Fe}^{\text{IV}}(\mu\text{-O})(\mu\text{-O}_2)\text{Fe}^{\text{III}}\}]^{3+}$ complex shows that is close to the BDE of DHA explaining the non-reactivity of even the oxidized complex. We have therefore used TEMPOH with a lower BDE as substrate to test for HAT reactivity. As expected from the $\text{BDFE}_{\text{ACN}}(\text{O-H})$, the oxidized $[(\text{susan}^{6-\text{Me}})\{\text{Fe}^{\text{IV}}(\mu\text{-O})(\mu\text{-O}_2)\text{Fe}^{\text{III}}\}]^{3+}$ complex showed HAT reactivity with TEMPOH while the parent $[(\text{susan}^{6-\text{Me}})\{\text{Fe}^{\text{III}}(\mu\text{-O})(\mu\text{-O}_2)\text{Fe}^{\text{III}}\}]^{2+}$ complex and the protonated $[(\text{susan}^{6-\text{Me}})\{\text{Fe}^{\text{III}}(\mu\text{-O})(\mu\text{-OOH})\text{Fe}^{\text{III}}\}]^{3+}$ complex still do not show HAT reactivity.

This low electrophilic character of the parent $[(\text{susan}^{6-\text{Me}})\{\text{Fe}^{\text{III}}(\mu\text{-O})(\mu\text{-O}_2)\text{Fe}^{\text{III}}\}]^{2+}$ complex not able to react with TEMPOH ($\text{BDE} = 70.6 \text{ kcal mol}^{-1}$) is in contrast to its decay reactivity by hydroxylating the C-H bond of the pyridine donors (BDE approximated by that of the methyl groups of toluene of 90 kcal mol^{-1}). This strongly favors that the $\mu\text{-1,2}$ -peroxo complex $[(\text{susan}^{6-\text{Me}})\{\text{Fe}^{\text{III}}(\mu\text{-O})(\mu\text{-1,2-O}_2)\text{Fe}^{\text{III}}\}]^{2+}$ converts to transient high valent species responsible for the C-H hydroxylation as we found for the analogous $\mu\text{-1,2}$ -peroxo complex of the ligand susan.

In order to incorporate these new results, we have revised the manuscript by inserting the following parts:

In the Supplementary Information:

“Electrophilic Reactivity of [(susan^{6-Me}){Fe^{III}(μ -O)(μ -1,2-O₂)Fe^{III}]}²⁺”

A solution of the substrate in CH₃CN was added to a solution of [(susan^{6-Me}){Fe(μ -O)(μ -O₂)Fe}](ClO₄)₂ (typical concentration of 0.60 – 0.95 mM) in CH₃CN at -40 °C and the reactions were followed by UV-Vis-NIR spectroscopy. Substrates used were PPh₃ (7 equivalents), 9,10-dihydroanthracene (DHA, 100 equivalents), and 1-hydroxy-2,2,6,6-tetramethylpiperidine (TEMPOH, 100 equivalents).

Electrophilic Reactivity of [(susan^{6-Me}){Fe^{IV}(μ -O)(μ -1,2-O₂)Fe^{III}]}³⁺”

A solution of thianthrenium perchlorate (1.0 eq.) in CH₃CN/CH₂Cl₂ (1:1) was added to a solution of [(susan^{6-Me}){Fe^{III}(μ -O)(μ -O₂)Fe^{III}}](ClO₄)₂ (typical concentration of 0.50 – 0.95 mM) in CH₃CN/CH₂Cl₂ (1:1) at -60 °C. After the formation of [(susan^{6-Me}){Fe^{IV}(μ -O)(μ -O₂)Fe^{III}]}³⁺ was complete, a solution of the substrate in CH₃CN/CH₂Cl₂ (1:1) was added at -60 °C. The reactions were followed by UV-Vis-NIR spectroscopy. Substrates used were PPh₃ (20 equivalents) and 1-hydroxy-2,2,6,6-tetramethylpiperidine (TEMPOH, 100 equivalents).

Due to the relatively low solubility of DHA in CH₃CN/CH₂Cl₂ (1:1) at -60 °C and the restricted stability of [(susan^{6-Me}){Fe^{IV}(μ -O)(μ -O₂)Fe^{III}]}³⁺, the experiments using 9,10-dihydroanthracene (DHA) were modified: a solution of thianthrenium perchlorate (1.0 eq.) in CH₃CN/CH₂Cl₂ (1:1) was added to a solution of [(susan^{6-Me}){Fe^{III}(μ -O)(μ -O₂)Fe^{III}}](ClO₄)₂ (0.5 mM) and DHA (100 eq.) in CH₃CN/CH₂Cl₂ (1:1) at -60 °C. In separate experiments, thianthrenium perchlorate showed no reactivity towards DHA.

Electrophilic Reactivity of [(susan^{6-Me}){Fe^{III}(μ -O)(μ -1,2-OOH)Fe^{III}]}³⁺”

A solution of HClO₄ (1.0 eq.) in CH₃CN/CH₂Cl₂ (1:1) was added to a solution of [(susan^{6-Me}){Fe^{III}(μ -O)(μ -O₂)Fe^{III}}](ClO₄)₂ (typical concentration of 0.50 – 0.95 mM) in CH₃CN/CH₂Cl₂ (1:2) at -60 °C. After the formation of [(susan^{6-Me}){Fe^{III}(μ -O)(μ -OOH)Fe^{III}]}³⁺ was complete, a solution of the substrate in CH₃CN/CH₂Cl₂ (1:2) was added at -60 °C. The reactions were followed by UV-Vis-NIR spectroscopy. Substrates used were PPh₃ (20 equivalents) and 1-hydroxy-2,2,6,6-tetramethylpiperidine (TEMPOH, 100 equivalents).

Due to the relatively low solubility of DHA in CH₃CN/CH₂Cl₂ (1:2) at -60 °C and the restricted stability of [(susan^{6-Me})₂{Fe^{III}(μ-O)(μ-OOH)Fe^{III}}]³⁺, the experiments using 9,10-dihydroanthracene (DHA) were modified: a solution of HClO₄ (1.0 eq.) in CH₃CN/CH₂Cl₂ (1:1) was added to a solution of [(susan^{6-Me})₂{Fe^{III}(μ-O)(μ-O₂)Fe^{III}}](ClO₄)₂ (0.5 mM) and DHA (100 eq.) in CH₃CN/CH₂Cl₂ (1:2) at -60 °C. In separate experiments, HClO₄ showed no reactivity towards DHA.

Determination of *pK_a* of [(susan^{6-Me})₂{Fe^{III}(μ-O)(μ-1,2-OOH)Fe^{III}}]³⁺

One equivalent of a solution of a given acid in MeCN (30 – 60 mM) was added to a solution of [(susan^{6-Me})₂{Fe^{III}(μ-O)(μ-1,2-O₂)Fe^{III}}](ClO₄)₂ in MeCN (typical concentration of 0.5 – 0.9 mM) at -40 °C. The change of the UV/vis spectrum within the first 20 seconds was measured. Plotting the change of the normalized absorbance at 15400 cm⁻¹ versus the *pK_a* of the acid added results in a S-shaped curve, which has been fitted with a sigmoidal Boltzmann function with a Levenberg Marquardt iteration algorithm in Origin 2016.”

“**Supplementary Fig. 20.** Reaction of [(susan^{6-Me})₂{Fe^{III}(μ-O)(μ-O₂)Fe^{III}}]²⁺ (0.96 mM) with DHA (100 eq) at -40 °C in CH₃CN. Despite the change of absorption due to dilution and temperature equilibration, the spectrum of [(susan^{6-Me})₂{Fe^{III}(μ-O)(μ-O₂)Fe^{III}}]²⁺ persists.

Supplementary Fig. 21. Reactivity of [(susan^{6-Me})₂{Fe^{III}(μ-O)(μ-O₂)Fe^{III}}]²⁺ (0.95 mM) with PPh₃ (7 eq) at -40 °C in CH₃CN. Despite the change of absorption due to dilution, the spectrum of [(susan^{6-Me})₂{Fe^{III}(μ-O)(μ-O₂)Fe^{III}}]²⁺ persists.

Supplementary Fig. 22. Reaction of $[(\text{susan}^{6\text{-Me}})\{\text{Fe}^{\text{IV}}(\mu\text{-O})(\mu\text{-O}_2)\text{Fe}^{\text{III}}\}]^{3+}$ with DHA: a solution of $[(\text{susan}^{6\text{-Me}})\{\text{Fe}^{\text{III}}(\mu\text{-O})(\mu\text{-O}_2)\text{Fe}^{\text{III}}\}]^{2+}$ (0.44 mM) with DHA (100 eq) at $-60\text{ }^\circ\text{C}$ in $\text{CH}_3\text{CN}/\text{CH}_2\text{Cl}_2$ (1:1) was reacted with a solution of (thia)(ClO_4) (1.0 eq). The spectrum of $[(\text{susan}^{6\text{-Me}})\{\text{Fe}^{\text{IV}}(\mu\text{-O})(\mu\text{-O}_2)\text{Fe}^{\text{III}}\}]^{3+}$ developed quickly. Despite a slight decrease in absorption due to temperature equilibration in the first 100 s, the spectrum of $[(\text{susan}^{6\text{-Me}})\{\text{Fe}^{\text{IV}}(\mu\text{-O})(\mu\text{-O}_2)\text{Fe}^{\text{III}}\}]^{3+}$ persists.

Supplementary Fig. 23. Reaction of $[(\text{susan}^{6\text{-Me}})\{\text{Fe}^{\text{IV}}(\mu\text{-O})(\mu\text{-O}_2)\text{Fe}^{\text{III}}\}]^{3+}$ with PPh_3 : a solution of $[(\text{susan}^{6\text{-Me}})\{\text{Fe}^{\text{III}}(\mu\text{-O})(\mu\text{-O}_2)\text{Fe}^{\text{III}}\}]^{2+}$ (0.66 mM) at $-60\text{ }^\circ\text{C}$ in $\text{CH}_3\text{CN}/\text{CH}_2\text{Cl}_2$ (1:1) was reacted with a solution of (thia)(ClO_4) (1.0 eq) resulting in the spectrum of $[(\text{susan}^{6\text{-Me}})\{\text{Fe}^{\text{IV}}(\mu\text{-O})(\mu\text{-O}_2)\text{Fe}^{\text{III}}\}]^{3+}$ (red line). The spike in the time trace originates from the absorption of thianthrenium (thia^+). Addition of a solution of 20 eq. PPh_3 resulted in restoring the spectrum of parent $[(\text{susan}^{6\text{-Me}})\{\text{Fe}^{\text{III}}(\mu\text{-O})(\mu\text{-O}_2)\text{Fe}^{\text{III}}\}]^{2+}$ (blue line).

Supplementary Fig. 24. Reaction of $[(\text{susan}^{6\text{-Me}})\{\text{Fe}^{\text{III}}(\mu\text{-O})(\mu\text{-OOH})\text{Fe}^{\text{III}}\}]^{3+}$ with DHA: a solution of $[(\text{susan}^{6\text{-Me}})\{\text{Fe}^{\text{III}}(\mu\text{-O})(\mu\text{-O}_2)\text{Fe}^{\text{III}}\}]^{2+}$ (0.50 mM) with DHA (100 eq) at $-60\text{ }^\circ\text{C}$ in $\text{CH}_3\text{CN}/\text{CH}_2\text{Cl}_2$ (1:2) was reacted with a solution of HClO_4 (1.5 eq). The spectrum of $[(\text{susan}^{6\text{-Me}})\{\text{Fe}^{\text{III}}(\mu\text{-O})(\mu\text{-OOH})\text{Fe}^{\text{III}}\}]^{3+}$ (red line) developed quickly and then showed a slow decay reaction that was also observed in a blank reaction (0.52 mM, bottom figures) without DHA demonstrating that this decay is not due to reaction with DHA.

Supplementary Fig. 25. Reaction of $[(\text{susan}^{6\text{-Me}})\{\text{Fe}^{\text{III}}(\mu\text{-O})(\mu\text{-OOH})\text{Fe}^{\text{III}}\}]^{3+}$ with PPh_3 : a solution of $[(\text{susan}^{6\text{-Me}})\{\text{Fe}^{\text{III}}(\mu\text{-O})(\mu\text{-O}_2)\text{Fe}^{\text{III}}\}]^{2+}$ (1.1 mM) at -60°C in $\text{CH}_3\text{CN}/\text{CH}_2\text{Cl}_2$ (1:2) was reacted with a solution of HClO_4 (1.0 eq) resulting in the spectrum of $[(\text{susan}^{6\text{-Me}})\{\text{Fe}^{\text{III}}(\mu\text{-O})(\mu\text{-OOH})\text{Fe}^{\text{III}}\}]^{3+}$ (red line). Addition of a solution of 7 eq PPh_3 resulted in the partial restoring of the spectrum of parent $[(\text{susan}^{6\text{-Me}})\{\text{Fe}^{\text{III}}(\mu\text{-O})(\mu\text{-O}_2)\text{Fe}^{\text{III}}\}]^{2+}$ (blue line).

Supplementary Fig. 26. Boltzman fit of the acid-dependent formation of $[(\text{susan}^{6\text{-Me}})\{\text{Fe}^{\text{III}}(\mu\text{-O})(\mu\text{-OOH})\text{Fe}^{\text{III}}\}]^{3+}$ from protonation of $[(\text{susan}^{6\text{-Me}})\{\text{Fe}^{\text{III}}(\mu\text{-O})(\mu\text{-O}_2)\text{Fe}^{\text{III}}\}]^{2+}$ at -40°C in CH_3CN followed by the change in absorbance at 15400 cm^{-1} .

Supplementary Fig. 27. Reaction of $[(\text{susan}^{6\text{-Me}})\{\text{Fe}^{\text{III}}(\mu\text{-O})(\mu\text{-O}_2)\text{Fe}^{\text{III}}\}]^{2+}$ (0.60 mM) with TEMPOH (100 eq) at -40°C in CH_3CN . Despite the change

of absorption due to dilution and temperature equilibration, the spectrum of $[(\text{susan}^{6\text{-Me}})\{\text{Fe}^{\text{III}}(\mu\text{-O})(\mu\text{-O}_2)\text{Fe}^{\text{III}}\}]^{2+}$ persists.

Supplementary Fig. 28. Reaction of $[(\text{susan}^{6\text{-Me}})\{\text{Fe}^{\text{III}}(\mu\text{-O})(\mu\text{-OOH})\text{Fe}^{\text{III}}\}]^{3+}$ with TEMPOH: a solution of $[(\text{susan}^{6\text{-Me}})\{\text{Fe}^{\text{III}}(\mu\text{-O})(\mu\text{-O}_2)\text{Fe}^{\text{III}}\}]^{2+}$ (1.0 mM) at $-60\text{ }^{\circ}\text{C}$ in $\text{CH}_3\text{CN}/\text{CH}_2\text{Cl}_2$ (1:2) was reacted with a solution of HClO_4 (1.5 eq) resulting in the spectrum of $[(\text{susan}^{6\text{-Me}})\{\text{Fe}^{\text{III}}(\mu\text{-O})(\mu\text{-OOH})\text{Fe}^{\text{III}}\}]^{3+}$ (red line). Addition of a solution of 100 eq TEMPOH resulted in the restoring of the spectrum of parent $[(\text{susan}^{6\text{-Me}})\{\text{Fe}^{\text{III}}(\mu\text{-O})(\mu\text{-O}_2)\text{Fe}^{\text{III}}\}]^{2+}$ (blue line) despite the change of absorption due to dilution and temperature equilibration.

Supplementary Fig. 29. Reaction of $[(\text{susan}^{6\text{-Me}})\{\text{Fe}^{\text{IV}}(\mu\text{-O})(\mu\text{-O}_2)\text{Fe}^{\text{III}}\}]^{3+}$ with TEMPOH: a solution of $[(\text{susan}^{6\text{-Me}})\{\text{Fe}^{\text{III}}(\mu\text{-O})(\mu\text{-O}_2)\text{Fe}^{\text{III}}\}]^{2+}$ (0.70 mM) at $-60\text{ }^{\circ}\text{C}$ in $\text{CH}_3\text{CN}/\text{CH}_2\text{Cl}_2$ (1:1) was reacted with a solution of (thia)(ClO_4) (1.0 eq) resulting in the spectrum of $[(\text{susan}^{6\text{-Me}})\{\text{Fe}^{\text{IV}}(\mu\text{-O})(\mu\text{-O}_2)\text{Fe}^{\text{III}}\}]^{3+}$ (red line). Addition of a solution of 100 eq. TEMPOH resulted in restoring the spectrum of parent $[(\text{susan}^{6\text{-Me}})\{\text{Fe}^{\text{III}}(\mu\text{-O})(\mu\text{-O}_2)\text{Fe}^{\text{III}}\}]^{2+}$ (blue line) despite the change of absorption due to dilution and temperature equilibration.

In the Results section:

“Further reactivity studies

The straightforward fast protonation of $[(\text{susan}^{6\text{-Me}})\{\text{Fe}^{\text{III}}(\mu\text{-O})(\mu\text{-O}_2)\text{Fe}^{\text{III}}\}]^{2+}$ with HClO_4 demonstrates the nucleophilic character of the $\mu\text{-1,2}$ -peroxo ligand. On the other hand, the reaction of $[(\text{susan}^{6\text{-Me}})\{\text{Fe}^{\text{III}}(\mu\text{-O})(\mu\text{-O}_2)\text{Fe}^{\text{III}}\}]^{2+}$ with 2-phenylpropanal as a typical substrate to evaluate the nucleophilic character of peroxo ligands^{51,52} is slow in CH_3CN at -5°C (Fig. S19). However, the formation of roughly one equivalent of acetophenone by performing this reaction on a preparative scale for 5 days supports the slow nucleophilic reactivity of the $\mu\text{-1,2}$ -peroxo ligand.

The electrophilic character was not only evaluated for $[(\text{susan}^{6\text{-Me}})\{\text{Fe}^{\text{III}}(\mu\text{-O})(\mu\text{-O}_2)\text{Fe}^{\text{III}}\}]^{2+}$ but also for the oxidized $\mu\text{-1,2}$ -peroxo- $\text{Fe}^{\text{IV}}\text{Fe}^{\text{III}}$ and the protonated $\mu\text{-1,2}$ -hydroperoxo-diferric species as both oxidation and protonation should increase the electrophilic character. This is already reflected in the different stabilities of these species (Fig. S10). As $[(\text{susan}^{6\text{-Me}})\{\text{Fe}^{\text{III}}(\mu\text{-O})(\mu\text{-1,2-O}_2)\text{Fe}^{\text{III}}\}]^{2+}$ shows no indication of decay for hours in CH_3CN at -40°C , oxidized $[(\text{susan}^{6\text{-Me}})\{\text{Fe}^{\text{IV}}(\mu\text{-O})_2(\mu\text{-O}_2)\text{Fe}^{\text{III}}\}]^{3+}$ and protonated $[(\text{susan}^{6\text{-Me}})\{\text{Fe}^{\text{III}}(\mu\text{-O})(\mu\text{-1,2-OOH})\text{Fe}^{\text{III}}\}]^{3+}$ decay with half-lives of $\tau_{1/2} \approx 90$ min $\tau_{1/2} \approx 11$ min, respectively. Therefore, the clean characterization of the latter two species required lower temperatures of -60°C and hence addition of a certain amount of CH_2Cl_2 .

The electrophilic character of the three complexes were initially investigated using 9,10-dihydroanthracene (DHA) and PPh_3 as typical substrates for HAT and OAT, respectively. The reactions with $[(\text{susan}^{6\text{-Me}})\{\text{Fe}^{\text{III}}(\mu\text{-O})(\mu\text{-$

$\text{O}_2\text{Fe}^{\text{III}}\}}]^{2+}$ were performed in CH_3CN at -40°C , while that with $[(\text{susan}^{6\text{-Me}}\{\text{Fe}^{\text{IV}}(\mu\text{-O})_2(\mu\text{-O}_2)\text{Fe}^{\text{III}}\})]^{3+}$ and $[(\text{susan}^{6\text{-Me}}\{\text{Fe}^{\text{III}}(\mu\text{-O})(\mu\text{-1,2-OOH})\text{Fe}^{\text{III}}\})]^{3+}$ at -60°C in $\text{CH}_3\text{CN}/\text{CH}_2\text{Cl}_2$ mixtures (*vide supra*). The parent $[(\text{susan}^{6\text{-Me}}\{\text{Fe}^{\text{III}}(\mu\text{-O})(\mu\text{-O}_2)\text{Fe}^{\text{III}}\})]^{2+}$ showed no reactivity towards DHA and PPh_3 (Fig. S20+S21). The oxidized $[(\text{susan}^{6\text{-Me}}\{\text{Fe}^{\text{IV}}(\mu\text{-O})_2(\mu\text{-O}_2)\text{Fe}^{\text{III}}\})]^{3+}$ also showed no reactivity towards DHA (Fig. S22), while the reaction with PPh_3 resulted in the reoccurrence of the UV-Vis signature of the parent $[(\text{susan}^{6\text{-Me}}\{\text{Fe}^{\text{III}}(\mu\text{-O})(\mu\text{-O}_2)\text{Fe}^{\text{III}}\})]^{2+}$ (Fig. S23). The reformation of the $\mu\text{-1,2-peroxo}\rightarrow\text{Fe}^{\text{III}}$ LMCT excludes an OAT reactivity between $[(\text{susan}^{6\text{-Me}}\{\text{Fe}^{\text{IV}}(\mu\text{-O})_2(\mu\text{-O}_2)\text{Fe}^{\text{III}}\})]^{3+}$ and PPh_3 but suggests an oxidation of PPh_3 ⁵³ by $[(\text{susan}^{6\text{-Me}}\{\text{Fe}^{\text{IV}}(\mu\text{-O})_2(\mu\text{-O}_2)\text{Fe}^{\text{III}}\})]^{3+}$. Analogous observations were made with the protonated $[(\text{susan}^{6\text{-Me}}\{\text{Fe}^{\text{III}}(\mu\text{-O})(\mu\text{-1,2-OOH})\text{Fe}^{\text{III}}\})]^{3+}$ that showed no reactivity with DHA (Fig. S24) and with PPh_3 the partial recovery of the UV-Vis signature of the parent $[(\text{susan}^{6\text{-Me}}\{\text{Fe}^{\text{III}}(\mu\text{-O})(\mu\text{-O}_2)\text{Fe}^{\text{III}}\})]^{2+}$ (Fig. S25). Again, the reformation of the $\mu\text{-1,2-peroxo}\rightarrow\text{Fe}^{\text{III}}$ LMCT excludes an OAT reactivity between $[(\text{susan}^{6\text{-Me}}\{\text{Fe}^{\text{III}}(\mu\text{-O})(\mu\text{-1,2-OOH})\text{Fe}^{\text{III}}\})]^{3+}$ and PPh_3 . The partial recovery of $[(\text{susan}^{6\text{-Me}}\{\text{Fe}^{\text{III}}(\mu\text{-O})(\mu\text{-O}_2)\text{Fe}^{\text{III}}\})]^{2+}$ indicates a protonation equilibrium between PPh_3 ⁵⁴ and $[(\text{susan}^{6\text{-Me}}\{\text{Fe}^{\text{III}}(\mu\text{-O})(\mu\text{-1,2-OOH})\text{Fe}^{\text{III}}\})]^{3+}$.

As it is not surprising^{55,35} that $[(\text{susan}^{6\text{-Me}}\{\text{Fe}^{\text{III}}(\mu\text{-O})(\mu\text{-O}_2)\text{Fe}^{\text{III}}\})]^{2+}$ exhibits no electrophilic reactivity against DHA, the non-reactivity of both oxidized $[(\text{susan}^{6\text{-Me}}\{\text{Fe}^{\text{IV}}(\mu\text{-O})_2(\mu\text{-O}_2)\text{Fe}^{\text{III}}\})]^{3+}$ and protonated $[(\text{susan}^{6\text{-Me}}\{\text{Fe}^{\text{III}}(\mu\text{-O})(\mu\text{-1,2-OOH})\text{Fe}^{\text{III}}\})]^{3+}$ is quite surprising. To further understand this non-reactivity, we determined the $\text{BDFE}(\text{OH})_{\text{CH}_3\text{CN}}$ of $[(\text{susan}^{6\text{-Me}}\{\text{Fe}^{\text{III}}(\mu\text{-O})(\mu\text{-1,2-OOH})\text{Fe}^{\text{III}}\})]^{3+}$. In this respect, we determined the pK_a of $[(\text{susan}^{6\text{-Me}}\{\text{Fe}^{\text{III}}(\mu\text{-O})(\mu\text{-1,2-OOH})\text{Fe}^{\text{III}}\})]^{3+}$ in CH_3CN that provided 9.5 ± 0.1 (Fig. S26). Using the typical square scheme (Fig. 6a) and the Bordwell relation eq. (1)⁵⁶⁻⁵⁸

$$\text{BDFE}(\text{O-H})_{\text{CH}_3\text{CN}} = 1.37 pK_a + 23.06 E^{\text{O}'} + 52.6 \text{ kcal mol}^{-1} \quad (1)$$

provided $\text{BDFE}(\text{O-H})_{\text{CH}_3\text{CN}} = 78\pm 2 \text{ kcal mol}^{-1}$. This means that $[(\text{susan}^{6\text{-Me}}\{\text{Fe}^{\text{IV}}(\mu\text{-O})_2(\mu\text{-O}_2)\text{Fe}^{\text{III}}\})]^{3+}$ should be capable as oxidant for HAT for substrates with a lower $\text{BDFE}(\text{X-H})_{\text{CH}_3\text{CN}}$. The $\text{BDE}(\text{C-H})$ of DHA is $76.3 \text{ kcal mol}^{-1}$.⁵⁹ However, the intrinsic difference between BDFE and BDE,⁵⁷ the temperature-dependence of BDFE especially for transition metal complexes, and the experimental error explain the non-reactivity of $[(\text{susan}^{6\text{-Me}}\{\text{Fe}^{\text{IV}}(\mu\text{-O})_2(\mu\text{-O}_2)\text{Fe}^{\text{III}}\})]^{3+}$.

$O)_2(\mu-O_2)Fe^{III}]^{3+}$ with DHA. In this respect, TEMPOH should be a suitable HAT substrate with $BDFE(O-H)_{CH_3CN} = 66.5 \text{ kcal mol}^{-1}$ and $BDE(O-H) = 70.6 \text{ kcal mol}^{-1}$ ⁵⁷ especially for $[(susan^{6-Me})\{Fe^{IV}(\mu-O)_2(\mu-O_2)Fe^{III}\}]^{3+}$.

The parent $[(susan^{6-Me})\{Fe^{III}(\mu-O)(\mu-O_2)Fe^{III}\}]^{2+}$ showed no reactivity with TEMPOH (Fig. S27), which is line that this one-electron reduced species $[(susan^{6-Me})\{Fe^{III}(\mu-O)(\mu-O_2)Fe^{III}\}]^{2+}$ should have a driving force for HAT significantly lower than 78 kcal mol^{-1} of $[(susan^{6-Me})\{Fe^{IV}(\mu-O)_2(\mu-O_2)Fe^{III}\}]^{3+}$. The reaction of protonated $[(susan^{6-Me})\{Fe^{III}(\mu-O)(\mu-1,2-OOH)Fe^{III}\}]^{3+}$ with TEMPOH resulted in the reoccurrence of the UV-Vis signature of the parent $[(susan^{6-Me})\{Fe^{III}(\mu-O)(\mu-O_2)Fe^{III}\}]^{2+}$ (Fig. S28). The reformation of the parent $\mu-1,2$ -peroxo-diferric complex excludes a HAT reactivity between $[(susan^{6-Me})\{Fe^{III}(\mu-O)(\mu-1,2-OOH)Fe^{III}\}]^{3+}$ and TEMPOH and suggests a protonation of TEMPOH.⁶⁰ The reaction of $[(susan^{6-Me})\{Fe^{IV}(\mu-O)_2(\mu-O_2)Fe^{III}\}]^{3+}$ with TEMPOH resulted in the reoccurrence of the UV-Vis signature of the parent $[(susan^{6-Me})\{Fe^{III}(\mu-O)(\mu-O_2)Fe^{III}\}]^{2+}$ (Fig. S29). This is in-line with HAT from TEMPOH to $[(susan^{6-Me})\{Fe^{IV}(\mu-O)_2(\mu-O_2)Fe^{III}\}]^{3+}$ resulting in $[(susan^{6-Me})\{Fe^{III}(\mu-O)(\mu-1,2-OOH)Fe^{III}\}]^{3+}$ that reacts again by protonation of excess TEMPOH to $[(susan^{6-Me})\{Fe^{III}(\mu-O)(\mu-O_2)Fe^{III}\}]^{2+}$.

Thus, only the oxidized $[(susan^{6-Me})\{Fe^{IV}(\mu-O)_2(\mu-O_2)Fe^{III}\}]^{3+}$ is capable for HAT from TEMPOH corroborated by the BDFE while the parent $[(susan^{6-Me})\{Fe^{III}(\mu-O)(\mu-O_2)Fe^{III}\}]^{2+}$ and the protonated $[(susan^{6-Me})\{Fe^{III}(\mu-O)(\mu-1,2-OOH)Fe^{III}\}]^{3+}$ does not exhibit enough electrophilic character. However, $[(susan^{6-Me})\{Fe^{III}(\mu-O)(\mu-O_2)Fe^{III}\}]^{2+}$ that shows no decay at -40°C exhibits a change in the UV-Vis spectra at room temperature with the formation of the typical signature of complexes with a $\{Fe^{III}X(\mu-O)Fe^{III}X\}$ core³⁸ without the observation of intermediates accompanied by deposition of an inhomogenous solid and a few single-crystals. The crystallographic analysis provided the structure of the decay product (Fig. 6b) based on $[(susan^{6-Me})\{Fe^{III}(OH)(\mu-O)Fe^{III}(OH)\}]^{2+}$ ⁶¹ with a disorder of the coordinated OH⁻ ligands, which could be resolved to coordination of oxidized 6-methyl groups. At Fe1, only 20% is OH⁻ while 80% is a carboxylate while at Fe2, 65% is OH⁻ and 35% consists of a benzylalcoholato donor. NMR spectroscopy of the demetalated bulk decay product shows the formation of more than one

product but a significant signal at 173.4 ppm for a benzoic acid group in the ^{13}C NMR spectrum indicates that the hydroxylation of the 6-methyl group is not only a minor reaction path.

Figure 6. Reactivity studies. a) Square scheme showing the PCET thermochemistry of $[(\text{susan}^{6\text{-Me}})\{\text{Fe}^{\text{III}}(\mu\text{-O})(\mu\text{-1,2-OOH})\text{Fe}^{\text{III}}\}]^{3+}$ in CH_3CN .

In the Discussion section:

“The study of the electrophilic reactivity demonstrated only a low electrophilic character of the parent $\mu\text{-1,2-peroxo-Fe}^{\text{III}}\text{Fe}^{\text{III}}$ complex that is not unexpected for such complexes.^{55,35} Interestingly, also protonation to the $\mu\text{-1,2-hydroperoxo-Fe}^{\text{III}}\text{Fe}^{\text{III}}$ species turned out to be not sufficient to increase the electrophilic character for HAT with substrates of weak to modest BDE (TEMPOH and DHA). Only the oxidized $\mu\text{-1,2-peroxo-Fe}^{\text{IV}}\text{Fe}^{\text{III}}$ species reacts with the relatively weak substrate TEMPOH. The determination of the $pK_a=9.5\pm 0.1$ and the bond dissociation free energy $\text{BDFE}(\text{O-H})_{\text{CH}_3\text{CN}} = 78\pm 2 \text{ kcal mol}^{-1}$ of the protonated $\mu\text{-1,2-hydroperoxo-Fe}^{\text{III}}\text{Fe}^{\text{III}}$ species quantifies this low electrophilic character even of the oxidized $\mu\text{-1,2-peroxo-Fe}^{\text{IV}}\text{Fe}^{\text{III}}$ species. In this respect, the intramolecular C-H activation of preorganized 6-methyl pyridine groups to benzylalcoholato and carboxylato donors by the parent $\mu\text{-1,2-peroxo-Fe}^{\text{III}}\text{Fe}^{\text{III}}$ complex is remarkable. Considering that this complex does not react with TEMPOH ($\text{BDE}(\text{O-H}) = 70.6 \text{ kcal mol}^{-1}$) and using the $\text{BDE}(\text{C-H}) = 90 \text{ kcal mol}^{-1}$ ⁵⁹ of the methyl group of toluene as an approximation for the $\text{BDE}(\text{C-H})$ of the 6-methyl groups of the coordinated pyridines, HAT should not occur *via* the

bridging μ -1,2-peroxo-ligand. This indicates that this intramolecular reaction requires the conversion of the μ -1,2-peroxo-diferric core to a more reactive high-valent species as already postulated for the CH₃OH oxidation⁶² of the analogous μ -1,2-peroxo-diferric complex of susan (*vide supra*).”

In the Abstract:

“Neither the oxidation nor the protonation induces a strong electrophilic reactivity. Hence, the observed intramolecular C-H hydroxylation of preorganized methyl groups of the parent μ -1,2-peroxo-diferric complex should occur *via* conversion to a more electrophilic high-valent species.”

In the Introduction section:

“The study of the electrophilic reactivity for oxygen-atom transfer (OAT) using PPh₃ and hydrogen-atom transfer (HAT) using DHA and TEMPOH provides not only a low electrophilic character of the parent μ -1,2-peroxo-Fe^{III}Fe^{III} complex but also for the oxidized 1,2-peroxo-Fe^{IV}Fe^{III} and protonated 1,2-hydroperoxo-Fe^{III}Fe^{III} species. Only the oxidized 1,2-peroxo-Fe^{IV}Fe^{III} species reacts with the relatively weak substrate TEMPOH. The determination of the $pK_a=9.5\pm 0.1$ and the bond dissociation free energy $BDFE(OH)_{CH_3CN} = 78\pm 2 \text{ kcal mol}^{-1}$ of the protonated 1,2-hydroperoxo-Fe^{III}Fe^{III} species quantifies the low electrophilic character even of the oxidized 1,2-peroxo-Fe^{IV}Fe^{III} species. Therefore, the intramolecular C-H activation of preorganized 6-methyl pyridine groups to benzylalcoholato and carboxylato donors in the parent 1,2-peroxo-Fe^{III}Fe^{III} complex should not occur *via* the 1,2-peroxo-ligand but *via* conversion to a more reactive but fluent high-valent species. The low electrophilic character and the spectroscopic signatures of this first μ -1,2-hydroperoxo-diferric model is discussed in relation to assignments of reactive intermediates postulated for diiron enzymes.”

The authors describe the formation of small amounts of crystals where the ligand is hydroxylated. This is quite an interesting reaction but It is not possible to deduce from this analysis if this is a minor path or if it is a major reaction path. To address this question authors can demetalate the final compound and analyse the organics to

know the percentage of ligand oxidized.

We have followed the advice of Reviewer #1 and have demetalated the final compound and analyzed by ^1H and ^{13}C NMR spectroscopy. While in both spectra no significant signals for a benzyl alcohol group appear, the ^{13}C NMR shows a significant signal for a benzoic acid group at 173.4 ppm. We have thus added the following to the revised manuscript:

“NMR spectroscopy of the demetalated bulk decay product shows the formation of more than one product but a significant signal at 173.4 ppm for a benzoic acid group in the ^{13}C NMR spectrum indicates that the hydroxylation of the 6-methyl group is not only a minor reaction path.”

The reaction with propanal requires authors to identify the final product. The decay observed upon addition of the aldehyde does not necessarily imply the suggested nucleophilic reaction.

We have performed the reaction with 2-phenylpropanal on a preparative scale providing the formation of roughly one equivalent of acetophenone as the product of the nucleophilic reactivity. Accordingly, we have revised the manuscript from

“The straightforward protonation of $[(\text{susan}^{6-\text{Me}})\{\text{Fe}^{\text{III}}(\mu\text{-O})(\mu\text{-O}_2)\text{Fe}^{\text{III}}\}]^{2+}$ demonstrates the nucleophilic character of the $\mu\text{-1,2}$ -peroxo ligand, although its reaction with 2-phenylpropanal, a typical reagent for nucleophilic peroxo ligands,^{49,50} at -5°C in CH_3CN is slow (Fig. S18).”

to the following in the revised manuscript:

“The straightforward and fast protonation of $[(\text{susan}^{6-\text{Me}})\{\text{Fe}^{\text{III}}(\mu\text{-O})(\mu\text{-O}_2)\text{Fe}^{\text{III}}\}]^{2+}$ with HClO_4 demonstrates the nucleophilic character of the $\mu\text{-1,2}$ -peroxo ligand. On the other hand, the reaction of $[(\text{susan}^{6-\text{Me}})\{\text{Fe}^{\text{III}}(\mu\text{-O})(\mu\text{-O}_2)\text{Fe}^{\text{III}}\}]^{2+}$ with 2-phenylpropanal as a typical substrate to evaluate the nucleophilic character of peroxo ligands,^{52,53} is slow in CH_3CN at -5°C (Fig. S18). Performing this reaction on a preparative scale for 5 days showed the formation of roughly one equivalent of acetophenone as the product of the nucleophilic reactivity of the coordinated peroxide.”

The MB analysis convincingly shows that the iron centers remain high spin in the peroxide, hydroperoxide and high valent species. I recommend authors to make an explicit mention to this aspect. Notice that the EPR of the mixed valence compound shows signals corresponding to a $S = 1/2$ system that could be also interpreted as a low spin ferric

center.

According to the reviewer's recommendation, we have added the following explicit mentioning regarding assignment to a Fe^{III} I.s. species:

"Mössbauer spectroscopy demonstrates that the iron ions remain high-spin ruling out an interpretation as Fe^{III} I.s. species."

I wonder if the spectroscopic data excludes the possibility that protonation leads to a terminally bound hydroperoxide ligand ligated to only one of the two centers. Some discussion will be appreciated.

In the light of the reviewer's comment, we have added the following discussion at page 10 of the revised manuscript:

"Although a terminally bound hydroperoxo-ligand would be in-line with the absence of a μ -1,2-peroxo \rightarrow Fe^{III} LMCT and two quadrupole doublets, the persistence of the μ -oxo \rightarrow Fe^{III} LMCTs around 19000 and 11800 cm⁻¹ strongly favors a doubly-bridged structure of almost the same $\angle(\text{Fe}^{\text{III}}-(\mu\text{-O})\text{-Fe}^{\text{III}})$ angle hence ruling out an almost linear $\{\text{Fe}^{\text{III}}(\text{OOH})(\mu\text{-O})\text{Fe}^{\text{III}}\text{X}\}$ core^{38,40} that is also inaccessible with the ligand susan^{6-Me} (*vide infra*)."

Minor questions;

Page two, authors refer to "higher isomer shift". For chemists not familiar with iron chemistry it may not be obvious that the authors refer to Mossbauer spectroscopy. I recommend explicit mention.

We have changed the manuscript in the light of the reviewer's comment from

"In other non-heme diiron enzymes,¹²⁻²⁰ a peroxide activation step has been proposed by the conversion of **P**-type to **P'**-type intermediates that lack the peroxo \rightarrow Fe^{III} LMCT around 14000-15000 cm⁻¹ and the higher isomer shift characteristic for **P**-type intermediates."

to the following in the revised manuscript:

"In other non-heme diiron enzymes,¹²⁻²⁰ a peroxo activation step has been proposed by the conversion of **P**-type to **P'**-type intermediates that lack the peroxo \rightarrow Fe^{III} LMCT around 14000-15000 cm⁻¹ and the higher Mössbauer isomer shift characteristic for **P**-type intermediates."

Page 7. Authors argue on a change from 98 to 95% implying some chemical reduction during the coulometric experiments. This seems a value within experimental error.

We agree with the reviewer that a change from 98% to 95% would be within experimental error. However, we argue the difference between restoring 95% of the UV-Vis spectrum of the peroxo starting complex, which required only 76% charge for the re-reduction of the oxidized species. This difference is without experimental error.

Page 10. I don't understand how the authors assign the weak 11800 cm⁻¹ feature to a CT band. Please explain.

We have studied the diferric complexes of our dinucleating ligand system extensively especially with regard to the UV-Vis-NIR spectra and made assignments with respect to the literature:

In general, linear mono- μ -oxo-bridged $\{\text{Fe}^{\text{III}}(\mu\text{-O})\text{Fe}^{\text{III}}\}$ complexes exhibit two weak transitions around 10000 cm⁻¹ and 18000 cm⁻¹ that were assigned to the ${}^6\text{A}_1 \rightarrow {}^4\text{T}_1$ and ${}^6\text{A}_1 \rightarrow {}^4\text{T}_2$ d-d transitions, respectively,[1] e.g. at 10540 cm⁻¹ with $\epsilon = 7 \text{ M}^{-1} \text{ cm}^{-1}$ and 17400 cm⁻¹ with $\epsilon = 140 \text{ M}^{-1} \text{ cm}^{-1}$ in $[(\text{susan})\{\text{Fe}^{\text{III}}\text{Cl}(\mu\text{-O})\text{Fe}^{\text{III}}\text{Cl}\}]^{2+}$. The very strong intensity for d-d transitions that are spin-forbidden in monoferric complexes were related to some allowed probability for transitions between total spin states created by the exchange coupling in μ -oxo-bridged diferric complexes.[1] However, the assignment of the higher-energy transition to a d-d transition is questionable.

On the other hand, by decreasing the Fe- μ -oxo-Fe angle due to a second or third bridging ligand as carboxylate, carbonate, or phosphate, the transition around 18000 cm⁻¹ is absent and a new transition around 14000 cm⁻¹ arises, e.g. in $[(\text{susan})\{\text{Fe}^{\text{III}}(\mu\text{-O})(\mu\text{-OAc})\text{Fe}^{\text{III}}\}]^{3+}$ at 14400 cm⁻¹ with $\epsilon = 130 \text{ mol}^{-1}\text{cm}^{-1}$. The change in the UV-Vis-NIR spectra can also be described as a shift of 17400 cm⁻¹ band in the linear mono-bridged complex to 14400 cm⁻¹ in the doubly-bridged complex. However, the weak low-energy d-d transition persists on decreasing the Fe- μ -oxo-Fe, e. g. the ${}^6\text{A}_1 \rightarrow {}^4\text{T}_1$ transition stays at 10000 cm⁻¹ by going to $[(\text{susan})\{\text{Fe}^{\text{III}}(\mu\text{-O})(\mu\text{-OAc})\text{Fe}^{\text{III}}\}]^{2+}$ (Fe- μ -oxo-Fe angle of 131.8°), which is an argument against the assignment of the 14400 cm⁻¹ band to the ${}^6\text{A}_1 \rightarrow {}^4\text{T}_2$ d-d transition, as both d-d transitions should have roughly the same shift in energy according to the Tanabe-Sugano diagram.

The group of Que observed the same transitions in tpa-based complexes with a decrease in energy by decreasing the Fe- μ -oxo-Fe angle.[2] From the comparison of several doubly-bridged μ -oxo-diferric complexes, they concluded that this transition cannot be the ${}^6\text{A}_1 \rightarrow {}^4\text{T}_2$ d-d transition and they assigned it to a forbidden μ -oxo $\rightarrow \text{Fe}^{\text{III}}$ LMCT transition. We came to the same conclusion in a series of doubly-bridged diferric

susan complexes.[3]

The UV-Vis-NIR spectra of all $\{\text{Fe}^{\text{III}}(\mu\text{-oxo})(\mu\text{-}1,2\text{-peroxo})\text{Fe}^{\text{III}}\}$ complexes exhibits two bands around 15000 and 19000 cm^{-1} of roughly $\varepsilon \approx 1000 \text{ M}^{-1} \text{ cm}^{-1}$ and a less-intense low-energy feature with $\varepsilon \approx 100 \text{ M}^{-1} \text{ cm}^{-1}$. While the assignment of the two more intense bands to $\mu\text{-peroxo} \rightarrow \text{Fe}^{\text{III}}$ LMCT and $\mu\text{-oxo} \rightarrow \text{Fe}^{\text{III}}$ LMCT, respectively, is well established in the literature, the assignment of the low energy feature is not explicitly mentioned in the literature and it could be a LMCT associated with the $\mu\text{-peroxo}$ and with the $\mu\text{-oxo}$ bridge.

In $[(\text{susan}^{6\text{-Me}})\{\text{Fe}^{\text{III}}(\mu\text{-O})(\mu\text{-}1,2\text{-O}_2)\text{Fe}^{\text{III}}\}]^{2+}$, this transition occurs at 11800 cm^{-1} with $\varepsilon = 190 \text{ mol}^{-1}\text{cm}^{-1}$. The Fe- $\mu\text{-oxo}$ -Fe angle decreases from 131.8° in $[(\text{susan})\{\text{Fe}^{\text{III}}(\mu\text{-O})(\mu\text{-OAc})\text{Fe}^{\text{III}}\}]^{3+}$ to 122.8° in $[(\text{susan}^{6\text{-Me}})\{\text{Fe}^{\text{III}}(\mu\text{-O})(\mu\text{-}1,2\text{-O}_2)\text{Fe}^{\text{III}}\}]^{2+}$. Thus, the 11800 cm^{-1} ($\varepsilon = 190 \text{ mol}^{-1}\text{cm}^{-1}$) band in the latter seems to be of the same origin as the 14400 cm^{-1} ($\varepsilon = 130 \text{ mol}^{-1}\text{cm}^{-1}$) transition in $[(\text{susan})\{\text{Fe}^{\text{III}}(\mu\text{-O})(\mu\text{-OAc})\text{Fe}^{\text{III}}\}]^{2+}$. With regard to the findings of Que *et al.* and our own mentioned above, we were already very certain that this transition is a $\mu\text{-oxo} \rightarrow \text{Fe}^{\text{III}}$ LMCT but would have not published this assignment.

In this respect, the persistence of this band at 11800 cm^{-1} and of the $\mu\text{-oxo} \rightarrow \text{Fe}^{\text{III}}$ LMCT band at 19300 cm^{-1} shows that these two bands are $\mu\text{-oxo} \rightarrow \text{Fe}^{\text{III}}$ LMCT transitions in $\mu\text{-oxo}$ -bridged diferric complexes with the $\text{Fe}^{\text{III}}\text{-}\mu\text{-oxo}\text{-Fe}^{\text{III}}$ close to 123°. Due to this additional piece of evidence, we have stated this explicitly in the manuscript.

- [1] C. A. Brown, G. J. Remar, R. L. Musselman and E. I. Solomon, *Inorg. Chem.*, 1995, **34**, 688–717
- [2] R. E. Norman, R. C. Holz, S. Menage, C. J. O'Connor, J. H. Zhang and L. Que Jr., *Inorg. Chem.*, 1990, **29**, 4629–4637
- [3] T. P. Zimmermann, T. Limpke, A. Stammer, H. Bögge, S. Walleck and T. Glaser, *Z. Anorg. Allg. Chem.*, 2018, **644**, 683-691

Page 10. It is a bit confusing that the UV-vis spectra use cm^{-1} as units and then the discussion in the text uses nm to describe the wavelength of Raman excitations. I recommend to plot the UV-vis data in nm.

We agree with the reviewer that using both cm^{-1} and nm disturbs the readability of the manuscript. We have provided the laser wavelengths in nm as it is usual in the laser community. However, although we do not want to make a big discussion on this but providing UV-Vis spectra on a wavelength or a wavenumber scale is not just a question of style. A wavelength scale is not proportional to the energy but only inverse proportional to energy while a wavenumber scale is directly proportional to the energy.

The use of wavelengths deforms the energy scale, *i. e.* an energy difference of 100 nm at high energy corresponds to a much larger energy difference than a 100 nm energy difference at low energy. As I was trained by Prof. Ed Solomon from Stanford University, I strongly prefer a non-distorted energy scale and always use wavenumbers to plot UV-Vis spectra.

However, for a better readability, we have changed throughout the manuscript from

“(bottom, 633 nm excitation)”

to

“(bottom, 633 nm (15800 cm^{-1}) excitation)”

Page 14. I am not sure the intensity extracted from extinction coefficients can be directly used to compare the donor strength, but instead the area of the bands should be used. Please reconsider.

The reviewer is correct that - as we stated in the manuscript - the donor strength of a given ligand can be extracted from the integrated absorption intensity. Although we have provided in the manuscript only the ϵ values, these correlate usually quite well with the integrated absorption intensities as it is the case here. To evaluate this we plotted the UV-Vis-NIR spectrum of our peroxy complex (right) as closely as that of the published spectrum of the peroxy complex **A** from Zang *et al.* (left). The comparison is shown below (please note that the peroxy complexes are plotted in magenta):

The ratio of the integrated absorption intensity of the μ -oxo \rightarrow Fe^{III} LMCT and μ -peroxy \rightarrow Fe^{III} LMCT is reversed in the two μ -peroxy complexes: in the μ -peroxy

complex **A** the μ -peroxo \rightarrow Fe^{III} LMCT is much more intense than the μ -oxo \rightarrow Fe^{III} LMCT while in our μ -peroxo complex the μ -oxo \rightarrow Fe^{III} LMCT is more intense than the μ -peroxo \rightarrow Fe^{III} LMCT.

As we have stated in our manuscript: "Without the intention of a quantitative analysis," we refrain from a more quantitative analysis. However, to be more precise as indicated by the reviewer, we have changed from the submitted manuscript

„Without the intention of a quantitative analysis, this indicates less charge-donation and hence more nucleophilic character of the μ -peroxo in the susan^{6-Me} complex.“

to the following in the revised manuscript:

„Note that also the integrated absorption intensity is smaller in the susan^{6-Me} complex than in **A** for the peroxo \rightarrow Fe^{III} LMCT indicating - without the intention of a quantitative analysis - less charge-donation and hence more nucleophilic character of the μ -peroxo.“

Page 14. I can not understand how the authors translate the Fe-O2-Fe vibration energy into nucleophilicity. Please explain.

The translation of the $\nu_{\text{sym}}(\text{Fe}-\mu\text{-O}_2\text{-Fe})$ vibration energy closely follows that of the μ -peroxo \rightarrow Fe^{III} LMCT. The analysis of μ -peroxo \rightarrow Fe^{III} LMCT indicates a stronger π donor interaction between the peroxo ligand and the Fe^{III} ions resulting in a stronger Fe-O^{peroxo} bond in complex **A**. A stronger bond is correlated with a stronger force constant k . As the two complexes do not differ too strongly, the reduced mass dependence should be negligible so that the vibrational frequency should correlate with the bond strengths. The stronger Fe-O^{peroxo} bond in complex **A** obtained from the analysis of μ -peroxo \rightarrow Fe^{III} LMCT is thus correlated with the higher value for $\nu_{\text{sym}}(\text{Fe}-\mu\text{-O}_2\text{-Fe})$ of 465 cm⁻¹ in **A** compared to 448 cm⁻¹ in our peroxo complex. As the peroxo ligand is a σ and π donor, a stronger bond is related with more electron donation from the peroxo ligand and hence less electron density at the peroxo oxygen atoms and hence less nucleophilicity.

Response to Reviewer #2

The authors report the reactivity of a dinuclear Fe(II) complex using a ligand similar to that described over 30 years ago by Armstrong and Toftlund (might be an idea to cite these studies!), which have been studied for catalytic properties in oxidation catalysis over the years.

We are well aware of the wonderful work of Hans Toftlund and his coworkers. They published in 1996 the dinucleating ligand btpa.[1] With this ligand, they made very nice research not only on oxidation catalysis but also on mononuclear iron complexes for spin transition.[2,3]

- [1] A. Døssing, A. Hazell, H. Toftlund, *Acta Chem. Scand.* **1996**, *50*, 95–101
- [2] S. Schenker, P. C. Stein, J. A. Wolny, C. Brady, J. J. McGarvey, H. Toftlund, A. Hauser, *Inorg. Chem.* **2001**, *40*, 134–139
- [3] C. Brady, P. L. Callaghan, Z. Ciunik, C. G. Coates, A. Døssing, A. Hazell, J. J. McGarvey, S. Schenker, H. Toftlund, A. X. Trautwein et al., *Inorg. Chem.* **2004**, *43*, 4289–4299

Our dinucleating ligand system does not contain a central bipyridine spacer but a central ethylenediamine spacer. In our publications describing the design and synthesis of this new ligand system, we have cited this beautiful work of Toftlund and coworkers.[4-7]

- [4] J. B. H. Strautmann, S. Dammers, T. Limpke, J. Parthier, T. P. Zimmermann, S. Walleck, G. Heinze-Brückner, A. Stammler, H. Bögge, T. Glaser, *Dalton Trans.* **2016**, *45*, 3340–3361
- [5] M. Aschenbrenner, A. Stammler, H. Bögge, T. Glaser, *Z. Anorg. Allg. Chem.* **2018**, *644*, 1439–1444
- [6] T. P. Zimmermann, S. Dammers, A. Stammler, H. Bögge, T. Glaser, *Eur. J. Inorg. Chem.* **2018**, *48*, 5229–5237
- [7] T. Glaser, *Coord. Chem. Rev.* **2019**, *380*, 353–377

However, for this manuscript, we have a restriction for the number of references. Dinuclear iron complexes with either mononucleating or dinucleating ligands used in oxidation catalysis are numerous in the literature and so we cannot cite them all. In this manuscript, we have restricted the references to those that explicitly have observed

and/or characterized peroxo diiron complexes with either mononucleating or dinucleating ligands. Due to the almost unlimited numbers of studies of diiron complexes for oxidation catalysis and the restricted numbers of references allowed, it would not be fair to cite very selected publications that do not involve explicit observation and characterization of peroxo diiron complexes. Therefore, we do not want to follow the advice of Reviewer #2 to cite these excellent studies of Hans Toftlund and coworkers in this manuscript.

On the other hand, concerning the recommendation to cite studies by Armstrong (with the interpretation that Reviewer #2 refers to William H. Armstrong, who made great contributions to manganese complexes related to photosystem II), we were not aware of publications where William H. Armstrong used a ligand similar to ours. Due to the recommendation of Reviewer #2, we have again searched the literature using *CSD* and *Web of Science*, but could not find a publication of William H. Armstrong using a ligand similar to that we used in this manuscript.

The presence of a methyl group at the 6 position of the pyridyl has been explored before mainly by the Que group and the oxidative degradation of similar ligands by several groups. Typically these complexes are isolated already in the Fe(III) oxidation state with oxido/carboxylato bridges and so their study in the Fe(II) state is novel and especially their reactivity with molecular oxygen is of interest for bioinorganic chemistry. Hence the work is novel and relevant.

We are thankful to Reviewer #2 for this comment.

The characterization, in particular by vibrational spectroscopy with labelling is indeed of a good standard but nowadays routine. Hence the fact that the manuscript is filled with superlatives (thoroughly, unprecedented...) which make the reading of it somewhat irksome. There is no need to oversell relatively routine/expected characterization as if it were well beyond the norm.

We agree with Reviewer #2 that the vibrational characterization of peroxo diferric complexes using resonance Raman spectroscopy on labeled compounds is routine

nowadays. However, such studies are usually performed in solution on *in situ* generated ^{18}O derivatives without isolation as pure solids preventing the vibrational characterization by FTIR spectroscopy. To the best of our knowledge, we present here the first vibrational study on such peroxo complexes using both resonance Raman and FTIR spectroscopy on all four possible ^{18}O isotopomers. This can be regarded as not routine. In addition, the assignment of vibrations in the FTIR spectra to μ -peroxo- Fe^{III} and μ -oxo- Fe^{III} modes will be very helpful for preparative studies where only microcrystalline powders are obtained that can be investigated with routine FTIR spectroscopy.

We have not mentioned this vibrational study in the abstract. In the introduction we have deleted the following part completely including one “thoroughly” and one “unprecedented”:

„The peroxo complex $[(\text{susan}^{6\text{-Me}})\{\text{Fe}^{\text{III}}(\mu\text{-O})(\mu\text{-}1,2\text{-O}_2)\text{Fe}^{\text{III}}\}]^{2+}$ was synthesized by the reaction of the diferrous complex $[(\text{susan}^{6\text{-Me}})\{\text{Fe}^{\text{II}}(\mu\text{-OH})_2\text{Fe}^{\text{II}}\}]^{2+}$ with O_2 and characterized thoroughly including X-ray crystallography, magnetic measurements, resonance Raman (rR) and FTIR spectroscopies. The stability even in solution at -40°C allowed an unprecedented thorough electrochemical and reactivity study,“
Neither in the results nor in the discussion section a phrase that could be claimed as a superlative has been used in the description of the vibrational characterization.

The discussion of resonance enhancement is somewhat presumptive. The concentrations used (20 mM) should be on the edge of allowing for Raman spectra to be obtained without enhancement and hence the enhancement is probably only a factor of 10 or so at most. The absence of enhancement for the other complexes is expected since the transitions are metal centered. Excitation in the UV may be more enlightening, but that of course brings the possibility of photochemistry.

We do not discuss the resonance enhancement of the resonance Raman features. Using Raman spectroscopy with the excitation wavelength 1064 nm provided no helpful spectra. We have recorded Raman spectra close to UV-Vis absorption bands to discriminate vibrations related to the $\{\text{Fe}^{\text{III}}(\mu\text{-oxo})(\mu\text{-peroxo})\text{Fe}^{\text{III}}\}$ core that are enhanced by a resonance effect from not-enhanced vibrational modes of the ligand. We have used 20 mM solutions to obtain good signal-to-noise-ratios, which is possible

because we have prepared the four $^{16}\text{O}/^{18}\text{O}$ isotopomers as solids and could dissolve them, which is not always possible for *in-situ* generated peroxy complexes. In this respect, the resonance enhancement factor is not a value for us at all, so that an enhancement factor of 10 proposed by the reviewer would be sufficient, if a discrimination between the vibrational modes associated with the $\{\text{Fe}^{\text{III}}(\mu\text{-oxo})(\mu\text{-peroxy})\text{Fe}^{\text{III}}\}$ core and those from the vibrational modes of the ligand is possible.

The transitions of the other complexes (we assume that this is related to the oxidized $\{\text{Fe}^{\text{IV}}(\mu\text{-oxo})(\mu\text{-peroxy})\text{Fe}^{\text{III}}\}$ and the protonated hydroperoxy $\{\text{Fe}^{\text{III}}(\mu\text{-oxo})(\mu\text{-hydroperoxy})\text{Fe}^{\text{III}}\}$ complexes) are assumed to be metal-centered transitions by Reviewer #2. Metal-centered transitions would be ligand field or d-d transitions. But the transitions at 14500, 16700, and 18700 cm^{-1} in the oxidized $\{\text{Fe}^{\text{IV}}(\mu\text{-oxo})(\mu\text{-peroxy})\text{Fe}^{\text{III}}\}$ complex are not d-d or ligand field transitions. The same applies to the 11800 and 18400 cm^{-1} bands for the protonated hydroperoxy $\{\text{Fe}^{\text{III}}(\mu\text{-oxo})(\mu\text{-hydroperoxy})\text{Fe}^{\text{III}}\}$ complex. As the reviewer stated, excitation in the UV region increased the possibility of photochemistry.

all in all a well worked study well described.

We are thankful to Reviewer #2 for this favorable summarizing comment on our work and the description of our work.

minor comments

in figure 5 'eq' is written 'Aq'

Corrected

trans-stilben should be trans-stilbene

Corrected

Response to Reviewer #3

The manuscript referenced NCOMMS-21-15904-T by Glaser and coworkers describes the preparation of a series of diiron (μ -oxo)(μ -peroxo/hydroperoxo) complexes of interest as potential models for diiron oxygenases and oxidation catalysts.

The characterization of the compounds is realized by combined spectroscopic analyses and is established beyond doubt. Overall the experimental work is very competently done and the conclusions appear sound. I find the DFT support confusing and unconvincing.

We are thankful to Reviewer #3 for these comments on our experimental work and conclusions. However, we are sorry that Reviewer #3 finds the DFT support confusing and unconvincing but we have tried to optimize the DFT support in the light of the reviewer's comments.

Hereafter are a few comments and questions on the experimental work.
1 - page 3: the second to last sentence concerning bond distances is not clear at all and should be rewritten.

In the light of the concerns of Reviewer #3 and Reviewer #4, we have changed from the submitted manuscript

“The core structure is asymmetric with the μ -1,2-peroxo ligand coordinated *trans* to a *tert*-amine (N2) at Fe1 and *trans* to a pyridine (N44) at Fe2. This results in a short Fe1-O^{peroxo} and a long Fe1-O^{oxo} bond, while Fe2-O^{oxo} is short and Fe2-O^{peroxo} long.”

to the following in the revised manuscript:

“Single-crystal X-ray diffraction provides an asymmetric core structure: the μ -1,2-peroxo ligand is coordinated with O1 *trans* to a *tert*-amine (N2) and with O2 *trans* to a pyridine (N44). This results in a shorter Fe1-O1^{peroxo} and a longer Fe2-O2^{peroxo} bond. The resulting different charge donation is compensated by a longer Fe1-O3^{oxo} and a shorter Fe2-O3^{oxo} bond.”

2 - pages 4 and 5: a direct link between covalency and antiferromagnetic coupling is stressed twice which is not fully obvious to me. This deserves to be explained.

The interaction between paramagnetic metal ions in polynuclear complexes has only a neglectable contribution from a magnetic dipolar interaction and the main contribution originates from the covalent interaction of the metal d orbitals with filled orbitals of the bridging ligands (superexchange mechanism). The strength of the superexchange interaction increases with the strength of the metal–bridging-ligand bonds. Especially diferric complexes are prominent example. The strengths of the Fe^{III}- μ -O bond indicated by the bond distances of ≈ 1.8 Å for μ -oxo-bridges, ≈ 1.9 Å for μ -hydroxo-bridges, and ≈ 2.0 Å for μ -aqua-bridges correlate with the approximate coupling constant J (convention $H = -2 J \mathbf{S}_1 \mathbf{S}_2$) of -100 cm⁻¹, -10 cm⁻¹, and -1 cm⁻¹, respectively. The short and strong Fe^{III}- μ -oxo bonds originates from combined stronger electrostatic and covalent contributions for the doubly negatively charged O²⁻ anion. The energy difference before interaction between the Fe^{III} d orbitals and the O²⁻ 2p orbitals is smallest for the μ -oxo donor and increases to the μ -aqua donor. This smaller energy difference before interaction results in a stronger overlap and hence a stronger mixing of the Fe^{III} d orbitals and the O²⁻ 2p orbitals hence a stronger covalency. This correlation is explained in the two references 41 and 42 provided in the submitted manuscript. A simple correlation was derived of $J \propto (\text{covalency})^2$. (T. Glaser, K. Rose, S. E. Shadle, B. Hedman, K. E. Hodgson, E. I. Solomon, *J. Am. Chem. Soc.* **2001**, *123*, 442–454) This reference is added to the manuscript.

To follow the advise of the reviewer without being too detailed, we have changed from the submitted version

“This antiferromagnetic coupling is significantly stronger than -100 ± 20 cm⁻¹ usually found for μ -oxo-diferric complexes^{41,42} including those with our dinucleating ligands.³⁷ This indicates a significant contribution of the Fe^{III}- μ -1,2-peroxo-Fe^{III} exchange pathway via the short covalent Fe^{III}- μ -peroxo bonds.”

to the revised manuscript

“This antiferromagnetic coupling is significantly stronger than -100 ± 20 cm⁻¹ usually found for μ -oxo-diferric complexes^{41,42} including those with our

dinucleating ligands.³⁷ This applies also to μ -oxo, μ -carboxylato-diferric complexes with decreased $\text{Fe}^{\text{III}}-(\mu\text{-O})\text{-Fe}^{\text{III}}$ angles as in the μ -1,2-peroxo, μ -oxo-diferric complex. This comparison indicates a significant contribution of the of the $\text{Fe}^{\text{III}}-\mu$ -1,2-peroxo- Fe^{III} exchange pathway due to shorter (1.88 and 1.93 Å) and hence more covalent $\text{Fe}^{\text{III}}-\mu$ -peroxo bonds⁴³ compared to the longer $\text{Fe}^{\text{III}}-\mu$ -carboxylato bonds (*e.g.* 1.97 and 2.07 in $[(\text{susan})\{\text{Fe}^{\text{III}}(\mu\text{-O})(\mu\text{-OAC})\text{Fe}^{\text{III}}\}]^{3+}$)⁴⁴ that are considered not to contribute significantly to the exchange coupling.”

3 - page 7, last sentence: an explanation (or at least an hypothesis) why no rR spectrum can be recorded for the oxidized compound should be provided. Is it due to photoreduction or lack of enhancement or anything else?

We understand the question of the reviewer as we really have made significant efforts to obtain resonance-enhanced Raman spectra of the oxidized complex which should be of rich information. However, the non-observability of resonance Raman spectra for complexes is not rare, but an explanation is difficult to provide and would only be highly speculative. On the one hand, there could be an intrinsic lack of enhancement. On the other hand, there could be a photochemical pathway for the excited state and photoreduction can occur for highly oxidized complexes as the oxidized μ -peroxo- $\text{Fe}^{\text{IV}}\text{Fe}^{\text{III}}$ complex.

However, as experimentalists we have to say in general that we cannot conclude from not obtaining a specific compound or a specific spectrum that the compound cannot be prepared or the spectrum cannot be obtained, respectively. Concerning resonance Raman spectra of the oxidized complex, there is still a possibility that optimized experimental conditions including optimized methodologies may offer the opportunity to obtain such a spectrum. *E. g.*, the groups of Lipscomb and Solomon tried very hard to obtain a resonance Raman spectrum of compound **Q** of sMMO without success. It was only very recently, that the group of Lipscomb obtained a resonance Raman spectrum of **Q** using time-resolved resonance Raman spectroscopy (TR3) that permits fingerprinting of intermediates through extended signal averaging for short-lived species. (R. Banerjee, Y. Proshlyakov, J. D. Lipscomb, D. A. Proshlyakov, *Nature* **2015**, *518*, 431–434) Therefore, we would still like to refrain to speculate why no resonance enhanced Raman spectrum could be obtained for the oxidized complex.

4 - page 8, last paragraph on Mössbauer: this presentation is unclear if not confusing, whereas the explanation in the SI is quite instructive. The expression "not-localized" is ambiguous: it suggests "delocalized" which is not exact in this case. I suggest that the authors rewrite this paragraph with the help of the SI to clarify the situation.

In the light of the reviewer's suggestion, we have severely rewritten this section on the Mössbauer study and the mixed valence nature of the oxidized complex from

“The 180 K Mössbauer spectrum of ^{57}Fe -labeled $[(\text{susan}^{6-\text{Me}})\{\text{Fe}(\mu\text{-O})(\mu\text{-1,2-O}_2)\text{Fe}\}]^{3+}$ (Fig. 4b) exhibits a 4-line spectrum suggesting the presence of two quadrupole doublets. Two different fit models are possible, but considerations explained in the Supplementary Information strongly favor the model with $\delta_1=0.39 \text{ mm s}^{-1}$ / $|\Delta E_Q|_1=1.29 \text{ mm s}^{-1}$ and $\delta_2=0.27 \text{ mm s}^{-1}$ / $|\Delta E_Q|_2=0.57 \text{ mm s}^{-1}$ (Table 1). Considering $\delta=0.53 \text{ mm s}^{-1}$ of the starting complex, isomer shifts of 0.39 and 0.27 mm s^{-1} imply a mainly metal-centered oxidation to the high-valent $\mu\text{-1,2-peroxo}$ complex $[(\text{susan}^{6-\text{Me}})\{\text{Fe}^{\text{IV}}(\mu\text{-O})(\mu\text{-O}_2)\text{Fe}^{\text{III}}\}]^{3+}$ with both Fe^{III} ions involved resulting in a not-localized mixed-valence $\text{Fe}^{\text{IV}}\text{Fe}^{\text{III}}$ species. These conclusions are supported by DFT calculations (Supplementary Information). The energies of the two possible options with either Fe1 or Fe2 being oxidized to Fe^{IV} revealed the configuration $\text{Fe}^{\text{IV}}1\text{Fe}^{\text{III}}2$ being $\approx 820 \text{ cm}^{-1}$ more stable. Both configurations provide computed Mössbauer parameters (Table 1) in agreement with the chosen fit model.

At lower temperatures, the Mössbauer spectra broadened (Fig. S14) due to a relaxation process that is fast relative to the Mössbauer timescale at only 180 K. The origin being electron hopping and/or paramagnetic effects is discussed in the Supplementary Information (Fig. S16). Interestingly, adding excess NEt_3 as reductant to the oxidized $[(\text{susan}^{6-\text{Me}})\{\text{Fe}^{\text{IV}}(\mu\text{-O})(\mu\text{-1,2-O}_2)\text{Fe}^{\text{III}}\}]^{3+}$ restores the Mössbauer spectrum of the starting peroxo complex $[(\text{susan}^{6-\text{Me}})\{\text{Fe}^{\text{III}}(\mu\text{-O})(\mu\text{-O}_2)\text{Fe}^{\text{III}}\}]^{2+}$ (Fig. S17) confirming the chemical reversibility. “

to the following in the revised manuscript:

“The 180 K Mössbauer spectrum of ^{57}Fe -labeled $[(\text{susan}^{6\text{-Me}})\{\text{Fe}(\mu\text{-O})(\mu\text{-1,2-O}_2)\text{Fe}\}]^{3+}$ (Fig. 4b) exhibits a 4-line spectrum suggesting the presence of two quadrupole doublets. Two different fit models are possible, but considerations explained in the Supplementary Information strongly favor the model with $\delta_1=0.39 \text{ mm s}^{-1}$ / $|\Delta E_Q|_1=1.29 \text{ mm s}^{-1}$ and $\delta_2=0.27 \text{ mm s}^{-1}$ / $|\Delta E_Q|_2=0.57 \text{ mm s}^{-1}$ (Table 1). Interestingly, adding excess NEt_3 as reductant to the oxidized $[(\text{susan}^{6\text{-Me}})\{\text{Fe}^{\text{IV}}(\mu\text{-O})(\mu\text{-1,2-O}_2)\text{Fe}^{\text{III}}\}]^{3+}$ restores the Mössbauer spectrum of the starting peroxo complex $[(\text{susan}^{6\text{-Me}})\{\text{Fe}^{\text{III}}(\mu\text{-O})(\mu\text{-O}_2)\text{Fe}^{\text{III}}\}]^{2+}$ (Fig. S16) confirming the chemical reversibility.

The decrease of the isomer shift from 0.53 mm s^{-1} of the starting complex to 0.27 and 0.39 mm s^{-1} confirms a mainly metal-centered oxidation to a high-valent $\mu\text{-1,2}$ -peroxo complex with both Fe^{III} ions involved resulting in a mixed-valence $\text{Fe}^{\text{IV}}\text{Fe}^{\text{III}}$ species. In the Robin-and-Day classification for mixed-valence systems,⁴⁹ class I implies no interaction (ruled out by the coupled $S_i=1/2$ spin ground state) while class III stands for a quantum-mechanically delocalized state. In class II systems exist two different states that correspond roughly to the excess electron localized on the one or the other metal ion. Between these two states is an energy barrier and there can be temperature-dependent and light-induced mechanisms to transfer the excess electron from the reduced to the oxidized metal ion. In symmetric cases, the two states are energetically degenerate while the asymmetry observed here results in an energy difference between these states (Fig. S17). The two quadrupole doublets can arise from one of these states populated exclusively up to 180 K or from an electron hopping between these two states at a rate faster than the Mössbauer timescale (10^{-7} s). Unfortunately, the Mössbauer spectra broadened at lower temperatures (Fig. S15) due to a relaxation process that is fast relative to the Mössbauer timescale at only 180 K. However, the origin cannot only arise from a decrease of the electron hopping rate but also from paramagnetic effects (Fig. S15). DFT calculations (Supplementary Information) provided two localized configurations $\text{Fe}^{\text{IV}}1\text{Fe}^{\text{III}}2$ and $\text{Fe}^{\text{III}}1\text{Fe}^{\text{IV}}2$ that both reproduce the

isomer shift decrease of both iron ions (Table 1) and hence confirm the assignment to class II. However, although these DFT calculations provided $\text{Fe}^{\text{IV}}1\text{Fe}^{\text{III}}2$ being lower in energy by $\approx 820 \text{ cm}^{-1}$, more advanced MO calculations that are beyond this study are required to obtain further insight.”

5 - page 9: in the expression "a job plot analysis" job must be written with a capital J. As a matter of fact, Paul Job was an analytical chemist of the XXth century who introduced this plot, and there is no relationship with the labor market.

Corrected.

6 - page 10, Figure 5: it is stated that the peroxo $\rightarrow \text{Fe}^{\text{III}}$ LMCT "disappears". Figure 5 shows that the oxo $\rightarrow \text{Fe}^{\text{III}}$ LMCT @ 19000 cm^{-1} appears "doubled". Accordingly, is it possible that the peroxo $\rightarrow \text{Fe}^{\text{III}}$ LMCT does not disappear, but moves to 18000 cm^{-1} ? The authors should discuss and possibly eliminate this possibility before concluding.

The wording “The disappearance of the $\mu\text{-}1,2\text{-peroxo} \rightarrow \text{Fe}^{\text{III}}$ LMCT at 15400 cm^{-1} ” simply describes the observation. The combined data clearly indicate that the protonation results in a $\mu\text{-}1,2\text{-hydroperoxo}, \mu\text{-oxo}$ complex. Thus, the $\mu\text{-}1,2\text{-peroxo} \rightarrow \text{Fe}^{\text{III}}$ LMCT transition cannot shift due to the absence of a peroxo ligand in the protonated complex. However, the resulting $\mu\text{-}1,2\text{-hydroperoxo}$ bridge could provide a $\mu\text{-}1,2\text{-hydroperoxo} \rightarrow \text{Fe}^{\text{III}}$ LMCT, that should be at higher energy due to the energetic stabilization of the oxygen p orbitals upon protonation with lower intensity due to a reduced covalency.

Therefore, the absorption feature between 17000 and 20000 cm^{-1} in the protonated complex with two resolved maxima could be due to a combination of a $\mu\text{-oxo} \rightarrow \text{Fe}^{\text{III}}$ LMCT and a $\mu\text{-}1,2\text{-hydroperoxo} \rightarrow \text{Fe}^{\text{III}}$ LMCT. On the other hand, the combined absorption features (“doubled” transition at $17000\text{-}20000 \text{ cm}^{-1}$ and the 11800 cm^{-1} transition) can also be assigned to a $\mu\text{-oxo}$ -bridged species of the same $\text{Fe}^{\text{III}}\text{-}\mu\text{-oxo}\text{-Fe}^{\text{III}}$ angle of 123° as in the parent complex but with two more strongly differing Fe^{III} sites in accordance with the observation of two quadrupole doublets in the Mössbauer spectrum. Unfortunately, resonance Raman spectroscopy was not helpful to differentiate between these two possibilities. As it is beyond the scope of this study to

rigorously assign the Vis-absorption features of the protonated μ -1,2-hydroperoxo, μ -oxo Fe^{III}Fe^{III} complex, we did not provide an assignment upon submission of the initial manuscript and we would like to refrain from an assignment also in this revised manuscript.

Comments on the DFT calculations

7 - page 8: For the oxidized complex, two configurations are envisaged and one is preferred on energetic grounds, the difference being 820 cm⁻¹. The two configurations cannot be distinguished from the calculated Mössbauer parameters. It is stated that the calculated Mössbauer parameters of both configurations match the experimental ones.

page 11: For the protonated complex, in spite of the fact that protonation at O1 is favored by DFT calculations by 1200 cm⁻¹, the authors prefer protonation at O2 because the calculated Mössbauer parameters match better the experimental ones.

We are sorry but there is a real misunderstanding of Reviewer #3. The protonation at O1 is *disfavoured* by the DFT calculation by 1200 cm⁻¹ and not as Reviewer 3# states is favored. The statement in the manuscript is

“Geometry optimizations provided three different tautomers with protonation at the μ -peroxo-O1 being 1200 cm⁻¹ higher in energy.”

8 - Concerning the oxidized compound, it is probably a class II mixed-valent species (stated in SI) or in other words it exhibits temperature-dependent partial residency of the extra electron on the two sites. How was this feature taken into account in the calculations? Could it be taken into account in the framework of DFT calculations or did it require CASSCF calculations? The ultimate question may be: is the computational approach adapted to the problem?

In a class II mixed-valence system there are two states that might be energetically degenerate for a symmetric situation or - as it is the case here - with two different energies due to the asymmetry of the two coordination sites. Our intention using DFT calculations was to evaluate whether one or two of these two states corresponding formally to Fe^{IV}Fe^{III} and Fe^{III}Fe^{IV} can be converged and whether the calculated Mössbauer parameters would provide further insight. In this respect, we were quite surprised how well these calculations reproduced the Mössbauer parameters.

Concerning the energy difference of 820 cm^{-1} (corresponding to 2.3 kcal mol^{-1}), it was not our intention to indicate that this small calculated energy difference corresponds to the correct energy difference as for such an assignment more sophisticated *ab initio* MO calculations would be necessary. We have explained this in more in the revised Supplementary Information

“The final single-point energy of $\text{Fe}^{\text{IV}}\text{Fe}^{\text{III}}_2$ was found to be 820 cm^{-1} ($2.34\text{ kcal mol}^{-1}$) lower in energy, which is in the error range of these calculations. These results indicate a localized $\text{Fe}^{\text{IV}}\text{Fe}^{\text{III}}$ class II description. However, the energy difference between the two configurations only reflect the intrinsic difference between the two coordination sites. In this respect it favors an energetic preference for one configuration without significant thermal population up to room temperature, while the level of theory is not sufficient to assign the lower energy configuration to the $\text{Fe}^{\text{IV}}\text{Fe}^{\text{III}}_2$ configuration despite it was found to be lower by these DFT calculations.”

and in the revised manuscript:

“However, although these DFT calculations provided $\text{Fe}^{\text{IV}}\text{Fe}^{\text{III}}_2$ being lower in energy by $\approx 820\text{ cm}^{-1}$, more advanced MO calculations that are beyond this study are required to obtain further insight.”

To evaluate the mixed valence properties of this new asymmetric class II mixed-valence $\text{Fe}^{\text{IV}}\text{Fe}^{\text{III}}$ complex require much more elaborated *ab initio* calculations, which are not only beyond the scope of this manuscript but also beyond our expertise.

9 - General comment on the DFT calculations. Their presentation is not sufficiently detailed to warrant the conclusions. These open-shell systems are intrinsically tough to calculate and one must be very cautious and provide valuable information on the way the calculations were conducted and eventually controlled to convince the reader. In particular, the authors must indicate how they proceeded to carry out broken-symmetry calculations. In addition, for the geometry optimizations they must indicate how they imposed the localization of the electron hole and provide the spin density distribution of the

optimized structure. Same analysis for the protonation site. Finally, it is usual practice to communicate in the SI the cartesian coordinates of the optimized geometries.

Based on these concerns of Reviewer #3, we have **completely rewritten and rearranged the part on the DFT calculations in the Supplementary Information**. Specifically, we have - as demanded - provided more valuable informations on the way the calculations were conducted, how we carried out the broken-symmetry calculations, and how we imposed the localization of the electron hole and we provided the spin density distributions of the optimized structures and the cartesian coordinates of the optimized geometries.

To summarize, the results presented are new and significant and deserve publication. However, their presentation must be improved, in particular by moving key arguments from the SI to the text. In addition, the computational procedures must be detailed to prove the validity of the calculations. I recommend publication once these points have been implemented.

In summary, we are thankful to Reviewer #3 as the revisions based on the comments and concerns of Reviewer #3 significantly improved our manuscript.

Response to Reviewer #4

The paper deals with a highly relevant topic, namely the enzymatic activation of dioxygen, and identifies experimentally various possible intermediates along the catalytic process. The main results are also backed up or further studied with theoretical calculations. Overall, the paper is written in very precise language, which is, on the one hand, a positive aspect, but also leads to the paper being very technical. It is extremely hard, if not impossible, to really “read” the paper. For the results part, this might be justified, but for the discussion part, the language should probably be less technical and more suited for a general readership.

We are thankful to this Reviewer for the positive comments. However, this Reviewer also criticized our language as too technical. While the Reviewer might be completely right with this criticism, on the other hand it is really difficult for a non-native English speaker to control the border between too technical and not adequate for a scientific publication.

Nevertheless, in the light of this concern of Reviewer #4, we have tried to revise our manuscript with respect to a less technical language:

We changed from

“The reversible oxidation of the μ -1,2-peroxo-diferric complex provides an unprecedented μ -1,2-peroxo $\text{Fe}^{\text{IV}}\text{Fe}^{\text{III}}$ complex and the reversible protonation an also unprecedented μ -1,2-hydroperoxo-diferric species.”

to the following in the revised manuscript:

“The reversible oxidation and protonation of the μ -1,2-peroxo-diferric complex provide unprecedented μ -1,2-peroxo $\text{Fe}^{\text{IV}}\text{Fe}^{\text{III}}$ and μ -1,2-hydroperoxo-diferric species, respectively.”

We changed from

“The core structure is asymmetric with the μ -1,2-peroxo ligand coordinated *trans* to a *tert*-amine (N2) at Fe1 and *trans* to a pyridine (N44) at Fe2. This results in a short $\text{Fe1-O}^{\text{peroxo}}$ and a long $\text{Fe1-O}^{\text{oxo}}$ bond, while $\text{Fe2-O}^{\text{oxo}}$ is short and $\text{Fe2-O}^{\text{peroxo}}$ long.”

to the following in the revised manuscript:

“Single-crystal X-ray diffraction provides an asymmetric core structure: the μ -1,2-peroxo ligand is coordinated with O1 *trans* to a *tert*-amine (N2) and with O2 *trans* to a pyridine (N44). This results in a shorter Fe1-O1^{peroxo} and a longer Fe2-O2^{peroxo} bond. The resulting different charge donation is compensated by a longer Fe1-O3^{oxo} and a shorter Fe2-O3^{oxo} bond.”

We changed from

“The complex [(susan){Fe^{III}(OH)(μ -O)Fe^{III}(OH)}]²⁺ catalyzes the oxidation of CH₃OH with H₂O₂ to HCHO.⁵⁸ We could observe spectroscopically the μ -1,2-peroxo intermediate [(susan){Fe^{III}(μ -O)(μ -1,2-O₂)Fe^{III}}]²⁺, that converts to a high-valent active species.”

to the following in the revised manuscript:

“The complex with hydroxides, [(susan){Fe^{III}(OH)(μ -O)Fe^{III}(OH)}]²⁺, catalyzes the oxidation of CH₃OH with H₂O₂ to HCHO.⁵⁸ We could observe the μ -1,2-peroxo intermediate [(susan){Fe^{III}(μ -O)(μ -1,2-O₂)Fe^{III}}]²⁺. However, the temperature-dependencies ruled out this μ -1,2-peroxo intermediate to be the active species indicating the conversion to a high-valent active species.”

We changed from

“Moreover, the steric repulsion of the 6-methyl group results in longer Fe-N^{6-Me-py} than Fe-N^{py} bonds.^{59–65} The resulting lower electron donation results in an anodic shift of +250 mV making Fe^{IV} less accessible with susan^{6-Me} than with susan.”

to the following in the revised manuscript:

“Moreover, the steric repulsion of the 6-methyl group enforces longer Fe-N^{6-Me-py} than Fe-N^{py} bonds,^{59–65} and hence a lower electron donation. This results in an anodic shift of +250 mV making Fe^{IV} less accessible with susan^{6-Me} than with susan.”

We changed from

“The μ -1,2-peroxo complex {Fe^{III}(μ -O)(μ -1,2-O₂)Fe^{III}} of susan is a reactive intermediate that decays *via* a high-valent Fe^{IV} species to its thermodynamic sink {Fe^{III}X(μ -O)Fe^{III}X}. Since with susan^{6-Me} this thermodynamic driving

force is absent and Fe^{IV} less accessible, the peroxo complex should be stabilized. Indeed, we could isolate the stable μ -1,2-peroxo complex [(susan^{6-Me}){Fe^{III}(μ -O)(μ -1,2-O₂)Fe^{III}}]²⁺. It is even further stabilized by a better encapsulation of the μ -1,2-peroxo ligand by the 6-methyl group of susan^{6-Me} (left in Fig. 6b) that would be absent with susan.”

to the following in the revised manuscript:

“In this respect, we thought that susan^{6-Me} should be able to stabilize a μ -1,2-peroxo complex {Fe^{III}(μ -O)(μ -1,2-O₂)Fe^{III}} that is with susan only a reactive intermediate decaying *via* a high-valent Fe^{IV} species to its thermodynamic sink {Fe^{III}X(μ -O)Fe^{III}X}. In contrast, with susan^{6-Me} not only this thermodynamic driving force is absent but Fe^{IV} is also less accessible. Indeed, we could isolate the stable μ -1,2-peroxo complex [(susan^{6-Me}){Fe^{III}(μ -O)(μ -1,2-O₂)Fe^{III}}]²⁺. Inspection of the space-filling model shows that the μ -1,2-peroxo ligand is even further stabilized by a better encapsulation with the 6-methyl group of susan^{6-Me} (left 1,2-peroxo-oxygen atom in Fig. 6b) that would be absent with susan.”

We changed from

“The principal accessibility of Fe^{IV} with susan^{6-Me} is demonstrated by the reversible oxidation to [(susan^{6-Me}){Fe^{IV}(μ -O)(μ -1,2-O₂)Fe^{III}}]³⁺, which is stabilized by the additional highly covalent μ -1,2-peroxo ligand and formally contains one oxidation-equivalent more than intermediate **Q**.”

to the following in the revised manuscript:

“Although, susan^{6-Me} is less suited for stabilization of Fe^{IV} than susan, the principal accessibility of Fe^{IV} with susan^{6-Me} is demonstrated by the reversible oxidation to [(susan^{6-Me}){Fe^{IV}(μ -O)(μ -1,2-O₂)Fe^{III}}]³⁺, which is stabilized by the additional highly covalent μ -1,2-peroxo ligand. It is interesting to note, that this high-valent μ -1,2-peroxo species stores one oxidation-equivalent more than intermediate **Q**.”

We changed from

“The nucleophilic character of a ligand should increase with less electron donation to the Fe^{III} ions, *i. e.* less covalent bonding and longer Fe-O-bonds.

But for **A**, the Fe^{III}- μ -O^{oxo} bonds are shorter than for [(susan^{6-Me}){Fe^{III}(μ -O)(μ -1,2-O₂)Fe^{III}}]²⁺ (1.72/1.74 Å vs 1.82/1.89 Å), whereas for the Fe^{III}- μ -O^{peroxo} bonds the situation is reversed (2.07/2.10 Å vs 1.88/1.93 Å). Both trends are in contrast to the experiment. However, the disorder of the μ -oxo/ μ -1,2-peroxo ligands in **A** questions the significance of this comparison, and as protonation should occur at a p π orbital and a Fe-O bond consists of σ - and π -bonding, a pure structural analysis does not need to provide the answer for the reactivity.”

to the following in the revised manuscript:

“The nucleophilic character of a ligand should increase with less electron donation to the Fe^{III} ions, *i. e.* less covalent longer bonds. But for **A**, the Fe^{III}- μ -O^{oxo} bonds are shorter than for [(susan^{6-Me}){Fe^{III}(μ -O)(μ -1,2-O₂)Fe^{III}}]²⁺ (1.72/1.74 Å vs 1.82/1.89 Å), whereas the situation is reversed for the Fe^{III}- μ -O^{peroxo} bonds (2.07/2.10 Å vs 1.88/1.93 Å). This structural argumentation is in contrast to the experimentally determined protonation sites. However, the disorder of the μ -oxo/ μ -1,2-peroxo ligands in **A** questions the significance of this comparison. Moreover, as protonation should occur at a p π orbital and a Fe-O bond consists of σ - and π -bonding, a pure structural analysis does not need to provide the answer for the reactivity.”

We changed from

“The strongest difference is observed for the ν_s (Fe-O₂-Fe), which are at 465 and 448 cm⁻¹ for **A** and [(susan^{6-Me}){Fe^{III}(μ -O)(μ -1,2-O₂)Fe^{III}}]²⁺, respectively, indicative for a higher nucleophilicity of the peroxo in [(susan^{6-Me}){Fe^{III}(μ -O)(μ -1,2-O₂)Fe^{III}}]²⁺.”

to the following in the revised manuscript:

“Interestingly, the strongest difference is observed for the ν_s (Fe-O₂-Fe), which are at 465 and 448 cm⁻¹ for **A** and [(susan^{6-Me}){Fe^{III}(μ -O)(μ -1,2-O₂)Fe^{III}}]²⁺, respectively, indicative for less covalent Fe^{III}- μ -O^{peroxo} bonds and hence a higher nucleophilicity of the peroxo in [(susan^{6-Me}){Fe^{III}(μ -O)(μ -1,2-O₂)Fe^{III}}]²⁺.”

Moreover, we have changed the wording slightly on several instances (highlighted in

yellow), where the presentation here seems not appropriate.

I have only a few more detailed comments and suggestions:

1) Introduction: There are a few intermediate structures highlighted explicitly in the introduction with labels P, P' and Q. It would be very helpful to visualize them so that at the outset of the paper it is clear on the first glance what the overall fundament of the paper is. A sentence such as “the peroxo intermediate P converts to a high-valent active species Q” is not very helpful as the general reader has no idea what is hidden behind the labels P and Q. [As discussed above, the technical focus of the paper makes it very difficult to digest for a general reader.]

We agree with Reviewer #4 that a sketch will be helpful for the not-specialized reader to follow the introduction and have therefore included a sketch visualizing the key intermediate structures referred to in the introduction: **P**, **Q**, as well as μ -1,2-hydroperoxo, μ -1,1-hydroperoxo, and μ -1,1-peroxo diferric cores proposed for **P'**.

2) On page 3, in the introduction, there are statements referring to past work without citations. “allowed an unprecedented ... reactivity study” and “was observed by X-ray crystallography” need a citation each to these individual studies.

The whole paragraph on page 3 starting from “Here, we present” to “for diiron enzymes” describes the work we present here in this manuscript, hence no citation to previous work is missing.

In the light of the reviewer’s comment and the GUIDE TO FORMATTING ARTICLES, we have revised this last paragraph of the introduction from

“Here, we present the rational stabilization of a μ -1,2-peroxo complex using the dinucleating ligand susan^{6-Me}.^{36–38} The peroxo complex [(susan^{6-Me}){Fe^{III}(μ -O)(μ -1,2-O₂)Fe^{III}}]²⁺ was synthesized by the reaction of the diferrous complex [(susan^{6-Me}){Fe^{II}(μ -OH)₂Fe^{II}}]²⁺ with O₂ and characterized thoroughly including X-ray crystallography, magnetic measurements, resonance Raman (rR) and FTIR spectroscopies. The stability even in

solution at -40°C allowed an unprecedented thorough electrochemical and reactivity study, establishing nucleophilic and electrophilic character of the μ -1,2-peroxo bridge, attenuated for exogenous organic substrates by encapsulation of the ligand scaffold. Instead, the intramolecular C-H activation of preorganized 6-methyl pyridine groups providing benzylalcoholato and carboxylato donors was observed by X-ray crystallography. $[(\text{susan}^{6\text{-Me}})\{\text{Fe}^{\text{III}}(\mu\text{-O})(\mu\text{-O}_2)\text{Fe}^{\text{III}}\}]^{2+}$ can be reversibly oxidized to the first high-valent μ -1,2-peroxo complex $[(\text{susan}^{6\text{-Me}})\{\text{Fe}^{\text{IV}}(\mu\text{-O})(\mu\text{-1,2-O}_2)\text{Fe}^{\text{III}}\}]^{3+}$ and reversibly protonated to the first μ -1,2-hydroperoxo complex $[(\text{susan}^{6\text{-Me}})\{\text{Fe}^{\text{III}}(\mu\text{-O})(\mu\text{-1,2-OOH})\text{Fe}^{\text{III}}\}]^{3+}$. This first μ -1,2-hydroperoxo-diferric model and its spectroscopic signatures should provide insight for the assignments of reactive intermediates postulated for diiron enzymes.”

to

“Here, we present the synthesis, characterization, and reactivity of the rationally stabilized μ -1,2-peroxo complex $[(\text{susan}^{6\text{-Me}})\{\text{Fe}^{\text{III}}(\mu\text{-O})(\mu\text{-1,2-O}_2)\text{Fe}^{\text{III}}\}](\text{ClO}_4)_2$ using the dinucleating ligand $\text{susan}^{6\text{-Me}}$.^{37–39} This μ -1,2-peroxo complex is stable even in solution at -40°C and shows nucleophilic character of the μ -1,2-peroxo ligand attenuated for exogenous organic substrates by encapsulation of the ligand scaffold. $[(\text{susan}^{6\text{-Me}})\{\text{Fe}^{\text{III}}(\mu\text{-O})(\mu\text{-O}_2)\text{Fe}^{\text{III}}\}]^{2+}$ is reversibly oxidized to the first high-valent μ -1,2-peroxo complex $[(\text{susan}^{6\text{-Me}})\{\text{Fe}^{\text{IV}}(\mu\text{-O})(\mu\text{-1,2-O}_2)\text{Fe}^{\text{III}}\}]^{3+}$ and reversibly protonated to the first μ -1,2-hydroperoxo complex $[(\text{susan}^{6\text{-Me}})\{\text{Fe}^{\text{III}}(\mu\text{-O})(\mu\text{-1,2-OOH})\text{Fe}^{\text{III}}\}]^{3+}$. The study of the electrophilic reactivity for oxygen-atom transfer (OAT) using PPh_3 and hydrogen-atom transfer (HAT) using DHA and TEMPOH provides not only a low electrophilic character of the parent μ -1,2-peroxo- $\text{Fe}^{\text{III}}\text{Fe}^{\text{III}}$ complex but also for the oxidized 1,2-peroxo- $\text{Fe}^{\text{IV}}\text{Fe}^{\text{III}}$ and protonated 1,2-hydroperoxo- $\text{Fe}^{\text{III}}\text{Fe}^{\text{III}}$ species. Only the oxidized 1,2-peroxo- $\text{Fe}^{\text{IV}}\text{Fe}^{\text{III}}$ species reacts with the relatively weak substrate TEMPOH. The determination of the $pK_a=9.5\pm 0.1$ and the bond dissociation free energy $\text{BDFE}(\text{OH})_{\text{CH}_3\text{CN}} = 78\pm 2 \text{ kcal mol}^{-1}$ of the protonated 1,2-hydroperoxo-

Fe^{III}Fe^{III} species quantifies the low electrophilic character even of the oxidized 1,2-peroxo-Fe^{IV}Fe^{III} species. Therefore, the intramolecular C-H activation of preorganized 6-methyl pyridine groups to benzylalcoholato and carboxylato donors in the parent 1,2-peroxo-Fe^{III}Fe^{III} complex should not occur *via* the 1,2-peroxo-ligand but *via* conversion to a more reactive but fluent high-valent species. The low electrophilic character and the spectroscopic signatures of this first μ -1,2-hydroperoxo-diferric model is discussed in relation to assignments of reactive intermediates postulated for diiron enzymes.”

3) Page 3, last few sentences. The structural parameters discussed have been obtained from single-crystal X-ray diffraction? This should be clarified.

In the light of the concerns of Reviewer #4 and Reviewer #3, we have changed from the submitted manuscript

“The core structure is asymmetric with the μ -1,2-peroxo ligand coordinated *trans* to a *tert*-amine (N2) at Fe1 and *trans* to a pyridine (N44) at Fe2. This results in a short Fe1-O^{peroxo} and a long Fe1-O^{oxo} bond, while Fe2-O^{oxo} is short and Fe2-O^{peroxo} long. The O-O bond length of 1.432(2) Å is the longest yet established for a peroxo-diferric complex (1.396–1.426 Å).^{22–26}”

to the following in the revised manuscript:

“Single-crystal X-ray diffraction provides an asymmetric core structure: the μ -1,2-peroxo ligand is coordinated with O1 *trans* to a *tert*-amine (N2) and with O2 *trans* to a pyridine (N44). This results in a shorter Fe1-O1^{peroxo} and a longer Fe2-O2^{peroxo} bond. The resulting different charge donation is compensated by a longer Fe1-O3^{oxo} and a shorter Fe2-O3^{oxo} bond. The O1-O2 bond length of 1.432(2) Å is the longest yet established for a peroxo-diferric complex (1.396–1.426 Å).^{22–26}”

4) Page 5: “We synthesized all four possible isotopomers as solids.”

Crystalline powder? Amorphous?

In the light of the reviewer's concern, we have changed from the submitted manuscript

“We synthesized all four possible $^{18}\text{O}/^{18}\text{O}_2$ -isotopomers as solids.”

to the following in the revised manuscript:

“We synthesized all four possible $^{18}\text{O}/^{18}\text{O}_2$ -isotopomers as microcrystalline solids.”

5) According to the introduction and the title, one of the most exciting findings of this study is the protonated peroxy form, discussed on pages 9-11. But very little is said about the chemistry. How was it synthesized? “Treatment of xx with yy” is not very helpful. Was it in solution? At room temperature? Open to air? Is the final compound a solid, or does it exist only in solution? Or only observed in situ? How was it isolated and purified? Only in the discussion part, the authors mention “the high stability” of the compound. What does this mean, and relative to what?

We cannot really follow the criticism of Reviewer #4 on the whole point. To write a manuscript on a study with so many results for the general readership of the journal *Nature Communications* demands many compromises. Due to space restrictions and for the readability, too many technical details that can be regarded as routine should not be included in the main text. On page 8 of the submitted manuscript, we wrote “treatment of $[(\text{susa}^{6-\text{Me}})\{\text{Fe}^{\text{III}}(\mu\text{-O})(\mu\text{-}1,2\text{-O}_2)\text{Fe}^{\text{III}}\}]^{2+}$ with HClO_4 at -60°C ...” clearly states that it is not at room temperature but at -60°C . Moreover, for a molecular chemist it seems to be trivial to say that the reaction of molecules is performed in solution. However, there is not only the main text, which should be, as mentioned above, as readable as possible, but there are two more places where the experimental conditions are described in more technical detail. On the one hand, this is the figure caption of Fig. 5a+b referred to in the main text of the manuscript. In the figure caption of Fig. 5 it is written, that this reaction was performed in $\text{CH}_3\text{CN}/\text{CH}_2\text{Cl}_2$ (1:2) at -60°C . For more detailed technical description of the reaction, there is still the experimental section in the supplementary information. Moreover, the handling of solutions at temperatures below 0°C requires the exclusion of air to prevent forming of ice from atmospheric water in low-temperature solution chemistry as in single-crystal X-ray diffraction. It

would be a very bad experimental handling to have a -60°C solution open to air. In turn, the explicit stating of the exclusion of air for handling a -60°C solution would imply that we want to stress how well we performed the reaction but this seems not appropriate as it is really more than routine.

The statement “Treatment of [(susan^{6-Me}){Fe^{III}(μ-O)(μ-1,2-O₂)Fe^{III}}]²⁺ with HClO₄ at -60°C resulted in the loss of the 15400 cm⁻¹ band while the 11800 and 19300 cm⁻¹ bands only slightly shifted (Fig. 5a+b).” including the figure caption of Fig. 5 clearly indicates, that the solution of the reactants were observed directly by UV-Vis spectrometry.

In order to reduce possible misunderstandings without being too technical in the main text, we have changed the first sentence from:

“Treatment of [(susan^{6-Me}){Fe^{III}(μ-O)(μ-1,2-O₂)Fe^{III}}]²⁺ with HClO₄ at -60°C resulted in the loss of the 15400 cm⁻¹ band while the 11800 and 19300 cm⁻¹ bands only slightly changed (Fig. 5a+b).”

to the following in the revised manuscript:

“Treatment of a solution of [(susan^{6-Me}){Fe^{III}(μ-O)(μ-1,2-O₂)Fe^{III}}]²⁺ with HClO₄ at -60°C resulted in the loss of the 15400 cm⁻¹ band while the 11800 and 19300 cm⁻¹ bands only slightly changed (Fig. 5a+b).”

We wrote in the discussion on page 12-13 of the submitted manuscript:

“Generally, protonation of a Fe-coordinated peroxy ligand is regarded to enhance its reactivity, e. g. protonation of the *cis*-μ-1,2-peroxy intermediate **P** of MMO was proposed to facilitate the conversion to intermediate **Q**.¹¹ The high stability of the μ-1,2-hydroperoxy complex [(susan^{6-Me}){Fe^{III}(μ-O)(μ-1,2-OOH)Fe^{III}}]³⁺ (no decay observed at -60°C, τ_{1/2}≈11 min at -40°C) is thus remarkable and must owe its origin to a low stabilization of the Fe^{IV} conversion product by susan^{6-Me}.“

Additionally, we wrote in the introduction that most μ-1,2-peroxy diferric species could only be identified spectroscopically as transient intermediates. This provides information on the low stability of peroxy diferric complexes in general and we stress in the citation above that protonation of such a peroxy diferric complex even enhances its reactivity. As we have stated several times, the μ-hydroperoxy diferric complex

described here is the first μ -hydroperoxo diferric complex ever obtained because of the low stability of μ -1,2-peroxo complexes which become even less stable upon protonation. Our statement “high stability” originates from the observation of no decay at -60°C . However, by stating that the complex is only stable with a half-life of 11 min at -40°C indicates that it is a reactive intermediate generated and characterized *in situ*. In order to prevent a possible misunderstanding of high stability meaning that the compound is indefinitely stable at room temperature and can be obtained as a solid, we have changed from the following in the submitted manuscript:

“The high stability of the μ -1,2-hydroperoxo complex $[(\text{susan}^{6-\text{Me}})\{\text{Fe}^{\text{III}}(\mu\text{-O})(\mu\text{-1,2-OOH})\text{Fe}^{\text{III}}\}]^{3+}$ (no decay observed at -60°C , $\tau_{1/2}\approx 11$ min at -40°C) is thus remarkable and must owe its origin to a low stabilization of the Fe^{IV} conversion product by $\text{susan}^{6-\text{Me}}$.”

to the following in the revised manuscript:

“The relatively high stability of the μ -1,2-hydroperoxo complex $[(\text{susan}^{6-\text{Me}})\{\text{Fe}^{\text{III}}(\mu\text{-O})(\mu\text{-1,2-OOH})\text{Fe}^{\text{III}}\}]^{3+}$ (no decay observed at -60°C , $\tau_{1/2}\approx 11$ min at -40°C) is thus remarkable and must owe its origin to a low stabilization of the Fe^{IV} conversion product by $\text{susan}^{6-\text{Me}}$.”

6) The crystal structure determination and refinement have been done accurately and expertly. However, I think it is very important to show all three structures with atomic displacement ellipsoids in Figure S4 in the SI. Especially the most complicated one is left out. If the model with all disorder parts is too difficult to visualize clearly, maybe only the major disorder components could be shown. But the sketch in Figure 6 as a ball-and-stick model cannot replace a representation of a model in the SI that includes the (possible problems with the) ADPs of the atoms.

We are thankful to Reviewer #4 for this comment as we simply forgot to include the thermal ellipsoid plots of the decay product. We have added the thermal ellipsoid plots of the decay product in Fig. S1.

REVIEWER COMMENTS

Reviewer #1 (Remarks to the Author):

The authors have specifically addressed all the questions I raised in my original review.

- Clarifications regarding some of the spectroscopic analyses are satisfactory.
- The HAT and OAT reactivity of the diferric-peroxide, the diferric hydroperoxide and the high valent dimer have been studied against common substrates (DHA, TEMPOH and PPh₃) and the nucleophilic nature of the reaction with an aldehyde has been clarified. The BDE of the O-H bond in the hydroperoxide species has been also nicely determined. I find surprising that the energy is relatively low, which explains its modest reactivity. It is interesting the consequence of the weak nature of this bond when considering the intramolecular C-H oxidation, indicating that the compound needs to convert into a more powerful oxidant. This is a proposal that has been suggested for explaining the activity of several O₂ activating enzymes, which accumulate rather unreactive peroxide intermediates.
- The apparent electron transfer reactivity of the high valent compound when reacting with PPh₃ is surprising but there are precedents.
- What I find more surprising and hard to understand is the apparent acidic nature of the hydroperoxide, which manifests when reacting with PPh₃ and also with TEMPO. According to ref 54 and 60, the pKa of PPh₃ and TEMPOH are 7.62 and around 7, respectively, while the pKa determined for the hydroperoxide is 9.5. Then, there should be no protonation of the phosphine or TEMPO if the latter is correct. This needs to be clarified since it brings doubts about the pKa value of the hydroperoxide and consequently about the O-H DBE. I am not sure the method employed to determine the pKa, using different acids, provides a reliable value. Furthermore, the pKa values of the acids used to make the plot do not seem correct, for example, methanesulphonic, trichloroacetic and trifluoroacetic have pKa values <1. May I be missing something?

Reviewer #2 (Remarks to the Author):

the authors have addressed all issues raised

the reference to Toftlund is:

Synthesis and Magnetic Properties of a μ -Oxo-di(μ -acetato)manganese(III) Complex of a Strapped Tripodal Pyridylamine Ligand N,N,N',N'-Tetrakis(2-pyridylmethyl)-1,3-propanediamine. A Model for the Mn₂ Site of Mn-Catalase Enzymes.

Toftlund, Hans; Markiewicz, Andrew; Murray, Keith S.

Pages: 443-446.

and Armstrong

Gamelin, D. R.; Kirk, M. L.; Solomon, E. I.; Stemmler, T. L.; Penner-Hahn, J. E.;

Pal, S.; Armstrong, W. H. *J. Am. Chem. Soc.* 1994, 116, 2392–2399.

Reviewer #3 (Remarks to the Author):

The authors have made a very good job in revising their manuscript thoroughly and extensively. They have fully clarified the obscure points in the original version and this should make following the work easy for any reader.

Eventually, the paper provides a substantiated and rationalized comparison of three biologically relevant forms of iron centers involved in important oxidative transformations. As such it is likely to be cited as a reference in the field.

I recommend publication.

Reviewer #4 (Remarks to the Author):

I have reviewed the revised version of the manuscript by Glaser et al. Still, it needs lots of concentration to understand the technical details described in the main text, however, I find everything to be correct and precisely described, and I acknowledge that the authors have made an effort to improve the readability.

My main concern in my original review was the crystal structure of the decay product depicted in a ball-and-stick model in Figure 6b. This structure has now been shown in Figure S1 c with both disorder components separately. This is fine and indicates no further problems. I would only suggest to refer to Figure S1 in the caption of Figure 6.

Otherwise, I believe that the manuscript is ready for publication.

Response to Reviewer #1

The authors have specifically addressed all the questions I raised in my original review.

- Clarifications regarding some of the spectroscopic analyses are satisfactory.

- The HAT and OAT reactivity of the diferric-peroxide, the diferric hydroperoxide and the high valent dimer have been studied against common substrates (DHA, TEMPOH and PPh₃) and the nucleophilic nature of the reaction with an aldehyde has been clarified. The BDE of the O-H bond in the hydroperoxide species has been also nicely determined. I find surprising that the energy is relatively low, which explains its modest reactivity. It is interesting the consequence of the weak nature of this bond when considering the intramolecular C-H oxidation, indicating that the compound needs to convert into a more powerful oxidant. This is a proposal that has been suggested for explaining the activity of several O₂ activating enzymes, which accumulate rather unreactive peroxide intermediates.

- The apparent electron transfer reactivity of the high valent compound when reacting with PPh₃ is surprising but there are precedents.

We are thankful for the above favorable comments of Reviewer #1 on our revisions of the manuscript.

- What I find more surprising and hard to understand is the apparent acidic nature of the hydroperoxide, which manifests when reacting with PPh₃ and also with TEMPO. According to ref 54 and 60, the pK_a of PPh₃ and TEMPOH are 7.62 and around 7, respectively, while the pK_a determined for the hydroperoxide is 9.5. Then, there should be no protonation of the phosphine or TEMPO if the latter is correct. This needs to be clarified since it brings doubts about the pK_a value of the hydroperoxide and consequently about the O-H DBE. I am not sure the method employed to determine the pK_a, using different acids, provides a reliable value. Furthermore, the pK_a values of the acids used to make the plot do not seem correct, for example, methanesulphonic, trichloroacetic and trifluoroacetic have pK_a values <1. May I be missing something?

The pK_a value describes a thermodynamic equilibrium. Hence, the pK_a value is not only temperature dependent but also solvent dependent.

Besides the long established pK_a values determined in H₂O, the systematic

determination of pK_a values in non-aqueous solvents started later. The group of F. G. Bordwell started the systematic determination of pK_a values in DMSO.^{1,2} Later, the group of Ivo Leito made large contributions for the systematic determination of the acidity of acids^{3–6} and the basicity of bases^{7–9} in CH₃CN. A special focus of their research concerned the strong solvent dependence of the pK_a value. *E.g.*, the pK_a value of acetic acid is 4.75 in H₂O, 12.3 in DMSO, and 23.51 in CH₃CN.³ An interesting comparison for the solvents used in our study is given by the pK_a value of CF₃SO₃H that is 0.7 in CH₃CN but -11.4 in 1,2-dichloroethane.

Based on our initial submission, Reviewer #1 had suggested to determine the pK_a value of [(susan^{6-Me}){Fe^{III}(μ -1,2-OOH)(μ -O)Fe^{III}}]³⁺ and hence the BDFE of the coordinated μ -1,2-hydroperoxo ligand. We followed this suggestion in our revision and adapted a method for the determination of pK_a values from a recent publication of Larry Que (included in the rerevised manuscript).¹⁰ Comparing the quality of our pK_a dependence makes us confident in the determination of our pK_a value.

However, the determination of the pK_a value of a reactive compound is not as straightforward as that of a stable compound and requires some compromises. First, the solvent used for a pK_a value determination is restricted to solvents for which the pK_a values for the acids used have been determined. Second, the pK_a values for the acids used are usually determined at room temperature. The pK_a value determination for a reactive compounds requires use of lower temperatures for stabilization. This is an intrinsic issue that cannot be resolved and is inherent in such studies.

We determined the pK_a value of [(susan^{6-Me}){Fe^{III}(μ -1,2-OOH)(μ -O)Fe^{III}}]³⁺ by protonation of the parent complex [(susan^{6-Me}){Fe^{III}(μ -1,2-O₂)(μ -O)Fe^{III}}]²⁺ using several acids in CH₃CN at -40 °C. The pK_a values of the acids used in CH₃CN have been taken from the work of Leito *et al* (included in the revised manuscript).^{3–6} Here, Reviewer #1 might be missing that the pK_a values of MeSO₃H, Cl₃COOH, and F₃COOH are < 1 in water but much higher in CH₃CN.

The protonated complex [(susan^{6-Me}){Fe^{III}(μ -1,2-OOH)(μ -O)Fe^{III}}]³⁺ is not sufficient stable in CH₃CN at -40 °C. Further stabilization required lower temperatures. As CH₃CN has a melting point of -45 °C, lowering the temperature required to use mixtures with solvents that also do not react with the protonated complex [(susan^{6-Me}){Fe^{III}(μ -1,2-OOH)(μ -O)Fe^{III}}]³⁺. It turned out that the protonated complex [(susan^{6-Me}){Fe^{III}(μ -1,2-OOH)(μ -O)Fe^{III}}]³⁺ is quite stable at -60 °C in a 1:2 CH₃CN:CH₂Cl₂ solvent mixture. Hence, the reactivities of protonated complex [(susan^{6-Me}){Fe^{III}(μ -1,2-OOH)(μ -O)Fe^{III}}]³⁺ with PPh₃ and TEMPOH were studied at -60 °C in 1:2 CH₃CN:CH₂Cl₂.

Thus, we attribute the discrepancy that the protonated complex [(susan^{6-Me}){Fe^{III}(μ -1,2-OOH)(μ -O)₂Fe^{III}}]³⁺ protonates PPh₃ and TEMPOH despite its apparently higher pK_a value to the variation of the solvent: CH₃CN at -40 °C for the determination of the pK_a value and CH₃CN:CH₂Cl₂ (1:2) at -60 °C for studying the reactivity. As indicated above, pK_a values in CH₃CN and 1,2-dichloroethane (as a close analogue to CH₂Cl₂) differ

tremendously. Hence, the use of two third CH_2Cl_2 can change the pK_a value for some units. Moreover, the pK_a value for TEMPOH has only been reported for water (around 7 in ref. 60) and is hence not a good comparison.

In order to provide an experimental argument for this solvent dependence, we compared the effect by treating the parent complex $[(\text{susan}^{6-\text{Me}})\{\text{Fe}^{\text{III}}(\mu-1,2-\text{O}_2)(\mu-\text{O})_2\text{Fe}^{\text{III}}\}]^{3+}$ with either one equivalent of the acid HClO_4 or one equivalent of $[\text{HPPPh}_3](\text{BF}_4)$ under identical conditions in CH_3CN at $-40\text{ }^\circ\text{C}$. We have prepared the latter according to a literature procedure¹¹ (included in the rerevised manuscript).

The addition of either acid lead to a direct decrease of the $\mu-1,2$ -peroxo $\rightarrow\text{Fe}^{\text{III}}$ LMCT at 15400 cm^{-1} . Hence, both acids HClO_4 and $[\text{HPPPh}_3](\text{BF}_4)$ are able to protonate the parent complex $[(\text{susan}^{6-\text{Me}})\{\text{Fe}^{\text{III}}(\mu-1,2-\text{O}_2)(\mu-\text{O})\text{Fe}^{\text{III}}\}]^{2+}$ to $[(\text{susan}^{6-\text{Me}})\{\text{Fe}^{\text{III}}(\mu-1,2-\text{OOH})(\mu-\text{O})\text{Fe}^{\text{III}}\}]^{3+}$. This experimentally demonstrates that $[\text{HPPPh}_3]^+$ is more acidic than the protonated complex $[(\text{susan}^{6-\text{Me}})\{\text{Fe}^{\text{III}}(\mu-1,2-\text{OOH})(\mu-\text{O})\text{Fe}^{\text{III}}\}]^{3+}$ in CH_3CN at $-40\text{ }^\circ\text{C}$ in accordance with the lower pK_a value of 7.62 for $[\text{HPPPh}_3]^+$ than 9.5 for $[(\text{susan}^{6-\text{Me}})\{\text{Fe}^{\text{III}}(\mu-1,2-\text{OOH})(\mu-\text{O})\text{Fe}^{\text{III}}\}]^{3+}$. This new experiment hence strongly support our attribution of the inverted protonation of the PPh_3 by the protonated $[(\text{susan}^{6-\text{Me}})\{\text{Fe}^{\text{III}}(\mu-1,2-\text{OOH})(\mu-\text{O})\text{Fe}^{\text{III}}\}]^{3+}$ in $\text{CH}_3\text{CN}:\text{CH}_2\text{Cl}_2$ (1:2) at $-60\text{ }^\circ\text{C}$ to different solvent dependence of the respective pK_a values by going from CH_3CN to $\text{CH}_3\text{CN}:\text{CH}_2\text{Cl}_2$ (1:2).

In the light of this discussion, we have revised our manuscript in the Supplementary Information:

Supplementary Fig. 30. Solvent-dependence of the acidity: addition of one equivalent HClO_4 to a solution of $[(\text{susan}^{6-\text{Me}})\{\text{Fe}^{\text{III}}(\mu-\text{O})(\mu-\text{O}_2)\text{Fe}^{\text{III}}\}]^{2+}$ (0.84 mM) at $-40\text{ }^\circ\text{C}$ in CH_3CN and addition of one equivalent $[\text{HPPPh}_3]\text{BF}_4$ to a solution of $[(\text{susan}^{6-\text{Me}})\{\text{Fe}^{\text{III}}(\mu-\text{O})(\mu-\text{O}_2)\text{Fe}^{\text{III}}\}]^{2+}$ (0.85 mM) at $-40\text{ }^\circ\text{C}$ in CH_3CN .

The protonated $[(\text{susan}^{6-\text{Me}})\{\text{Fe}^{\text{III}}(\mu\text{-}1,2\text{-OOH})(\mu\text{-O})_2\text{Fe}^{\text{III}}\}]^{3+}$ was deprotonated by addition of PPh_3 in $\text{CH}_3\text{CN}:\text{CH}_2\text{Cl}_2$ (1:2) at -60°C (Supplementary Fig. 25) despite its apparently higher pK_a value (9.5 vs 7.62). However, these pK_a values were determined in CH_3CN . It was found that pK_a values are strongly solvent dependent.^{5-8,40-44} Hence, the inverted acidity observed in $\text{CH}_3\text{CN}:\text{CH}_2\text{Cl}_2$ (1:2) is attributed to a different solvent dependence of the pK_a values of these two acids. The addition of either one equivalent of the acid HClO_4 or one equivalent of $[\text{HPPh}_3](\text{BF}_4)$ under identical conditions in CH_3CN at -40°C lead to a direct decrease of the $\mu\text{-}1,2\text{-peroxo}\rightarrow\text{Fe}^{\text{III}}$ LMCT at 15400 cm^{-1} . Hence, both acids HClO_4 and $[\text{HPPh}_3](\text{BF}_4)$ are able to protonate the parent complex $[(\text{susan}^{6-\text{Me}})\{\text{Fe}^{\text{III}}(\mu\text{-}1,2\text{-O}_2)(\mu\text{-O})_2\text{Fe}^{\text{III}}\}]^{2+}$ to $[(\text{susan}^{6-\text{Me}})\{\text{Fe}^{\text{III}}(\mu\text{-}1,2\text{-OOH})(\mu\text{-O})_2\text{Fe}^{\text{III}}\}]^{3+}$. These experiments demonstrate the correct higher acidity of $[\text{HPPh}_3]^+$ than of $[(\text{susan}^{6-\text{Me}})\{\text{Fe}^{\text{III}}(\mu\text{-}1,2\text{-OOH})(\mu\text{-O})_2\text{Fe}^{\text{III}}\}]^{3+}$ in pure CH_3CN . The same applies to the reaction with TEMPOH (Supplementary Fig. 28) for which a pK_a value was only determined in water.⁵⁰

1. Matthews, W. S. *et al.* Equilibrium acidities of carbon acids. VI. Establishment of an absolute scale of acidities in dimethyl sulfoxide solution. *J. Am. Chem. Soc.* **97**, 7006–7014 (1975).
2. Bordwell, F. G. Equilibrium acidities in dimethyl sulfoxide solution. *Acc. Chem. Res.* **21**, 456–463 (1988).
3. Kütt, A. *et al.* A comprehensive self-consistent spectrophotometric acidity scale of neutral Brønsted acids in acetonitrile. *J. Org. Chem.* **71**, 2829–2838 (2006).
4. Eckert, F. *et al.* Prediction of acidity in acetonitrile solution with COSMO-RS. *J. Comput. Chem.* **30**, 799–810 (2009).
5. Kütt, A. *et al.* Equilibrium acidities of superacids. *J. Org. Chem.* **76**, 391–395 (2011).
6. Kütt, A. *et al.* Strengths of Acids in Acetonitrile. *Eur. J. Org. Chem.* **2021**, 1407–1419 (2021).

7. Kaljurand, I. *et al.* Extension of the self-consistent spectrophotometric basicity scale in acetonitrile to a full span of 28 pKa units: unification of different basicity scales. *J. Org. Chem.* **70**, 1019–1028 (2005).
8. Haav, K., Saame, J., Kütt, A. & Leito, I. Basicity of Phosphanes and Diphosphanes in Acetonitrile. *Eur. J. Org. Chem.* **2012**, 2167–2172 (2012).
9. Tshepelevitsh, S. *et al.* On the Basicity of Organic Bases in Different Media. *Eur. J. Org. Chem.* **2019**, 6735–6748 (2019).
10. Crossland, P. M., Guo, Y. & Que, L. Spontaneous Formation of an Fe/Mn Diamond Core: Models for the Fe/Mn Sites in Class 1c Ribonucleotide Reductases. *Inorg. Chem.* **60**, 8710–8721 (2021).
11. Hausoul, P. J. C. *et al.* Facile access to key reactive intermediates in the Pd/PR₃-catalyzed telomerization of 1,3-butadiene. *Angew. Chem. Int. Ed.* **49**, 7972–7975 (2010).

Response to Reviewer #2

the authors have addressed all issues raised

While we are grateful for this summarizing comment of Reviewer #2, we are still not in favor of incorporating the references suggested of Reviewer #2 into our manuscript and we would like to explain why.

the reference to Toftlund is:

Synthesis and Magnetic Properties of a μ -Oxo-di(μ -acetato)manganese(III) Complex of a Strapped Tripodal Pyridylamine Ligand N,N,N',N'-Tetrakis(2-pyridylmethyl)-1,3-propanediamine. A Model for the Mn²⁺ Site of Mn-Catalase Enzymes. Toftlund, Hans; Markiewicz, Andrew; Murray, Keith S.
Pages: 443-446.

and Armstrong

Gamelin, D. R.; Kirk, M. L.; Solomon, E. I.; Stemmler, T. L.; Penner-Hahn, J. E.;
Pal, S.; Armstrong, W. H. J. Am. Chem. Soc. 1994, 116, 2392-2399.

First of all, there is a restriction of the number of references for this manuscript that is 70

By preparing the revised manuscript with the inclusion of necessary references for the reactivity studies, we arrived initially to a number of references of 86 and it was difficult to reduce this number to 70 without missing some really essential references.

The second argument refers to the ligands of the suggested references. In both publications,^{[1],[2]} (the reference provided by Reviewer #2^[3] is not the original synthetic reference of Armstrong but a detailed electronic structure study by Solomon, Armstrong, and Penner-Hahn on the already published complex of Armstrong) the ligand tpen was used for the synthesis of dinuclear manganese complexes.

Although there must be attributed quite some similarities between these two ligands, the ligand *tpen* is a hexadentate N-donor ligand while our ligand *susan*^{6-Me} is an octadentate N-donor ligand. However, the ligand *tpen* is a widely used ligand in coordination chemistry. A search in CSD provided 213 hits where the ligand *tpen* or a derivative of it coordinates as a mononucleating or a dinucleating ligand.

With regards to the limitation in the number of references, the difference of the ligand *susan*^{6-Me} to the ligand *tpen*, the widespread use of the ligand *tpen* by various authors, the non-iron based nature of the references suggested, and the point that many more relevant references on diiron complexes and their reactivity cannot be cited due to the limitations in references, we are not in favor of adding the two suggested references as this would imply the consequence that two more relevant references with regard to this manuscript has to be deleted.

- [1] H. Toftlund, A. Markiewicz, K. S. Murray, Synthesis and Magnetic-Properties of a μ -Oxo-Di(μ -Acetato)Manganese(II) Complex of a Strapped Tripodal Pyridylamine Ligand N ;N' ;N' ;N'-Tetrakis(2-Pyridylmethyl)-1 ;3-Propanediamine, *Acta Chem. Scand.* **1990**, *44*, 443–446.
- [2] S. Pal, J. W. Gohdes, W. C. A. Wilisch, W. H. Armstrong, Synthesis, structure, and properties of a [manganese] complex that consists of an $\{Mn_2O_2(O_2CCH_3)\}_2^{2+}$ core and a spanning hexadentate ligand, *Inorg. Chem.* **1992**, *31*, 713–716.
- [3] D. R. Gamelin, M. L. Kirk, T. L. Stemmler, S. Pal, W. H. Armstrong, J. E. Pennerhahn, E. I. Solomon, Electronic-Structure and Spectroscopy of Manganese Catalase and Di- μ -Oxo [Mn(II)Mn(IV)] Model Complexes, *J. Am. Chem. Soc.* **1994**, *116*, 2392–2399.

Response to Reviewer #3

The authors have made a very good job in revising their manuscript thoroughly and extensively. They have fully clarified the obscure points in the original version and this should make following the work easy for any reader.

Eventually, the paper provides a substantiated and rationalized comparison of three biologically relevant forms of diron centers involved in important oxidative transformations. As such it is likely to be cited as a reference in the field.

I recommend publication.

We are thankful for these favorable comments of Reviewer #3 on our revisions of the manuscript.

Response to Reviewer #4

I have reviewed the revised version of the manuscript by Glaser et al. Still, it needs lots of concentration to understand the technical details described in the main text, however, I find everything to be correct and precisely described, and I acknowledge that the authors have made an effort to improve the readability.

My main concern in my original review was the crystal structure of the decay product depicted in a ball-and-stick model in Figure 6b. This structure has now been shown in Figure S1 c with both disorder components separately. This is fine and indicates no further problems. I would only suggest to refer to Figure S1 in the caption of Figure 6.

Otherwise, I believe that the manuscript is ready for publication.

We are thankful for these favorable comments of Reviewer #4 on our revisions of the manuscript.

In the light of the reviewer suggestion, we have changed the caption of Figure 6:

Figure 6. Reactivity studies. a) Square scheme showing the PCET thermochemistry of $[(\text{susan}^{6\text{-Me}})\{\text{Fe}^{\text{III}}(\mu\text{-O})(\mu\text{-}1,2\text{-OOH})\text{Fe}^{\text{III}}\}]^{3+}$ in CH_3CN . b) Molecular structure of the decay product of $[(\text{susan}^{6\text{-Me}})\{\text{Fe}^{\text{III}}(\mu\text{-O})(\mu\text{-}1,2\text{-O}_2)\text{Fe}^{\text{III}}\}]^{2+}$ including the ligand disorders (please see also Supplementary Figure 1c): at Fe1 80% carboxylate and 20% hydroxide (shown with dotted lines), at Fe2 35% benzylalcoholato and 65% hydroxide (shown with dotted lines). c) Space-filling model of $[(\text{susan}^{6\text{-Me}})\{\text{Fe}^{\text{III}}(\mu\text{-O})(\mu\text{-}1,2\text{-O}_2)\text{Fe}^{\text{III}}\}]^{2+}$ to illustrate the encapsulation of the peroxy ligand by a CH_2 group (right Fe) and a 6-methyl group (left Fe).

REVIEWER COMMENTS

Reviewer #1 (Remarks to the Author):

The authors have made a convincing response to my original concerns regarding the pKa's obtained. The arguments regarding the solvent dependence of pKa's in the response letter, in the manuscript and SI are convincing. I find particularly nice and definitive the experiment with the protonated triphenylphosphine. Publication in the current form is therefore recommended.